# Immune modulation by complement receptor 3-dependent human monocyte TGF-β1-transporting vesicles

Luke D. Halder[1], Emeraldo A. H. Jo[1], Mohammad Z. Hasan[1], Marta Ferreira-Gomes[2], Thomas Krüger[3], Martin Westermann[4], Diana I. Palme[1], Günter Rambach[5], Niklas Beyersdorf[6], Cornelia Speth [5], Ilse D. Jacobsen [7,8], Olaf Kniemeyer[3], Berit Jungnickel[2], Peter F. Zipfel [1,8] & Christine Skerka [1✉]

Extracellular vesicles have an important function in cellular communication. Here, we show that human and mouse monocytes release TGF-β1-transporting vesicles in response to the pathogenic fungus *Candida albicans*. Soluble β-glucan from *C. albicans* binds to complement receptor 3 (CR3, also known as CD11b/CD18) on monocytes and induces the release of TGF-β1-transporting vesicles. CR3-dependence is demonstrated using CR3-deficient (CD11b knockout) monocytes generated by CRISPR-CAS9 genome editing and isolated from CR3-deficient (CD11b knockout) mice. These vesicles reduce the pro-inflammatory response in human M1-macrophages as well as in whole blood. Binding of the vesicle-transported TGF-β1 to the TGF-β receptor inhibits *IL1B* transcription via the SMAD7 pathway in whole blood and induces *TGFB1* transcription in endothelial cells, which is resolved upon TGF-β1 inhibition. Notably, human complement-opsonized apoptotic bodies induce production of similar TGF-β1-transporting vesicles in monocytes, suggesting that the early immune response might be suppressed through this CR3-dependent anti-inflammatory vesicle pathway.

[1] Department of Infection Biology, Leibniz Institute for Natural Product Research and Infection Biology, 07745 Jena, Germany. [2] Department of Cell Biology, Institute of Biochemistry and Biophysics, Friedrich Schiller University, 07745 Jena, Germany. [3] Department of Molecular and Applied Microbiology, Leibniz Institute for Natural Product Research and Infection Biology, 07745 Jena, Germany. [4] Electron Microscopy Center, University Hospital Jena, 07743 Jena, Germany. [5] Division of Hygiene and Medical Microbiology, Medical University of Innsbruck, A-6020 Innsbruck, Austria. [6] Institute for Virology and Immunobiology, University of Würzburg, 97070 Würzburg, Germany. [7] Research Group Microbial Immunology, Leibniz Institute for Natural Product Research and Infection Biology, 07745 Jena, Germany. [8] Friedrich Schiller University, 07743 Jena, Germany. ✉email: christine.skerka@hki-jena.de

Extracellular vesicles (EVs) are released by a variety of human cells, and are important mediators in the coordination of the immune response in order to maintain host homeostasis[1]. Depending on the stimulus, EVs transport a combination of nucleic acids, proteins, and lipids to other cells and have been implicated in certain pathological conditions, particularly microbial infections and cancer[2,3]. *C. albicans* infections are a severe threat to life for immunocompromised persons, including patients who received an organ transplant, who are undergoing anticancer therapy, or who are infected with human immunodeficiency virus (HIV), as well as patients who have experienced major trauma or have extended stays in the intensive care unit[4]. The pathogenicity of *C. albicans* depends on a broad range of virulence factors[5], and the fungus has developed evasion mechanisms to survive in the human host[6]. In the process of systemic infection, *C. albicans* is recognized by immune cells due to the presentation of pathogen-associated molecular patterns (PAMPs), resulting in the initiation of a series of immune response mechanisms. β-glucan, which has been described as a major recognition molecule of *C. albicans*, is mostly detected via the dectin-1 receptor[7]. Following recognition, innate immune responses against *C. albicans* include complement activation, phagocytosis, reactive oxygen species generation, pro-inflammatory cytokine release, and extracellular trap formation[8], but whether immune cells respond to in vivo fungal infection by generation of human EVs is yet unknown.

This study provides insight into the immunomodulatory properties of TGF-β1-transporting EVs that are generated by monocytes in response to the human pathogenic fungus *C. albicans*, as well as human apoptotic cells.

## Results

### *C. albicans* induces vesicle release from human blood monocytes.
Human monocytes directly recognize *C. albicans* and react in multiple ways to the fungus. They take up fungal cells by phagocytosis; release DNA traps[9], similar to neutrophils, to immobilize the fungus; and secrete toxic reactive oxygen species. As monocytes also produce vesicles to communicate with other cells[10], we addressed the question whether *C. albicans* induces vesicle release in monocytes. Human blood monocytes were isolated from buffy coats by magnetic sorting of CD14-positive cells (~95% purity), and incubated with complement-pre-opsonized *C. albicans* on a coverslip. After 1 h of incubation, the cells were fixed onto a microscopy slide, and the monocytes were monitored for the presence of vesicles using the previously described vesicle marker tetraspanin (CD63)[11]. Monocytes alone without *C. albicans* showed several vesicles, which predominantly surrounded the nucleus (Fig. 1a). When monocytes were incubated with *C. albicans*, vesicle formation substantially increased. Again, vesicles surrounded the nucleus, but were also found extracellularly, indicating vesicle release. Vesicles formed in response to *C. albicans* are referred to from here on as opsonized *Candida*-induced monocytic extracellular vesicles (MEVs$_{Ca}$) to distinguish them from the monocytic extracellular vesicles (MEVs) that are spontaneously produced in the absence of *C. albicans*.

To follow vesicle formation in real time, CD14$^+$ monocytes in the presence of opsonized *C. albicans* were tracked by live cell imaging in culture dishes using nucleic acid staining——Sytox Orange, which does not penetrate living cells but can penetrate extracellular vesicles. Live cell imaging revealed phagocytosis of *C. albicans* by monocytes within minutes and generation of nucleic acid-containing vesicles. Release of vesicles was observed after ~20–40 min (Fig. 1b). Vesicle generation and release from monocytes in presence of *C. albicans* was captured in real time

using dynamic light-scattering microscopy (DLSM) (Supplementary Video 1), confirming fast release of generated vesicle. To track vesicle generation by monocytes under more physiologic conditions, live cell imaging of monocytes was performed in an ex vivo whole-blood model system. Whole blood was infected with *C. albicans*, and monocytes were stained with anti-CD14 and vesicles were stained with anti-CD63. Vesicle generation was seen within 10 min after infection, and increased after 20 min (Fig. 1c). Released vesicles were also confirmed by scanning electron microscopy (SEM) (Fig. 1d). In summary, monocytes treated with opsonized *C. albicans* released EVs within 1 h after infection. In all subsequent experiments, *C. albicans* infection was performed for 1 h, unless otherwise indicated.

### MEVs$_{Ca}$ are double-layered vesicles.
For detailed characterization, MEVs$_{Ca}$ generated by isolated human blood monocytes in response to opsonized *C. albicans* were isolated using a polymer precipitation method. These vesicles were analyzed for their number and size by measuring the Brownian movement of vesicles in suspension using DLSM (Fig. 2a). The number of MEVs$_{Ca}$ harvested from *C. albicans*-infected monocytes ($5 \times 10^5$) was about ten times higher than the number of MEVs harvested from the same number of uninfected monocytes ($5 \times 10^5$) (Fig. 2b). Thus, monocytes release substantially more vesicles upon infection with the fungus. Five major populations of MEVs were identified, with sizes ranging from 50 to 450 nm, and three major populations of MEVs$_{Ca}$ were observed, with sizes ranging from 50 to 300 nm (Fig. 2c). For further characterization, cryogenic electron microscopy (Cryo-EM) and freeze-fracture electron microscopy (FFEM) were performed on MEVs$_{Ca}$. Cryo-EM verified the presence of 200 nm vesicles, and showed a double-layered membrane (Fig. 2d). Under FFEM, which revealed fractured concave and convex vesicles, the inner and outer membranes became visible (Fig. 2e).

In addition, the composition of MEVs and MEVs$_{Ca}$ was determined using label-free liquid chromatography-tandem mass spectrometry (LC-MS/MS)-based proteomics. Proteins were analyzed from vesicles derived from the same number of infected and control monocytes ($1 \times 10^7$) after tryptic digest. MEVs$_{Ca}$ showed a significant increased level of 361 proteins compared with MEVs, while 29 proteins were significantly decreased in abundance (Fig. 2f). The significantly increased proteins were further categorized using Gene Ontology enrichment analysis for biological process, molecular function, and cellular compartment. Most (209) of the proteins with higher abundance in MEVs$_{Ca}$ were extracellular exosome-related proteins (Fig. 2g). Heat shock proteins, along with histone fragments commonly found in vesicles[12], were found in MEVs and MEVs$_{Ca}$. The presence of CD14 in both MEVs and MEVs$_{Ca}$ confirmed their monocytic origin. In addition, myeloid lineage marker such as CD11b, complement receptor type I (CR1), and Toll-like receptor 2 (TLR2) were increased in MEVs$_{Ca}$ by 2-, 8-, and 11-fold, respectively (Fig. 2j). In addition, the tetraspanin CD9 was also identified on MEVs$_{Ca}$, together with commonly found annexins. Kyoto Encyclopedia of Genes and Genomes (KEGG) pathway enrichment analysis showed a high abundance of physiologically important complement pathway proteins (Fig. 2h, i, j). As cytokines are important regulators of the immune system and are difficult to detect by mass spectroscopy[13], cytokine levels in MEVs and MEVs$_{Ca}$ were determined by sandwich ELISA. Substantial amounts of transforming growth factor-β1 (TGF-β1) were detected in MEVs$_{Ca}$ derived from infected monocytes but not in MEVs derived from the same number of uninfected monocytes (Fig. 2k). By contrast, levels of IL-6 (Fig. 2l), IL-10, and IL-1β (not shown) were very low or were not detected. To

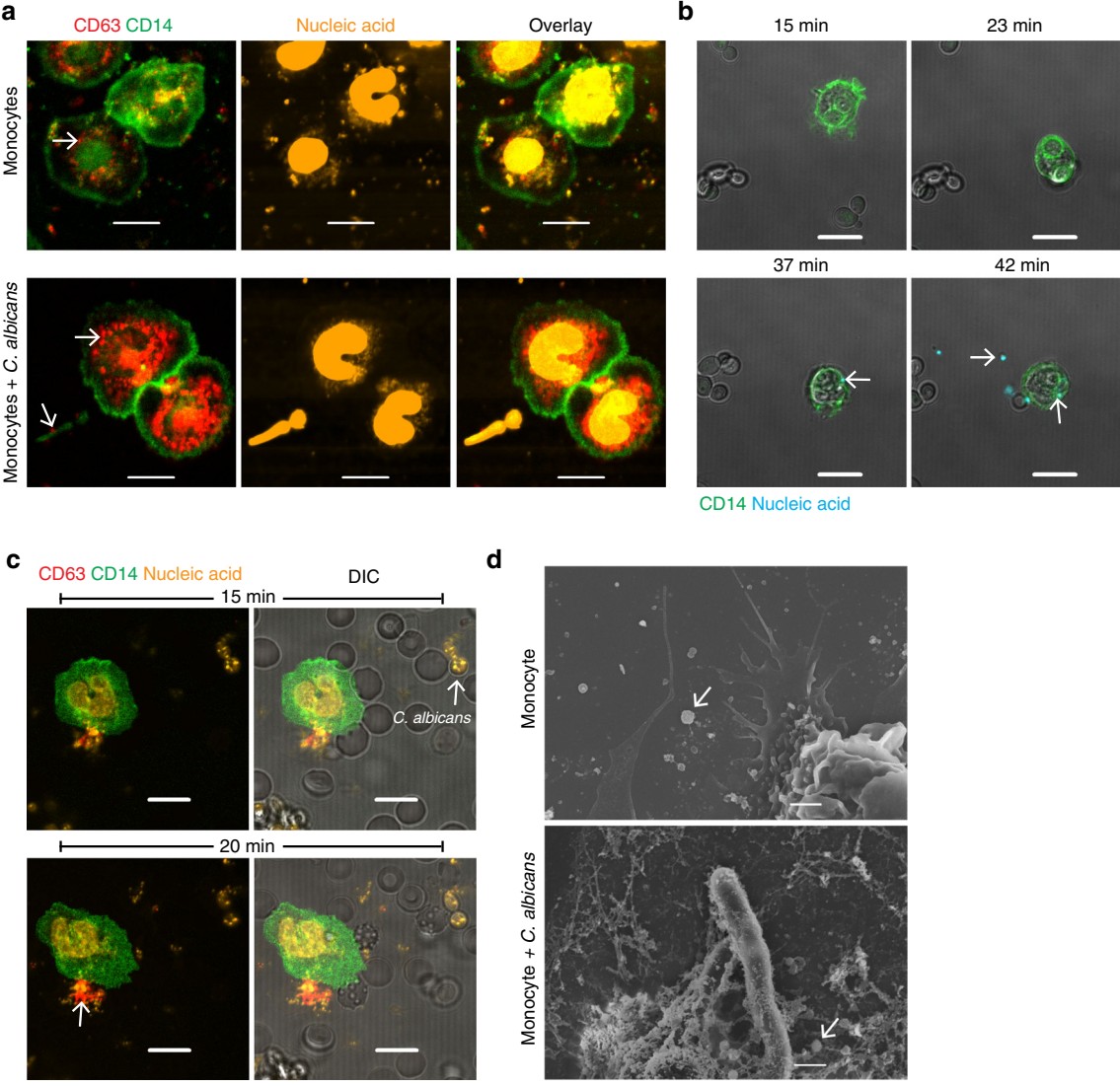

**Fig. 1 *C. albicans* induces vesicle release from human blood monocytes. a** Vesicle formation (arrow) increases in *C. albicans*-infected monocytes using confocal laser-scanning microscopy (CLSM). Staining– red: CD63, green: CD14, and orange: nucleic acids after 1 h of co-incubation. Bars: 5 μm. Representative data of five independent experiments are shown. **b** Release of EVs from the infected monocytes visualized by live cell imaging using CLSM. After phagocytosis (23 min) of *C. albicans*, the EVs (42 min) (arrows) are released. Staining– green: CD14 and cyan: nucleic acids. Bars: 10 μm. Representative data of three independent experiments are shown. **c** Infected monocytes in an ex vivo whole-blood model show early (20 min) release of MEVs$_{Ca}$ (arrow) by live cell imaging (CLSM). Staining– red: CD63, green: CD14, and orange: nucleic acids. Bars: 10 μm. Representative data of five independent experiments are shown. **d** Scanning electron microscopy (SEM) confirms the presence of MEVs and MEVs$_{Ca}$ released by untreated and infected monocytes (arrows) respectively. Bars: 1 μm. Representative data of three independent experiments are shown.

exclude extracellular proteins in vesicle polymer precipitates, the results were confirmed also with EVs isolated by ultracentrifugation and size-exclusion chromatography (Supplementary Fig. 1). Furthermore, vesicle markers CD9 as well as HSP90 were identified on TGF-β1-transporting vesicles (70–130 nm) by size-exclusion chromatography, which in addition, characterized these vesicles as exosomes. Presence of few TGF-β1-transporting vesicles without vesicle markers is explained by simultaneous use of different antibodies.

**MEVs$_{Ca}$ transport TGF-β1**. To verify the presence of TGF-β1 in MEVs$_{Ca}$, monocytes were incubated with *C. albicans* for 1 h on cover slips, and cells were fixed and stained with an antibody against TGF-β1. *C. albicans*-infected monocytes but not uninfected monocytes showed significant production of MEVs$_{Ca}$ of 100–200 nm in size by confocal laser-scanning microscopy

(CLSM) (Fig. 3a, b; Supplementary Fig. 2a). To visualize MEVs$_{Ca}$ in more detail, TGF-β1 was labeled with immunogold, and the vesicles were analyzed by SEM. MEVs$_{Ca}$ but not MEVs showed gold labeling on the surface, demonstrating the presence of TGF-β1 on the outer membrane. The size of MEVs and MEVs$_{Ca}$ was ~100–300 nm (Fig. 3c), which was in the same range as the size measured by DLSM (Fig. 2c).

To observe MEVs$_{Ca}$ under physiological conditions, MEVs$_{Ca}$ were also tracked ex vivo in *C. albicans*-infected whole blood using live cell imaging and CLSM. Monocytes were tracked over time with anti-CD14 labeling, and vesicles were tracked with anti-TGF-β1 labeling. Shortly after infection (15 min), the monocytes in the blood began to form TGF-β1-transporting vesicles intracellularly, which can be seen as red dots within the monocytes (Fig. 3d). When *C. albicans* started forming hyphae (45 min later), TGF-β1-transporting vesicles from the same

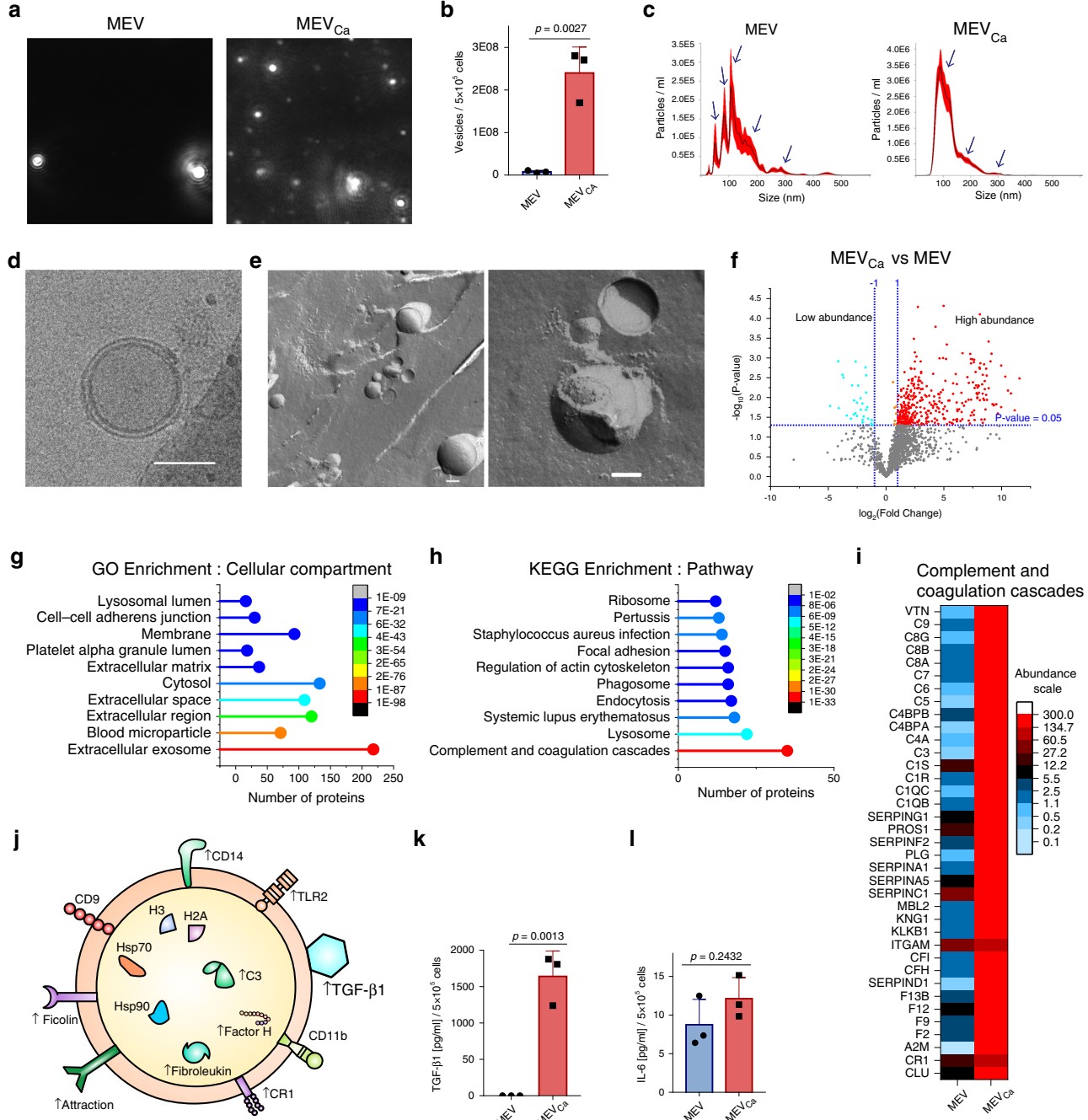

**Fig. 2 MEV_Cas are double-layered vesicles. a** Tracking of EVs by dynamic light-scattering microscopy (DLSM). EVs were isolated from $5 \times 10^5$ uninfected or opsonized *C. albicans*-infected monocytes (MEVs or MEVs_Ca, respectively) by polymer precipitation. Representative data of three independent experiments (three donors) are shown. **b** MEVs_Ca are significantly increased compared with MEVs (data are presented as mean values $+/-$ SD, $p = 0.0027$, unpaired two-tailed *t* test, $n = 3$ different donors). EVs isolated from same number of infected or uninfected monocytes were counted by DLSM. **c** Size distribution of MEVs and MEVs_Ca as determined by DLSM using NanoSight NTA 3.2 software. Graphs were generated by overlaying the size distribution of MEVs and MEVs_Ca from $n = 3$ donors. **d** The double-membrane structure and round shape of the MEVs_Ca are visible by cryogenic electron microscopy (Cryo-EM). Bars: 100 nm. **e** MEVs_Ca structure and shape is confirmed by freeze-fracture electron microscopy (FFEM). Bars: 100 nm. **f** MEVs_Ca show significantly higher protein content compared with MEVs ($p < 0.05$, unpaired two-tailed *t* test, $n = 3$ different donors). Proteins from MEVs and MEVs_Ca (each from $1 \times 10^7$ monocytes) were detected by label-free LC-MS/MS-based proteomics. **g** Most of the proteins enriched in MEVs_Ca compared with MEVs are extracellular exosome-related proteins (Gene Ontology (GO) enrichment analysis). **h** Complement pathway proteins were significantly increased in MEVs_Ca (Kyoto Encyclopedia of Genes and Genomes (KEGG) enrichment analysis). **i** Heatmap of complement and coagulation proteins upregulated in MEV_Cas. Comparisons in **g-i** rely on protein content of MEVs_Ca from three different donors. **j** Diagram of an MEVs_Ca showing the presence of receptors, marker proteins, and cytokines detected in the current study. **k** TGF-β1 (data are presented as mean values $+/-$ SD, $p = 0.0013$, unpaired two-tailed *t* test, $n = 3$ donors), **l** but not IL-6, is significantly increased in MEVs_Ca compared with MEVs (data are presented as mean values $+/-$ SD, $p = 0.2432$, unpaired two-tailed *t* test, $n = 3$ different donors). Cytokines from MEVs and MEVs_Ca (each isolated from monocytes ($5 \times 10^5$)) were determined by ELISA.

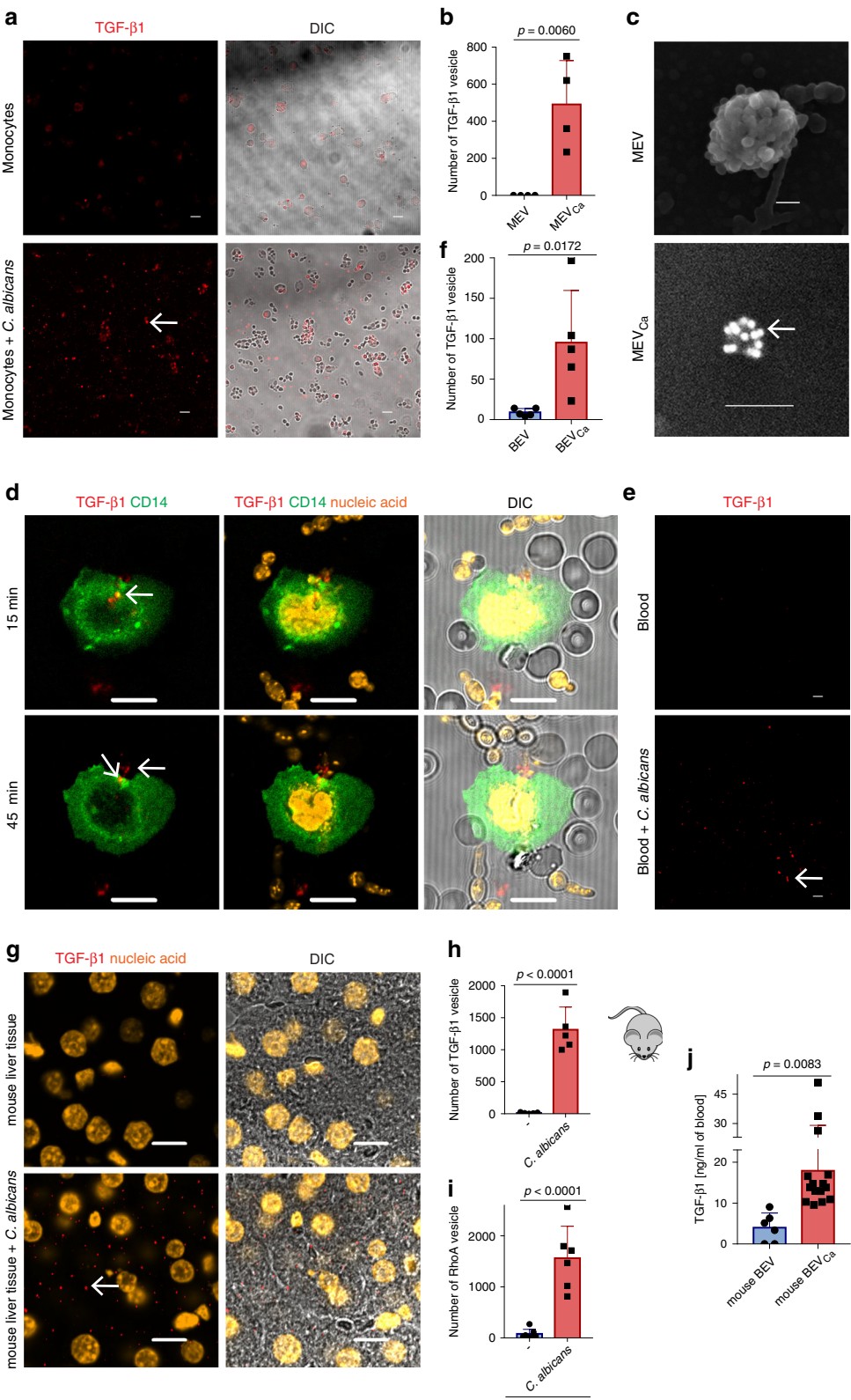

monocytes were detected extracellularly (Fig. 3d). After 1 h of infection, significantly increased numbers of TGF-β1-transporting vesicles (BEVs$_{Ca}$) were detected in the blood, in contrast to uninfected blood (Fig. 3e, f).

To confirm the formation of TGF-β1-transporting vesicles in vivo, mice were infected with *C. albicans* and killed 1 day later, and extensively perfused liver tissue was stained for TGF-β1.

TGF-β1-transporting vesicles were abundant in the liver tissue of infected, but not control, mice (Fig. 3g, h). Similarly, the vesicle marker RhoA was abundant (Supplementary Fig. 2b; Fig. 3i). To further confirm the generation of TGF-β1-transporting vesicles, blood was collected from the mice, vesicles were isolated, and the TGF-β1 content of the vesicles was analyzed by ELISA. The results confirmed a significant ($p = 0.0083$, unpaired two-tailed

**Fig. 3 MEV$_{sCa}$ transport TGF-β1. a** MEVS$_{Ca}$ transporting TGF-β1 (arrows) are frequently observed by CLSM when monocytes are incubated with *C. albicans*. Representative data of n = 4 experiments (four donors) are shown. Staining– red: TGF-β1. Bars: 10 μm. **b** Expression of TGF-β1 in MEVS$_{Ca}$ compared with MEVs. TGF-β1-positive EVs were counted using Image J software (data are presented as mean values +/− SD, p = 0.0060, unpaired two-tailed *t* test, n = 4 different donors). **c** TGF-β1 (arrow) is detected on the membranes of MEVS$_{Ca}$ but not MEVs by immunogold labeling of TGF-β1 using SEM. Bars: 100 nm. **d** Monocytes generate (15 min) and release (45 min) MEVS$_{Ca}$ during whole-blood infection. MEVS$_{Ca}$ were tracked by live cell imaging using CLSM. Staining– red: TGF-β1, green: CD14, and orange: nucleic acids. Bars: 10 μm. Representative data of three experiments (three donors) are shown. **e** TGF-β1-transporting vesicle tracking in a whole-blood ex vivo infection model by live cell imaging using CLSM. Staining– red: TGF-β1. Bars: 10 μm. Representative data of n = 5 experiments (five donors) are shown. **f** Blood infected with *C. albicans* shows a significant increase of TGF-β1-transporting vesicles. TGF-β1-positive EVs were counted using Image J software (data are presented as mean values +/− SD, p = 0.0172, unpaired two-tailed *t* test, n = 5 donors). **g** High frequency of TGF-β1-transporting vesicles in liver tissue of *C. albicans*-infected mice (24 h). Staining– red: TGF-β1 and orange: nucleic acids. Bars: 10 μm. Representative immunohistochemistry of n = 3 different experiments (three donors). **h** TGF-β1 and **i** Rho A-labeled vesicles are counted using Image J (data are presented as mean values +/− SD, p < 0.0001, unpaired two-tailed *t* test, n = 3 donors, five images for TGF-β1 and six images for Rho A). **j** High content of TGF-β1-transporting vesicles in the blood of *C. albicans*-infected mice (24 h) (data are presented as mean values +/− SD, p = 0.0083, unpaired two-tailed *t* test, n = 6 uninfected and n = 14 infected mice). Blood vesicles were isolated, and TGF-β1 was detected by ELISA.

*t* test, n = 6–14) increase in TGF-β1-transporting vesicles in the blood of infected mice compared with blood from untreated control mice (Fig. 3j).

TGF-β1-transporting vesicles were also purified by positive selection and analysis revealed four different vesicle populations ranging from 30 to 350 nm, with predominant population residing under 100 nm (Supplementary Fig. 2c, d). CD9 transporting vesicles were isolated from selected TGF-β1-transporting vesicles resulting in a fraction of CD9 plus TGF-β1-transporting vesicles (Supplementary Fig. 2e–g). These results confirm the presence of TGF-β1 on vesicles.

To understand the release of TGF-β1-transporting vesicles, infected (30 min) and control monocytes were fixed and stained for intracellular TGF-β1. Infected monocytes showed substantially lower concentrations of intracellular TGF-β1 compared with control monocytes, suggesting that TGF-β1 is released on vesicles from infected monocytes. Furthermore, comparative qPCR revealed no upregulation of *TGFB1* transcription in infected monocytes (Supplementary Fig. 2h–j).

**Fungal β-glucan interaction with CR3 induces MEVS$_{Ca}$ release.** To determine how TGF-β1-transporting MEVS$_{Ca}$ are generated, we focused on CR3, as CR3 has been described as a key recognition receptor for pathogens[14]. Therefore, CR3 was blocked with the anti-cholesterol drug simvastatin[15]. When CR3-blocked monocytes were infected with opsonized *C. albicans*, induction of TGF-β1-transporting vesicles was significantly reduced (p = 0.0019, unpaired two-tailed *t* test, n = 3) (Fig. 4a). To further verify the involvement of the CR3 receptor in this process, CR3 receptor subunit CD11b was knocked out in monocytic THP-1 cells using the clustered regularly interspersed short palindromic repeats (CRISPR)/CRISPR-associated protein 9 (CAS9) system. CD11b knockout (KO) THP-1 cells lost expression of CR3, as shown by flow cytometry and western blot analysis (Fig. 4b).

Like human blood monocytes, THP-1 cells expressing CR3 released TGF-β1-transporting vesicles in response to opsonized *C. albicans*. However, CD11b KO THP-1 cells completely failed to release TGF-β1-containing vesicles (Fig. 4c).

In addition, CR3 dependence was investigated in an ex vivo system using whole blood from wild-type and CR3-deficient (CD11b KO) mice (Supplementary Fig. 3a, b). Peripheral blood cells (1 × 10$^7$) from mice of each genotype were infected with *C. albicans* for 1 h, and vesicles were subsequently isolated and assayed for TGF-β1. Unlike blood cells from wild-type mice, cells from CD11b KO mice produced small amounts of TGF-β1-transporting vesicles (Fig. 4d). To connect these results to monocytes, mouse monocytes were generated from isolated bone marrow cells from both wild-type and CD11b KO mice (Supplementary Fig. 3a–d), and MEVs and MEVS$_{Ca}$ were isolated.

No significant difference in TGF-β1 content was observed between MEVs and MEVS$_{Ca}$ derived from CD11b KO mouse monocytes, but MEVS$_{Ca}$ generated from wild-type monocytes showed a significant 4–6-fold increase in TGF-β1 levels compared with MEVs (p = 0.0021, unpaired two-tailed *t* test, n = 5) (Fig. 4e).

Having shown the central role of CR3 in the production of TGF-β1-transporting vesicles by *C. albicans*-infected monocytes, we were interested in identifying the ligand of CR3 responsible for this effect. As iC3b is deposited onto *C. albicans* due to opsonization, iC3b was used to induce TGF-β1 vesicle release by binding to the so-called I-domain of CR3 (Fig. 4f). Human blood monocytes were stimulated for 1 h with iC3b, and vesicles were isolated and assayed for their TGF-β1 content by ELISA. No significant difference was observed between MEVs and vesicles induced with iC3b, indicating that iC3b is not relevant in TGF-β1-containing vesicle release. As soluble β-glucan is commonly expressed on the fungus[16,17] and has been previously described to bind to the lectin-like site (LLS) of CR3[18], purified soluble β-glucan from *Saccharomyces cerevisiae* was used to induce vesicles (MEVS$_{Sc-sβG}$) in human monocytes. Similarly, soluble β-glucan was extracted/enriched from *C. albicans* and used to induce vesicles (MEVS$_{Ca-sβG}$). Both types of monocyte-derived vesicle showed significant TGF-β1 content (p = 0.0227, p = 0.0077, unpaired two-tailed *t* test, n = 3) (Fig. 4g). Similar to induction with whole *C. albicans* cells, the number of vesicles released from monocytes increased by about tenfold upon stimulation with soluble β-glucan from *C. albicans* (Fig. 4i). MEVS$_{Ca-sβG}$ were composed of six major vesicle populations, with sizes ranging from 50 nm to 650 nm (Fig. 4j). To determine whether β-glucan interacted directly with CR3, a proximity ligation assay (PLA) was performed. In the presence of *C. albicans*, soluble β-glucan on the *C. albicans* surface interacted with CD11b on the monocyte surface (Fig. 4k). When enriched soluble β-glucan was used instead of whole *C. albicans* cells in the same PLA, the interaction between soluble β-glucan and CD11b was confirmed. No fluorescence was detected from CR3-expressing monocytes alone in the absence of soluble β-glucan (Fig. 4k). iC3b also bound to CD11b on the monocyte (Supplementary Fig. 3e), and both iC3b and soluble β-glucan-induced reactive oxygen species (ROS) formation in monocytes upon binding (Supplementary Fig. 3f). Label-free LC-MS/MS-based proteomics was also used to determine the composition of β-glucan-induced vesicles. MEVS$_{Ca-sβG}$ showed a significant increase of 420 proteins compared with MEVs, while the level of 13 proteins were significantly decreased (Fig. 5a). According to a Gene Ontology enrichment analysis for cellular compartments, again most of the identified proteins (238) were extracellular exosome-related proteins, similarly to the proteins identified in whole *C.*

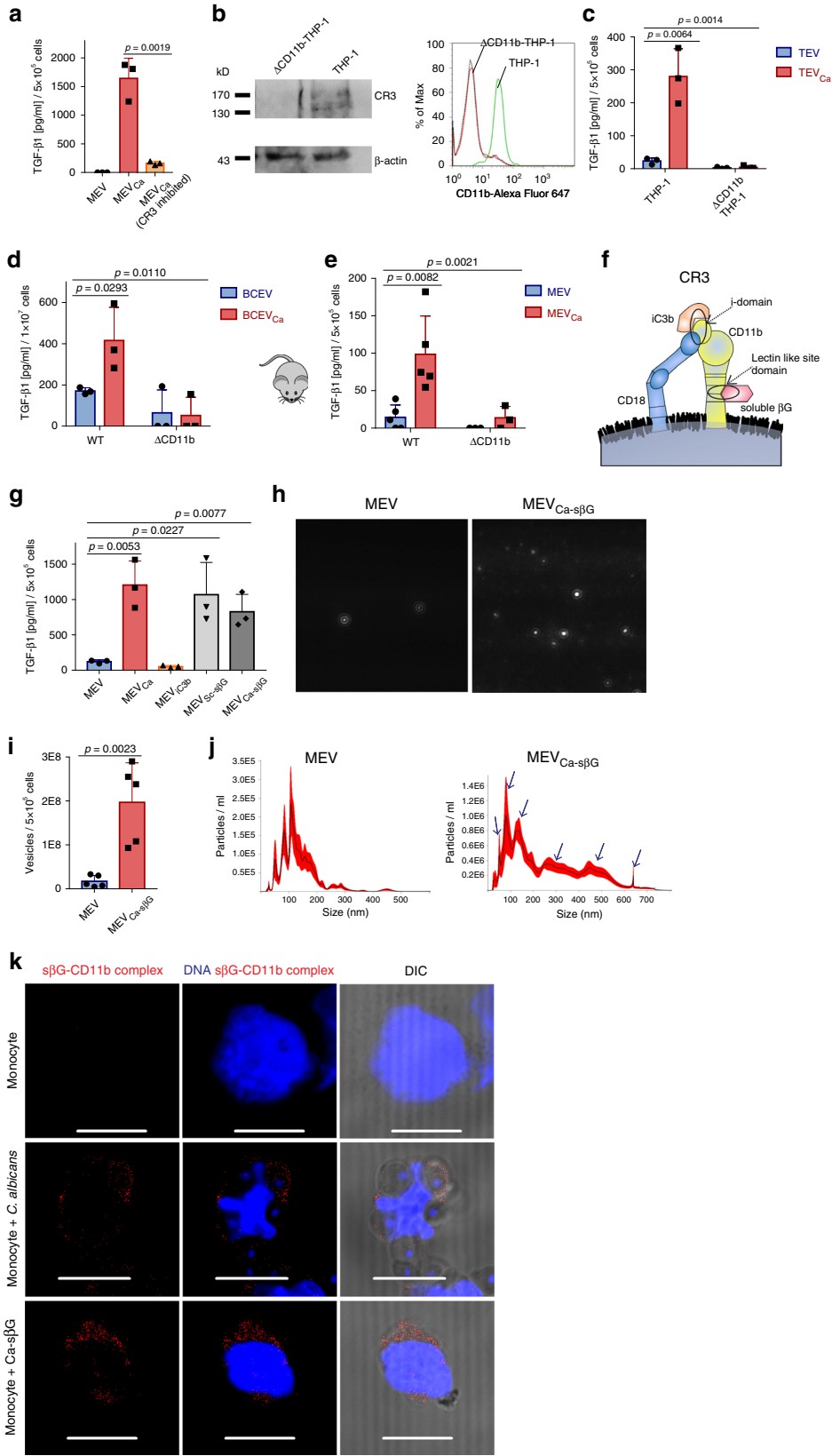

albicans-induced vesicles (Fig. 5b). KEGG pathway enrichment analysis also showed a high abundance of complement pathway proteins (Supplementary Fig. 4a). In addition, out of 100 commonly reported vesicle markers (ExoCarta)[19], 68 were upregulated in both type of vesicles (MEVs$_{Ca}$ and MEVs$_{Ca-s\beta G}$;

Fig. 4c). These data confirm that soluble β-glucan is a component of C. albicans that induces TGF-β1-transporting vesicles.

**Apoptotic bodies induce release of TGF-β1-transporting MEVs.** TGF-β1 has been described primarily as an anti-

**Fig. 4 Fungal soluble β-glucan interaction with CR3 induces MEVs$_{sCa}$ release. a** TGF-β1 is significantly upregulated in MEVs$_{Ca}$ compared with MEVs, but not when CR3 was blocked (data are presented as mean values $+/-$ SD, $p = 0.0019$, unpaired two-tailed $t$ test, $n = 3$ donors). MEVs were isolated from 1-h-infected monocytes ($5 \times 10^5$), and TGF-β1 was detected by ELISA. **b** CD11b knockout (KO) THP-1 cells generated via CRISPR-CAS9 lacks CD11b expression by both western blot and flow cytometry (see uncropped WB and gating strategy in Supplementary Fig. 7a, c). **c** TGF-β1 significantly increases in wild-type THP-1-derived TVs$_{Ca}$, but not EVs from CD11b KO THP-1 cells. EVs were isolated from uninfected or *C. albicans*-infected (1 h) wild-type or CD11b KO THP-1 cells (TVs or TVs$_{Ca}$, respectively) (each $5 \times 10^5$) (data are presented as mean values $+/-$ SD, $p = 0.0064$, unpaired two-tailed $t$ test, $p = 0.0014$, one-way ANOVA, $n = 3$ donors). **d** TGF-β1 significantly increases in ex vivo blood cell infection-derived vesicles (BCEVs$_{Ca}$) compared with control blood cell-derived vesicles (BCEVs) in wild-type mice (C57BL/6), but not CR3-deficient (CD11b KO) mice (B6.129S4-Itgam$^{tm1Myd}$/J). In all cases, vesicles were isolated from 1-h-infected blood cells ($1 \times 10^7$) (data are presented as mean values $+/-$ SD, $p = 0.0293$, unpaired two-tailed t test, p = 0.0110, one-way ANOVA, $n = 3$ donors). **e** MEVs$_{Ca}$ from wild-type but not from CR3-deficient mice show increased TGF-β1 concentration compared to MEVs (data are presented as mean values $+/-$ SD, $p = 0.0082$, unpaired two-tailed $t$ test, $p = 0.0021$, one-way ANOVA, $n = 5$ wild-type mice and $n = 3$ or CR3-deficient mice). Monocytes were generated from mouse bone marrow-derived stem cells. MEVs and MEVs$_{Ca}$ (1 h infection) were each isolated from $5 \times 10^5$ monocytes. **f** CR3 (CD11b/ CD18) harbors two binding domains: the I-domain (binds iC3b) and the lectin-like site (LLS) domain (binds soluble β-glucan (sβG)) (modified from O'Brien et al.[18]). **g** MEVs$_{Sc-sβG}$ and MEVs$_{Ca-sβG}$ (induced for 1 h by sβG from *S. cerevisiae* and *C. albicans*, respectively) but not by iC3b show increased TGF-β1 concentrations compared with MEVs. EVs were isolated from activated monocytes ($5 \times 10^5$) (data are presented as mean values $+/-$ SD, $p = 0.0053$, $p = 0.0227$, $p = 0.0077$, unpaired two-tailed $t$ test, $n = 3$ donors). **h** MEVs or MEVs$_{Ca-sβG}$ tracking by DLSM. Data are representative of four independent experiments. **i** MEVs$_{Ca-sβG}$ significantly increase compared with MEVs (data are presented as mean values $+/-$ SD, $p = 0.0023$, unpaired two-tailed $t$ test, $n = 5$ donors). **j** Size distributions of EVs determined with NanoSight NTA 3.2 software. EVs were isolated from $5 \times 10^5$ induced or control monocytes. Graphs were generated by overlaying the size distribution of MEVs and MEVs$_{Ca-sβG}$ derived from five donors. **k** sβG from *C. albicans* binds to CR3 on monocytes as seen by proximity ligation assay (PLA). PLA staining– red: sβG /CD11b complexes, blue: DNA. Bars: 10 µm. Representative data of $n = 5$ experiments (five donors).

inflammatory cytokine[20], and targeted disruption of the mouse *Tgfbr1* gene results in several inflammatory diseases[21]. To determine whether *C. albicans* exploits a physiological regulatory mechanism to dampen the immune response to the fungus, we aimed to identify a physiological condition where TGF-β1-transporting vesicles are released by monocytes in the absence of an infection. As apoptosis is characterized as a process of cell clearance without inflammation, we wondered whether human apoptotic bodies induce similar TGF-β1-transporting vesicles in monocytes. Apoptotic bodies were generated from human umbilical vein endothelial cells (HUVECs), isolated, opsonized in complement-active human serum, and incubated for 1 h with human blood monocytes. The vesicles induced in response to opsonized apoptotic bodies (MEVs$_{AB}$) were isolated and assayed for TGF-β1 by ELISA. Apoptotic bodies induced significantly higher amounts of TGF-β1-transporting vesicles (MEVs$_{AB}$) compared with MEVs ($p = 0.0008$, unpaired two-tailed $t$ test, $n = 3$) (Fig. 5d). This effect was also analyzed in the ex vivo blood system. Apoptotic body-induced blood cell vesicles (BCEVs$_{AB}$) carried about twice as much TGF-β1 compared with blood cell vesicles (BCEV) induced in the absence of apoptotic bodies (Fig. 5e). To investigate whether CR3 is involved in this process, apoptotic cells were incubated in whole blood in the presence of simvastatin, which blocks CR3 activation. Under these conditions, the amount of TGF-β1 in BCEVs$_{AB}$ was significantly reduced ($p = 0.0133$, unpaired two-tailed $t$ test, $n = 3$) and comparable with the levels in BCEVs derived in the absence of apoptotic bodies (Fig. 5e).

In addition, CR3 dependence was investigated in an ex vivo system using whole blood from wild-type and CD11b KO mice. Peripheral blood cells ($1 \times 10^7$) from mice of each genotype were incubated with opsonized apoptotic bodies generated from whole blood of respective mouse for 1 h, and vesicles were subsequently isolated and assayed for TGF-β1. No significant difference in TGF-β1 content was observed between BCEVs and BCEVs$_{AB}$ derived from CD11b KO mouse monocytes, but BCEVs$_{AB}$ generated from wild-type monocytes revealed a significant increase in TGF-β1 levels compared with BCEVs ($p = 0.0163$, unpaired two-tailed $t$ test, $n = 3$) (Fig. 5f). Thus, TGF-β1-containing vesicles are released by monocytes in a CR3-dependent manner in response to apoptotic bodies, and this pathway is also used by *C. albicans*.

**MEVs$_{Ca}$ downregulate IL-6 expression in human macrophages.** Detection of increased abundance of endocytosis proteins in MEVs$_{Ca}$ and MEVs$_{Ca-sβG}$ compared with MEVs (Supplementary Fig. 4b, c) suggested uptake by and interaction of these EVs with other cells. As discussed above, TGF-β1 acts predominantly as an anti-inflammatory cytokine. To assess the functional effects of TGF-β1-transporting vesicles, human blood monocytes were differentiated into macrophages and subsequently incubated with TGF-β1-transporting vesicles. To inhibit inflammation, TGF-β1 on the vesicles interacts with TGF-βRII on macrophages. Therefore, binding of TGF-β1-transporting vesicles to this receptor was evaluated by PLA. Co-incubation revealed complex formation between TGF-β1 on the vesicle and TGF-βRII on the macrophage, in contrast to MEVs, which did not show any interaction with TGF-βRII in the PLA (Fig. 5g).

As TGF-β1-transporting vesicles were expected to dampen the inflammatory response after binding to the TGF-βRII, macrophages were first incubated with lipopolysaccharide (LPS) to induce production of the inflammatory cytokine IL-6[22]. TGF-β1-transporting vesicles significantly reduced IL-6 production by macrophages ($p = 0.0339$, unpaired two-tailed $t$ test, $n = 3$), as measured by ELISA (Fig. 5h). When TGF-β1 on vesicles was blocked with TGF-β1-neutralizing antibodies, LPS-induced IL-6 was not reduced (Fig. 5h). This result confirmed the anti-inflammatory role of TGF-β1 vesicles.

**MEVs$_{Ca}$ reduce inflammation via the SMAD7 pathway.** To determine whether TGF-β1-transporting vesicles regulate IL-6 also in systemic *C. albicans* infection, human whole blood was infected with *C. albicans* for 4 h, and *IL-6* transcription was assayed in whole blood cells by qPCR. No changes of *IL-6* transcription was detected after infection and with TGF-β1-neutralizing antibodies (Fig. 6a). To further follow the function of TGF-β1-transporting vesicles in systemic *C. albicans* infection, interaction of these vesicles with human blood monocytes was assessed by PLA. Incubation of monocytes with TGF-β1-transporting vesicles for 15 min revealed the formation of complexes between TGF-β1 on vesicles and TGF-βRII on monocytes. Vesicles from uninfected monocytes did not show any fluorescent signal (Fig. 6b). This result demonstrates that TGF-β1-transporting vesicles generated in response to *C. albicans* bind to monocytes via TGF-βRII. Tracking these vesicles with SEM confirmed binding of multiple vesicles also to the *C. albicans* surface

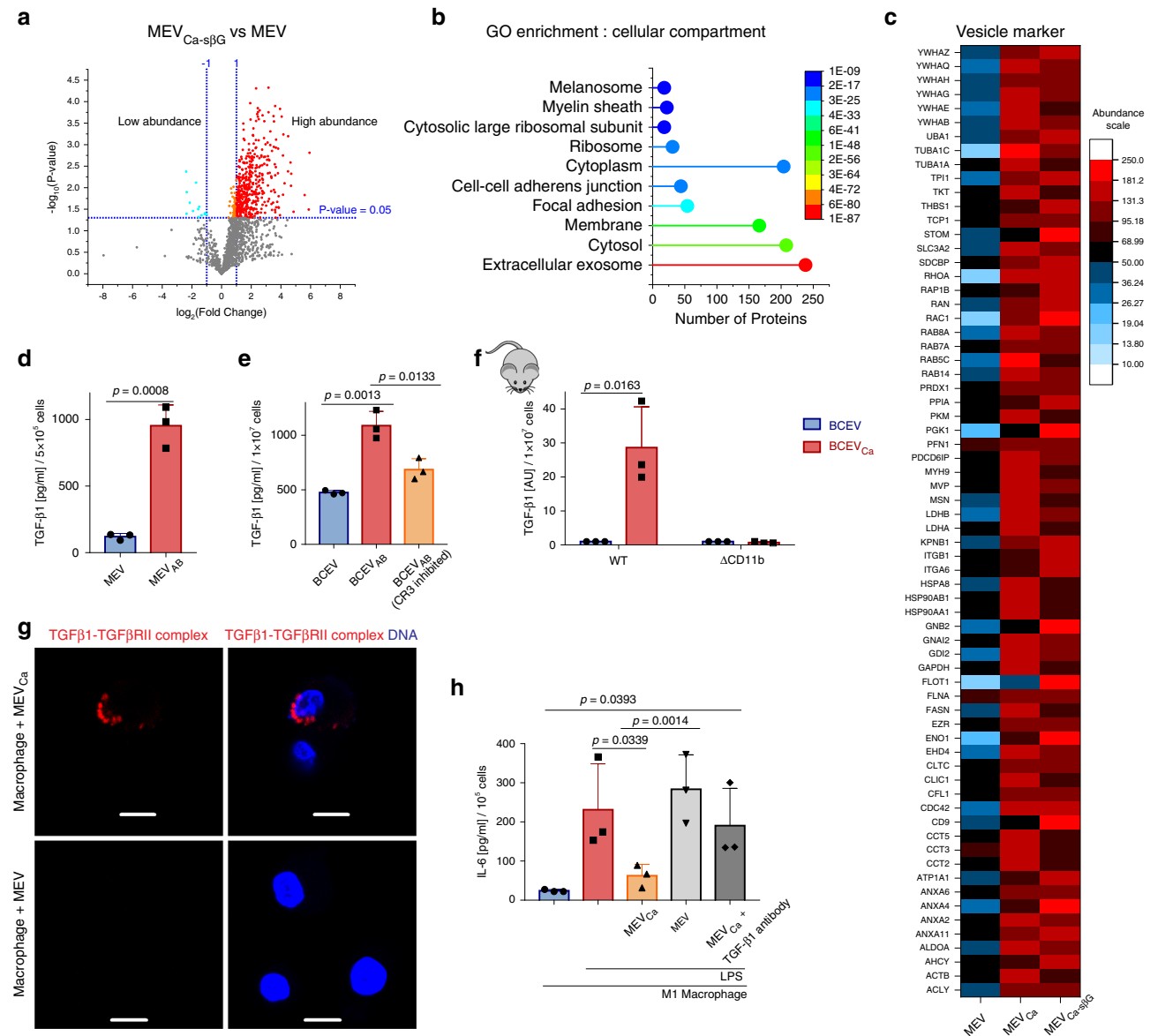

**Fig. 5 MEVs_Ca downregulate IL-6 expression in human macrophages. a** MEVs_Ca-sβG show significantly increased protein concentrations compared with MEVs. MEVs and MEVs_Ca-sβG were isolated from $1 \times 10^7$ monocytes, and proteins were detected using label-free LC-MS/MS-based proteomics ($p < 0.05$, unpaired two-tailed $t$ test, $n = 3$ different donors). **b** Most of the proteins in MEVs_Ca-sβG are extracellular exosome-related as determined by GO enrichment analysis for cellular compartments. Comparison was performed between protein contents of MEVs_Ca-sβG and MEVs from $n = 3$ donors. **c** MEVs_Ca and MEVs_Ca-sβG show upregulation of 68 vesicle marker proteins (ExoCarta) compared with MEVs. Protein contents of MEVs_Ca and MEVs_Ca-sβG were compared with that of MEVs from $n = 3$ donors. **d** TGF-β1 increases in MEVs_AB compared with MEVs. EVs were isolated after 1 h of co-incubated monocytes ($5 \times 10^5$) with opsonized apoptotic bodies (data are presented as mean values $+/-$ SD, $p = 0.0008$, unpaired two-tailed $t$ test, $n = 3$ donors). **e** TGF-β1 significantly increases in ex vivo blood cell-derived vesicles induced by apoptotic bodies (BCEVs_AB) compared with control blood cell-derived vesicles (BCEVs), but not when CR3 was blocked. Vesicles were isolated from 1 h co-incubated blood cells ($1 \times 10^7$) (data are presented as mean values $+/-$ SD, $p = 0.0013$, $p = 0.0133$, unpaired two-tailed $t$ test, $n = 3$ donors). **f** TGF-β1 significantly increases in ex vivo blood cell apoptotic body-induced vesicles (BCEVs_AB) compared to control blood cell-derived vesicles (BCEVs) in wild-type mice, but not in CD11b KO mice. In all cases, vesicles were isolated from 1 h induced blood cells ($1 \times 10^7$) (data are presented as mean values $+/-$ SD, $p = 0.0163$, unpaired two-tailed $t$ test, $n = 3$ donors). **g** TGF-β1 on MEVs_Ca but not MEVs binds to TGF-βRII on macrophages by proximity ligation assay (PLA). Cells were incubated with EVs for 30 min, and PLA assay was performed to detect TGF-β1/TGF-βRII complexes by CLSM. PLA staining– red: TGF-β1/TGF-βRII complexes, blue: DNA. Bars: 10 μm. A representative PLA of three experiments ($n = 3$ donors) is shown. **h** IL-6 is induced in LPS-stimulated macrophages in the presence of MEVs, but not MEVs_Ca. Blocking of TGF-β1 on MEVs_Ca did not reduce IL-6 production. M1 macrophages were differentiated from human blood monocytes, and IL-6 was measured by ELISA (data are presented as mean values $+/-$ SD, $p = 0.0339$, $p = 0.0014$, $p = 0.00393$, unpaired two-tailed $t$ test, $n = 3$ donors).

(Fig. 6c). When TGF-β1 was labeled with immunogold, TGF-β1-transporting vesicles were observed, particularly on *C. albicans* hyphae (Fig. 6d). To determine the effect of TGF-β1-containing vesicles in *C. albicans* infection, human whole blood was infected with *C. albicans* for 4 h, and then blood cells were lysed, and

proteins were separated by SDS-PAGE and immunoblotted for markers of TGF-β1 pathway activation, including phosphorylated SMAD2/3 and SMAD7. *C. albicans* infection resulted in strong induction of SMAD2/3 phosphorylation (Fig. 6e). However, when TGF-β1 was blocked during *C. albicans* whole blood infection with

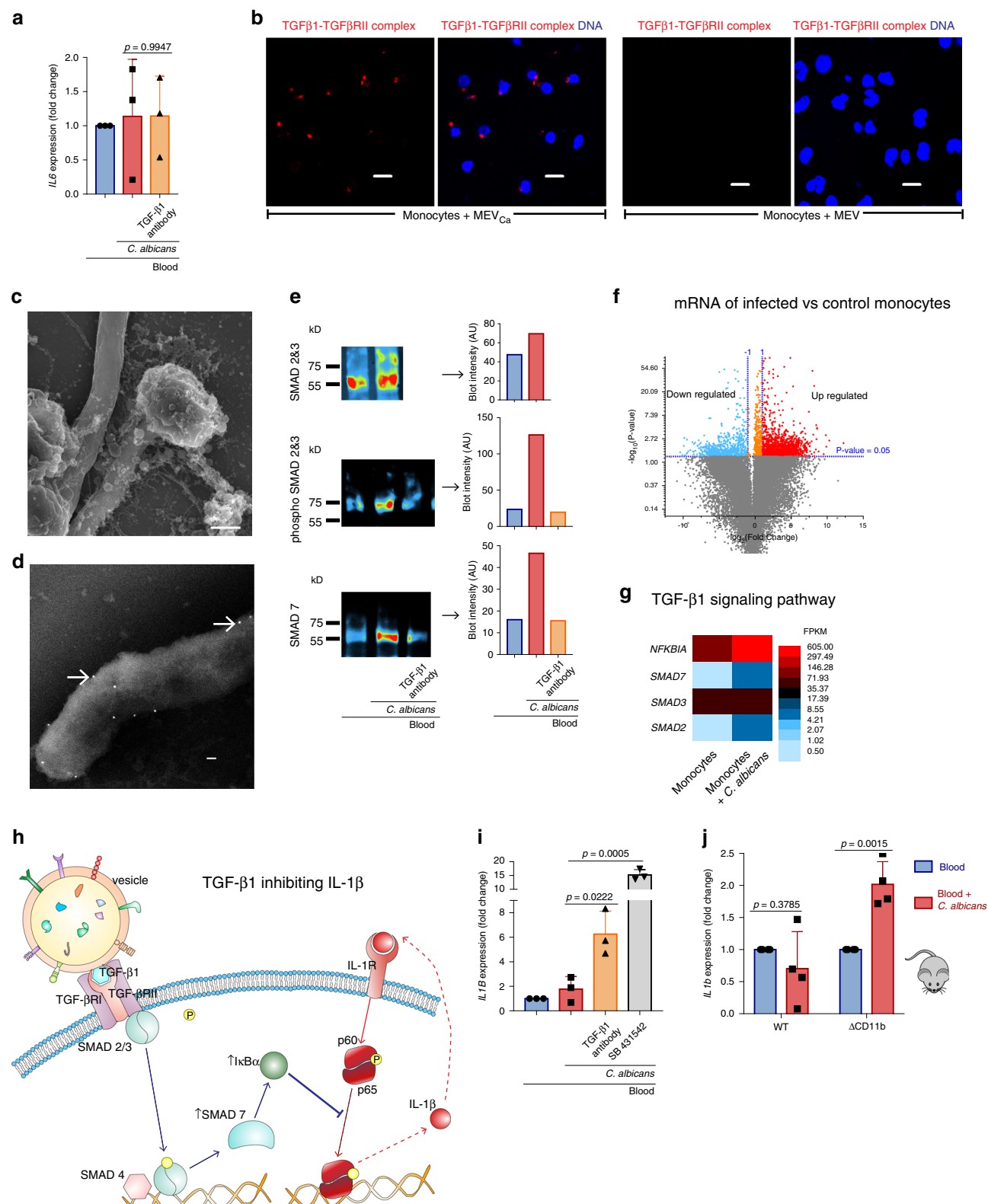

a neutralizing antibody, SMAD2/3 phosphorylation was inhibited (Fig. 6e). As SMAD2/3 phosphorylation leads to the upregulation of SMAD7, which subsequently signals through NF-κB[23], we also determined the expression of SMAD7 by western blot analysis. SMAD7 was substantially upregulated during *C. albicans* infection, and was reduced in the presence of an anti-TGF-β1 antibody (Fig. 6e).

RNA-seq of infected and control monocytes confirmed these results, as infected monocytes showed significant upregulation of *SMAD7* and *NFKBIA* as well as *SMAD2* (Fig. 6f, g). Induction of SMAD7 and IκBα is also reported to inhibit IL-1β and TNF-α in renal inflammation by blocking NF-κB-mediated transcription (Fig. 6h)[24]. A low amount of *IL1B* production was detected early during whole blood infection with *C. albicans* (4 h), and blocking

**Fig. 6 MEV$_{sCa}$ reduce inflammation via the SMAD7 pathway. a** No change in *IL6* transcription occurs in whole-blood ex vivo infection with *C. albicans*, also after blocking TGF-β1 (data are presented as mean values +/− SD, $p = 0.9947$, unpaired two-tailed *t* test, $n = 3$ donors). **b** TGF-β1 on MEVs$_{sCa}$ binds to TGF-βRII on monocytes. No such interaction was observed with MEVs. Cells were incubated with EVs for 30 min, and TGF-β1/TGF-βRII complexes detected by PLA using CLSM. PLA staining– red: TGF-β1/TGF-βRII complexes, blue: DNA. Bars: 10 μm. **c** MEVs$_{sCa}$ and **d** TGF-β1 adhere to *C. albicans* and its hyphae. Monocytes were infected with opsonized *C. albicans* for 1 h, and immunogold-labeled TGF-β1 visualized by SEM. **e** Phosphorylated SMAD2/3 and SMAD7 increase in whole blood infected with *C. albicans* ex vivo (1 h), but not when TGF-β1 was neutralized. Cells were lysed and intracellular proteins were for total and phosphorylated SMAD2/3 and SMAD7 detected by Western blot. (see uncropped WB in Supplementary Fig. 7d) The experiments in **b**, **c**, **d**, **e** are representatives of each $n = 3$ experiments. **f** Illumina-based RNA-seq reveals 6363 significantly upregulated genes in *C. albicans*-infected monocytes (1 h) compared with control monocytes. **g** RNA-seq analysis shows upregulation of *SMAD2*, *NFKBIA*, and *SMAD7*. For **f**, **g** RNA from four different donors were pooled before sequencing. **h** TGF-β1 inhibits IL-1β synthesis in renal fibrosis. TGF-β1 binds to TGF-βRII, and induces the phosphorylation of SMAD2/3. Phosphorylated SMAD2/3 in combination with SMAD4 upregulate SMAD7, which in turn upregulates IκBα. IκBα inhibits the IL-1β-positive feedback loop by inhibiting phosphorylation of NFκB[54]. **i** Little upregulation of *IL1B* expression is observed in whole blood infected with *C. albicans* ex vivo, but strong upregulation of *IL1B* occurs when TGF-β1 or TGF-βRI was blocked during ex vivo infection (data are presented as mean values +/− SD, $p = 0.0222$, $p = 0.0005$, unpaired two-tailed *t* test, $n = 3$ donors). **j** *IL1b* expression is not upregulated in whole blood from wild-type mice (C57BL/6) infected with *C. albicans*, but in blood from CR3-deficient (CD11b KO) mice (B6.129S4-Itgam$^{tm1Myd}$/J) (data are presented as mean values +/− SD, $p = 0.3785$, $p = 0.0015$, unpaired two-tailed *t* test, $n = 4$ donors). After 4 h whole blood infection, cells were lysed, and RNA was isolated and subjected to comparative qPCR.

of TGF-β1 significantly increased *IL1B* synthesis (Fig. 6i). Similarly, a large amount of *IL1B* was produced upon selective blocking of the TGF-β1 receptor-like kinase ALK5 and its relatives ALK4 and ALK7 with SB431542 (Fig. 6i). Also whole-blood infection with *C. albicans* (4 h) resulted in a minor increase of soluble TGF-β1 (Supplementary Fig. 5a). This demonstrates a reduction of the pro-inflammatory response in early infection.

The regulating effect of TGF-β1-transporting vesicles was subsequently confirmed in *C. albicans*-infected whole mouse blood. The release of TGF-β1-transporting vesicles was seen in wild-type mouse blood in early infection (Fig. 4d) with no significant upregulation of *IL1b* transcription (Fig. 6j). In contrast, no TGF-β1-transporting vesicles were released in infected CD11b KO blood cells (Fig. 4d), but high expression of *IL1b* transcription (Fig. 6j).

**MEVs$_{Ca}$ act anti-inflammatory on endothelial cells.** To verify the presence of TGF-β1-transporting vesicles in vivo, liver tissue sections from *C. albicans*-infected mice (24 h) were screened for TGF-β1 expression by CLSM. The sections were stained with an anti-TGF-β1 primary antibody and a fluorescently labeled secondary antibody together with the nucleic acid dye Sytox Orange. Strong TGF-β1 staining was identified along, and in endothelial cells of blood vessels of infected mice, but little or no staining was observed in tissues from uninfected mice (Fig. 7a). As blood vesicles easily come in contact with endothelial cells, these cells were subjected to further studies. First, BCEVs and BCEVs$_{Ca}$ were isolated by polymer precipitation. The total amount of BCEVs$_{Ca}$ was significantly higher (about 4 fold) compared with BCEV (Fig. 7c). BCEVs$_{Ca}$ size ranged from 40 to 400 nm (Fig. 7d). To confirm the presence of TGF-β1-transporting vesicles during whole-blood infection, the membrane and cytosol fractions of the vesicles were separated by lysis of the vesicles and subsequent centrifugation. Supernatants and membrane fractions were separated and immunoblotted using an anti-TGF-β1 antibody. BCEVs$_{Ca}$ but not BCEVs showed the presence of TGF-β1 in the membrane fraction (Fig. 7e). To understand the effect of TGF-β1-transporting vesicles on the blood vessel endothelium in systemic candidiasis, the interaction of the vesicles with human endothelial cells was assessed. First, the interaction of BCEVs$_{Ca}$ with HUVECs was assayed by PLA. After 30 min of incubation, multiple complexes between TGF-β1 on isolated vesicles and TGF-βRII on the HUVEC surface were detected. Vesicles from uninfected blood cells, however, did only generate low or no signals (Fig. 7f, g). Following incubation of vesicles with HUVECs for 6 h, the effect of TGF-β1-transporting vesicles on the HUVEC

phenotype was determined by qPCR. HUVECs treated with TGF-β1-transporting vesicles revealed an upregulation of several anti-inflammatory cytokines, including *TGFB1* and *IL4* (Fig. 7h; Supplementary Fig. 6). Increased production of TGF-β1 in endothelial cells was confirmed by immunofluorescence using LSM and measuring intracellular TGF-β1 by ELISA. HUVECs showed significantly more intracellular TGF-β1 when they were incubated with TGF-β1-containing blood cell vesicles from *C. albicans*-infected blood cells than when they were incubated with vesicles from uninfected blood (Fig. 7i–k). Treatment of the HUVECs with vesicles did not affect the cell viability (Supplementary Fig. 6). To confirm that this effect is mediated by TGF-β1 on vesicles, HUVECs were then incubated with recombinant TGF-β1, which also resulted in upregulation of *TGFB1* (Fig. 7l). To exclude an effect by the pathogen itself, HUVECs were incubated with *C. albicans* for 6 h. *C. albicans* alone failed to induce anti-inflammatory cytokines in HUVECs (Fig. 7m). These results suggest that TGF-β1-transporting vesicles are induced by *C. albicans* in blood cells, which subsequently bind to endothelial cells to amplify the TGF-β1 signal. Altogether these results demonstrate that TGF-β1-transporting vesicles are released in response to *C. albicans*, attach to endothelial cells, and upregulate the expression of TGF-β1 in vitro and in vivo.

## Discussion

EVs are central regulators of the immune response and are used as a form of cellular communication, particularly during bacterial or viral infections[2]. However, the generation, composition, and action of vesicles released by immune cells during in vivo fungal infections are so far not described. We show here that the human pathogenic fungus *C. albicans* mimics apoptotic cells in inducing vesicle release from human blood monocytes that transport TGF-β1 to other cells (Fig. 8). TGF-β1 significantly contributes to the development of immune tolerance on mucosal surfaces and supports *C. albicans* commensalism[25]. However, in early systemic infection, TGF-β1-transporting vesicles can reduce the host immune response against the fungus, which favors survival of the fungal cells.

About tenfold more vesicles were released per monocyte when the cells were incubated with *C. albicans* cells. Release of vesicles from monocytes was observed in real time within 20 min upon infection with *C. albicans* and thus constitutes an immediate innate immune response. A similar early vesicle production was reported from neutrophils exposed to *Staphylococcus aureus*[14]. The vesicles released in response to *C. albicans* showed a spherical-to-round morphology, as previously described for other

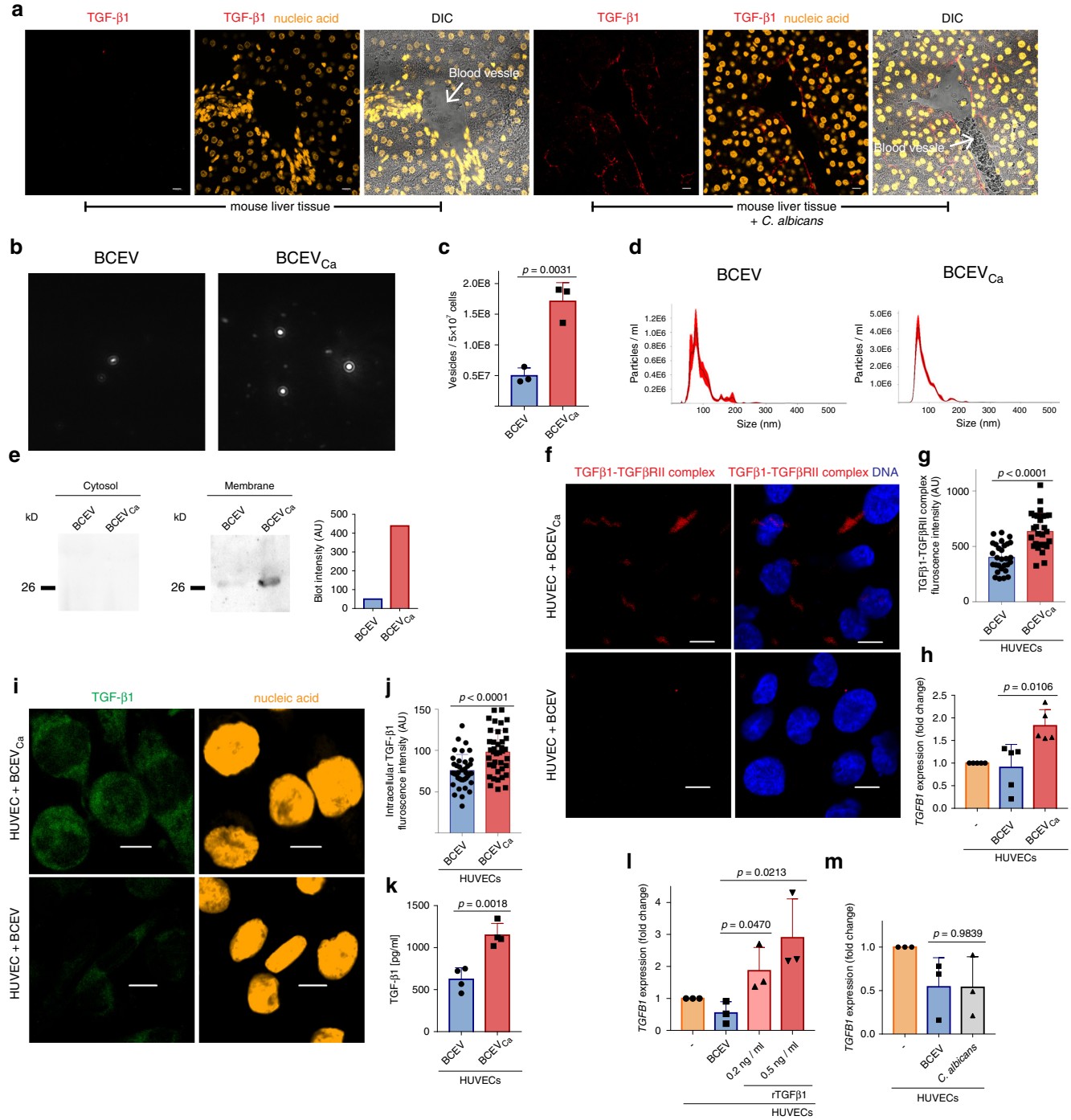

vesicles[26], and ranged from 80 to 500 nm in size, which is characteristic for exosomes and microvesicles. Intracellularly, the vesicles formed within monocytes stained positive for CD63 and tetraspanins, as previously reported for vesicles generated by monocytes in response to bacterial infection[27]. However, CD63 was lost on released EVs, and thus represents a less consistent marker for monocyte-derived vesicles[14,28]. By contrast, the vesicle marker CD9[28,29] was consistently detected on MEVs generated, both with and without *C. albicans* infection. *C. albicans*-induced vesicles were found to contain complement proteins (C3, factor H (FH)), which are often transported in vesicles[30]. Complement plays an important role in *C. albicans* infection, and the fungus has developed several evasion mechanisms[31]. Finally, like monocytes, the vesicles also expressed CD11b and CD14. EV

formation in response to *C. albicans*[32] and *Aspergillus fumigatus*[33] was recently also reported. However, experimental conditions were substantially different in these studies regarding cell type and incubation time.

The identification of heat shock proteins like HSP90 in *C. albicans*-induced vesicles suggested a mechanism of cytokine packaging into vesicles[34]. EVs often transport cytokines to protect these digestion-sensitive proteins in the blood[35]. As cytokines play a major role in the immune response to *C. albicans*[36], vesicles were screened for the presence of cytokines. Surprisingly, low amounts of the pro-inflammatory cytokines IL-1β and IL-6 were detected in *C. albicans*-induced vesicles from isolated monocytes or infected whole-blood cells, but high concentrations of the anti-inflammatory cytokine TGF-β1 were detected. So far,

**Fig. 7 MEV$_{sCa}$ act anti-inflammatory on endothelial cells. a** TGF-β1 vesicles align along the endothelial cells of blood vessels in liver tissue of *C. albicans*-infected mice as shown by immunohistochemistry as compared with uninfected mice. Liver tissue was collected 24 h after *C. albicans* infection of mice. Staining– red: TGF-β1 and orange: nucleic acids. Bars: 10 μm. Representative stainings of four experiments ($n = 4$ mice) are shown. **b** Tracking of purified blood cell EVs by DLSM. EVs were isolated from $5 \times 10^7$ uninfected or opsonized *C. albicans*-infected blood cells (BCEVs or BCEVs$_{Ca}$, respectively) by polymer precipitation. Representative profiles of three donors are shown. **c** BCEVs$_{Ca}$ significantly increase compared with BCEVs (data are presented as mean values $+/-$ SD, $p = 0.0031$, unpaired two-tailed $t$ test, $n = 3$ donors). **d** Profiles of BCEVs or BCEVs$_{Ca}$ as determined by DLSM using NanoSight NTA 3.2 software. Graphs were generated by overlaying the size distribution of BCEVs and BCEVs$_{Ca}$ from three donors. **e** TGF-β1 is present in the membrane fraction of BCEVs$_{Ca}$, but not in BCEVs and the cytosol fractions of BCEVs or BCEVs$_{Ca}$. BCEVs or BCEV$_{CaS}$ were isolated from $5 \times 10^7$ blood cells, cytosol and membrane fractions separated, and TGF-β1 detected by western blot (see uncropped WB in Supplementary Fig. 7e). **f** TGF-β1 on BCEVs$_{Ca}$ binds to TGF-βRII on HUVECs. HUVECs were incubated with BCEVs or BCEVs$_{Ca}$ for 30 min, stained for TGF-β1 and TGF-βRII, and TGF-β1/TGF-βRII complexes were detected by PLA. PLA staining– red: TGF-β1/TGF-βRII complexes, blue: DNA. Bars: 10 μm. Representative data of three experiments ($n = 3$ donors) are shown in **e**, **f**. **g** HUVECs incubated with BCEVs$_{Ca}$ show increased PLA fluorescence intensity compared with HUVECs incubated with BCEVs as measured by ZEN Black 2011 software (data are presented as mean values $+/-$ SD, $p < 0.0001$, unpaired two-tailed $t$ test, $n = 32$ individual cell from $n = 3$ donors). **h** HUVECs treated with BCEVs$_{Ca}$ but not with BCEVs upregulated *TGFB1* expression (data are presented as mean values $+/-$ SD, $p = 0.0106$, unpaired two-tailed $t$ test, $n = 5$ donors). HUVECs ($3 \times 10^6$) were treated for 6 h with BCEVs or BCEVs$_{Ca}$ each isolated from $5 \times 10^8$ blood cells. Purified RNA was evaluated by comparative qPCR. **i** HUVECs treated with BCEVs$_{Ca}$ for 6 h increase intracellular TGF-β1 as observed in CLSM. Staining– green: TGF-β1 and orange: nucleic acids. Bars: 10 μm. Representative data of three independent experiments ($n = 3$ donors) are shown. **j** Intracellular TGF-β1 as measured by ZEN 2011 (data are presented as mean values $+/-$ SD, $p < 0.0001$, unpaired two-tailed $t$ test, $n = 39$ individual cells from $n = 3$ donors). **k** BCEV$_{Ca}$-treated HUVECs increase intracellular TGF-β1 as measured by ELISA (data are presented as mean values $+/-$ SD, $p = 0.0018$, unpaired two-tailed $t$ test, $n = 4$ donors). **l** HUVECs treated with recombinant TGF-β1 increase *TGFB1* expression in a dose-dependent manner compared to treatment with BCEVs (data are presented as mean values $+/-$ SD, $p = 0.0470$, $p = 0.0213$, unpaired two-tailed $t$ test, $n = 3$ donors). **m** HUVECs incubated with *C. albicans* show no increase in *TGFB1* expression (data are presented as mean values $+/-$ SD, $p = 0.9839$, unpaired two-tailed $t$ test, $n = 3$ donors).

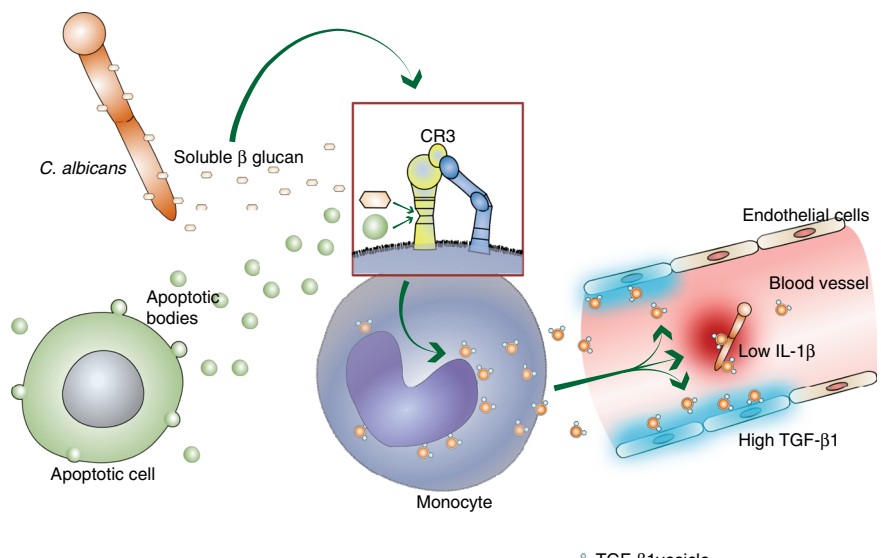

**Fig. 8 TGF-β1-transporting vesicle generation in human monocytes and modulation of the immune response.** The human pathogenic fungus *Candida albicans* as well as human apoptotic cells induce via the complement receptor 3 on monocytes the release of TGF-β1-transporting vesicles which act immune inhibitory and which induce synthesis of TGF-β1 in endothelial cells.

TGF-β1-containing vesicles have been observed in a parasitic infection model and were shown to be released by cancer cells[37,38], but the mechanism of their induction and their role in the immune response remain unclear. Generation of TGF-β1-transporting vesicles by *C. albicans* is shown here using an ex vivo whole-blood infection model, and in vivo in blood and liver tissue from *C. albicans*-infected mice.

Notably, the induction of TGF-β1-transporting vesicles from monocytes depends on the presence of CR3 on the monocytes. Previous studies indicated that CR3 is associated with TGF-β1 generation in dendritic cells[39,40], but did not report the significant link between CR3 and TGF-β1-transporting vesicle formation. In addition, opsonization with C3b, which binds to CR3, was previously considered to be important in bacteria-induced immune cell vesicle production[14], but was not reported for *C. albicans*. We identified soluble β-glucan of *C. albicans* as a ligand of CR3 and a

trigger for the release of TGF-β1-transporting vesicles. This is in agreement with previous studies showing that CD11b, which forms CR3 together with CD18[18,41], harbors a carbohydrate-binding site LLS that binds fungal-derived soluble, but not insoluble, β-glucan[18,41]. In addition, CD11b and CD18 together form the so-called I-domain, which then binds the complement protein iC3b. iC3b is generated from the cleavage of C3b deposited onto *C. albicans*[42,43]; however, in our study, iC3b did not induce any TGF-β1-transporting vesicles.

Apoptotic cells and apoptotic bodies have been reported to induce TGF-β1 in phagocytic cells[44–46]. We hypothesized that production of TGF-β1-transporting vesicles might be a conserved mechanism in human physiology used by *C. albicans* and evaluated vesicle induction from monocytes incubated with apoptotic cells. Indeed, incubation of monocytes with apoptotic cells resulted in the release of TGF-β1-transporting vesicles, as

observed with *C. albicans*. Such vesicle generation was also very fast (within minutes), and again the presence of CR3 was a prerequisite for vesicle release. As TGF-β1-transporting vesicles were generated by monocytes and whole-blood cells in response to apoptotic bodies, we concluded that *C. albicans* likely exploits this pathway for immune suppression.

TGF-β1 is a pleiotropic cytokine produced by many cell types, including immune and nonimmune cells, and it regulates multiple cellular functions. TGF-β1 primarily inhibits the function of inflammatory immune cells[47]. Human macrophages induce CD4+ Foxp3+-regulatory T cells via binding and release of TGF-β1[48]. High concentrations of TGF- β1 promote this Foxp3 induction and consequently formation of immune suppressive iTreg cells[49]. Autocrine activity and TGF-β1 release is generally believed to function as an important feedback system for limiting the extent and duration of an inflammatory response. Accordingly, targeted disruption of the mouse *Tgfb1* gene results in multifocal inflammatory disease[21]. Functionally, we found that TGF-β1-transporting vesicles downregulate LPS-induced IL-6 production by macrophages. The pro-inflammatory cytokine IL-6 plays vital roles in the inflammatory process during infection, and is involved in the pathogenesis of several immune-related diseases[50]. TGF-β1 downregulates IL-6 in monocytes[51], but can also induce IL-6 release from smooth muscle cells[52] and other tissues.

TGF-β1 is clearly exposed on the outer surface of the vesicle, and binding of TGF-β1-transporting vesicles to monocytes via TGF-βRII was demonstrated in whole blood, and resulted in the phosphorylation of SMAD2/3 and an upregulation of SAMD7. Vesicle-transported TGF-β1 is inactive, and is activated intracellularly as reported in a very recent study, which showed TGF-β1 to trigger SMAD-dependent pathways in human mesenchymal stem cells[53]. In renal inflammation, SMAD7 inhibits transcription mediated by phosphorylated NF-κB by upregulating IκBα, which in turn inhibits IL-1β- and TNF-α-driven positive feedback loops[23,24,54]. The inhibitory role of TGF-β1 in T-cell differentiation is well documented[25], and it has also been reported that TGF-β1 downregulates the pro-inflammatory cytokine IL-1β under both physiologic and pathologic conditions[24,54,55]. Lack of such IL-1β-mediated pro-inflammatory responses to infectious microbes like *C. albicans* favors the survival of the pathogen[56], with potentially fatal consequences for the host[57].

As TGF-β1 is exposed on vesicles, it can bind to TGF-βRII expressed on target cells. Endothelial cells that express TGF-βRII on their surface[58] bind *C. albicans*-induced TGF-β1-transporting vesicles in whole blood and upregulate the transcription of *TGFB1* representing a positive feedback loop, as previously described[45,59,60]. Within 6 h of co-incubation, the TGF-β1-transporting vesicles induce TGF-β1 synthesis in endothelial cells and also increase their production of other anti-inflammatory cytokines. This was confirmed with recombinant TGF-β1, excluding a direct effect by *C. albicans* alone. In agreement with previous reports demonstrating the upregulation of TGF-β1 upon *C. albicans* infection[61] and the presence of TGF-β1 in hepatocytes of neutropenic rabbits infected with *C. albicans*[61], we show that TGF-β1 is expressed in the livers of *C. albicans*-infected mice and is localized to the endothelial cells.

In summary, we describe an early-acting, previously undescribed immune response by human monocytes to a fungal infection with *C. albicans*. Due to the binding of soluble, *C. albicans*-derived β-glucan to CR3 on host monocytes, the monocytes form and release EVs transporting TGF-β1. Similar vesicles are released by monocytes in response to apoptotic cells binding to CR3. TGF-β1-transporting vesicles downregulate inflammation in whole-blood cells, and amplify the anti-inflammatory response in endothelial cells. Thus, *C. albicans* can reduce the very early induction of pro-inflammatory

cytokines in the host[56,62] by inducing the production of TGF-β1 vesicle. We hypothesize that this early suppression of a pro-inflammatory response occurs in the gut in response to the microbiome, but that the suppression is detrimental for the host in systemic infection[57]. As candidiasis has a very high mortality rate[63,64], the current study may provide therapeutic interventions for *C. albicans* infection, and also suggests to test the use of soluble β-glucan for immune suppression in inflammatory diseases.

## Methods

**Human cells.** Human monocytes were isolated from sterile buffy coats (JENA University Hospital, Germany) or from fresh blood collected in Na-heparin tubes (BD Biosciences) from healthy volunteers after informed consent. Peripheral blood mononuclear cells (PBMCs) were isolated from blood or buffy coats using Biocoll (Density 1.077 g/ml) (Biochrom) gradient centrifugation. Contaminant platelets were removed by low speed centrifugation of 160×*g*. The lymphocyte population from PBMCs was decreased by using 46% Percoll (density 1.135 g/ml) (GE healthcare) gradient centrifugation. Monocytes were then isolated by negative selection procedure using the pan monocyte isolation kit, (Miltenyi Biotec, cat no. 130-096-537) according to the manufacturer's protocol. Purity of the monocytes was assessed by detecting the presence of CD14 surface marker using flow cytometer.

Human monocytic THP-1 cell line (ACC 16, DSMZ) was maintained in $M_{growth}$ medium (RPMI 1640 medium (Lonza) supplemented with 10% FCS (Thermofischer), 2 mM ultra glutamine (Lonza), 25 µg/ml of gentamicin sulphate (Lonza)) at 37 °C, 95% humidity, and 5% $CO_2$. Human M1 macrophages were generated from lymphocyte-depleted PBMCs. Monocytes from lymphocyte-depleted PBMCs were allowed to adhere to the cell culture flask T75 in $M_{growth}$ medium for 2 h. M1 macrophages were obtained by differentiation of adherent monocytes in $M_{growth}$ medium supplemented with 50 ng/ml of recombinant human GM-CSF (Peprotech) for 7 days at 37 °C, 95% humidity, and 5% $CO_2$. During differentiation procedure, fresh $M_{growth}$ medium supplemented with GM-CSF was introduced after 4 days. After 7 days, macrophages were harvested using accutase (Capricorn Scientific).

Human fresh blood was collected in Na-Heparin tubes from healthy volunteer by venipuncture. Whole blood cells were isolated by centrifugation at 2000×*g*, 15 min, 4 °C. Human umbilical vein endothelial cells (HUVECs) were maintained in $H_{growth}$ medium (DMEM medium (Lonza) supplemented with 10% FCS (Thermofischer), 2 mM ultra glutamine (Lonza), 25 µg/ml of gentamicin sulphate (Lonza)) at 37 °C, 95% humidity, and 5% $CO_2$.

**Mouse cells.** Experiments were conducted in compliance with European and German regulations. Protocols were approved by the responsible Federal State authority and ethics committee (Zuchtrahmenantrag 02-05/16 and Tötunganzeige FSU-02-2017). C57BL/6 mice and B6.129S4-Itgam$^{tm1Myd/J}$ mice (CD11b knocked out) (Jackson Laboratory) were housed in ventilated cages with free access to water and food. C57BL/6 mice were maintained with heterozygous breeding, and B6.129S4-Itgam$^{tm1Myd/J}$ mice were maintained with KO breeding. Male and female mice with the age between 9 and 22 weeks were used for the experiments. Mice were euthanized with $CO_2$. Mouse fresh blood was collected from euthanized mouse immediately by cardiac puncture in hirudin monovette (Sarstedt). Whole-blood cells were isolated by centrifugation at 2000×*g*, 15 min, 4 °C. Serum from the supernatant was used for *C. albicans* opsonization. Femur and tibia bones were isolated from euthanized mouse. Bones were sterilized with ethanol, bone orifice was cut open, and flashed with $M_{growth}$ using a 27-g needle (B. Braun) to harvest the cells of marrow. Collected bone marrow suspension was strained through a 70-µm nylon web. Cells were transferred to ultra-low attachment T25 cell culture flasks (Corning). Bone marrow stem cell differentiation was carried out in $M_{growth}$ supplemented with 30 ng/ml of mouse recombinant M-CSF (BioLegend) at 37 °C, 95% humidity, and 5% $CO_2$. After 5 days of differentiation, mouse bone marrow-derived monocytes were harvested from the supernatant. Purity of the monocytes were analyzed by detecting the presence of CD115 using flow cytometer (Supplementary Figs. 3 and 7) using Alexa Fluor 488 anti-mouse CD115 antibody (BioLegend, cat no. 135512) (1:200). Presence of CD11b was also detected on monocytes using flow cytometer using Alexa Fluor 647 anti-mouse CD11b antibody (BioLegend, cat no. 101218) (1:200). Presence of the *ITGAM* gene was analyzed in all mice. Isolated genomic DNA from C57BL/6 or B6.129S4-Itgam$^{tm1Myd/J}$ was subjected to PCR reaction using common forward primer, reverse primer for C57BL/6, and reverse primer for B6.129S4-Itgam$^{tm1Myd/J}$ (Supplementary Table 1). This yields 264 and 166-bp long PCR products, respectively (Supplementary Fig. 3a). PCR products were resolved on 2% agarose gel visualized with ethidium bromide stain (uncropped gel picture in Supplementary Fig. 7f).

**Microbial strain and culture.** *C. albicans* wild-type (SC5314)[65] was maintained on yeast extract–peptone–dextrose (YPD) agar (2% D-glucose, 1% peptone, 5% yeast

extract in agar) for no more than 1 month. After *C. albicans* was grown overnight in YPD medium at 30 °C with shaking, fungal cells were reseeded in fresh YPD medium, and grown into the log phase by incubation at 30 °C for 4 h prior the experiments. *C. albicans* from the log phase was opsonized by incubation in 10% NHS at 37 °C for 30 min.

**Soluble β-glucan extraction**. Soluble β-glucan was enriched from *C. albicans* yeast cell according to refs. [16,17]. *C. albicans* was cultivated in carbon-limiting medium originally described by Shepard and Sullivan[66]. Recipe for 1 liter of C-limiting medium included sucrose 10 g, $(NH4)_2SO_4$ 2 g, $KH_2PO_4$ 2 g, $CaCl_2·2H_2O$ 0.05 g, $MgSO_4·7H_2O$ 0.05 g, $ZnSO_4·7H_2O$ 1 mg, $CuSO_4·5H_2O$ 1 mg, $FeSO_4·7H_2O$ 0.01 g, biotin 25 μg final pH 5.2. *C. albicans* was grown overnight at 28 °C in C-limiting medium. From the overnight culture, *C. albicans* cells ($1 \times 10^8$) were cultured in 200 mL of C-limiting medium for 24 h at 28 °C. *C. albicans* was collected from the culture, and washed in ethanol and acetone and subsequently dried. 6 g of dried yeast cells were resuspended in 400 mL 0.1 M NaOH and oxidized using 50 mL NaClO solution for 1 day at 4 °C. After reaction was completed, the insoluble fraction was collected using centrifugation. Insoluble fraction was washed first with distilled water followed by washing with ethanol and acetone, and subsequently dried. Dried fraction was then dissolved in DMSO with boiling and sonication. Insoluble fraction was removed, and soluble fraction was precipitated from DMSO in four parts ethanol to one part DMSO. Precipitated soluble β-glucan was washed with ethanol and dried. Dried soluble β-glucan was dissolved in 5 mL of DPBS, sterile filtered, aliquoted, and stored at −80 °C for further use.

**Apoptotic body generation**. Apoptotic bodies were generated from HUVEC cell line and whole-blood cells using UV irradiation[67]. After harvesting, HUVEC cells and whole-blood cells were washed with and suspended in DPBS. Cells were then transferred into six-wells plates and exposed to UV light for 4 h for whole-blood cells and 6 h for HUVEC cells. Dead cells were removed by centrifugation at 300×*g* for 10 min, and apoptotic bodies were harvested by centrifugation at 3000×*g* for 20 mins. Apoptotic bodies were opsonization by incubation in 10% NHS at 37 °C for 30 min.

**ΔCR3 THP-1 cell**. CRISPR/Cas9 method was used to knockout the *CD11b* from THP-1 cell line and generate ΔCR3 THP-1 cells. Adjacent to protospacer adjacent motif (PAM), 20 base-guide oligo sequences were designed to target the *ITGAM* gene. The guide oligo sequence was designed to be inserted upstream the single-guide RNA scaffold into the plasmid pSpCas9(BB)-2A-GFP (PX458) (which was a gift from Feng Zhang (Addgene plasmid # 48138; http://n2t.net/addgene:48138; RRID:Addgene_48138)) into the BbsI insertion site under the guide of the U6 promoter using the golden gate assembly protocol[68]. The guide oligos were as follows (Supplementary Table 1). Guide oligos were annealed and phosphorylated in a thermal cycler with 100 μM of each sense- and anti- guide oligo, 1 μL of 10× T4 ligation buffer, 1 μL T4-polynukleotidkinase. Annealing and phosphorylation was performed at 37 °C for 30 min, 95 °C for 5 min, and ramped down to 25 °C at 5 °C min$^{-1}$. For the insertion of annealed guide oligos, 1 μg of pSpCas9 (BB)-2A-GFP was digested with 1 U BbsI (New England Biolabs) at 37 °C for 30 min. Digested plasmid was separated by agarose gel electrophoresis, and the insert purified using QIAquick gel extraction kit (Qiagen) according to the manufacturer's protocol. In all, 1 μL of (1:200) annealed guide oligo was ligated to 50 ng of digested pSpCAS9(BB)-2A-GFP using 1 U quick ligase enzyme (New England Biolabs) at room temperature for 10 min. Guide oligo-pSpCas9(BB)-2A-GFP was then transformed into *E. coli* DH5α by heat shock. Successful transformants were selected with ampicillin on LB agar plates. Transformant colonies were again cultured under selection pressure, and guide oligo-pSpCas9(BB)-2A-GFP was isolated from *E. coli*. Guide oligo-pSpCas9(BB)-2A-GFP was then transfected into THP-1 cells by Amaxa® Human Monocyte Nucleofector® Kit according to[69]. Transfected cells were maintained in similar conditions as THP-1 cells, and after 3 days, cells were stained for CD11b with mouse anti-human CD11b (BioLegend, cat no. 393102) (1:200) and secondary goat anti-Mouse Alexa Fluor 647 (Thermo Fisher, cat no. A-21235) (1:1000). Stained cells were then sorted for CD11b-negative and GFP-positive signals using BD FACSAria Fusion cell sorter. Sorted cells were taken into the culture, and further tested for CD11b using flow cytometry and western blotting.

**SEM**. Human monocytes were infected with opsonized *C. albicans* (MOI 1:1)) for 1 h in $M_{induction}$ medium (RPMI 1640 medium supplemented 2 mM ultra glutamine) on poly-L-lysine-coated 12-mm coverslips. After co-incubation, medium was aspirated, and cells were fixed with 0.01% glutaraldehyde and 4% formaldehyde in DPBS. Aldehydes were removed by washing with 100 mM glycine in DPBS, and cells were blocked with 1% BSA, 0.5% fish gelatin, 0.005% Tween 20. Fixed cells were stained with 5 μg/mL mouse anti-TGF-β1 antibody (R&D Systems, cat no. MAB240), and secondary goat anti-mouse IgG (H + L) antibody conjugated with 20-nm gold (BBI Solutions, cat no. EM.GMHL20) (1:25) diluted in blocking solution. Samples were fixed for 1 h in 2.5% glutaraldehyde in sodium cacodylate buffer (0.1 M, pH 7.0). The samples were washed with cacodylate buffer and postfixed for 1 h with 1% osmium tetroxide in cacodylate buffer. Samples were dehydrated in rising ethanol concentrations followed by critical point drying in a

Leica EM CPD300 Automated Critical Point Dryer (Leica) and finally coated with carbon (10 nm) in a BAL-TEC MED 020 Sputter Coating System (BAL-TEC). SEM images were acquired at different magnifications in a Zeiss-LEO 1530 Gemini field-emission scanning electron microscope (Carl Zeiss) at 8-kV acceleration voltage and a working distance of 7 mm using an intense secondary electron detector for secondary electron imaging and a scintillation type backscatter electron detector (Centaurus Detector, K.E. Developments) for antibody-gold detection.

**In vitro and ex vivo vesicle generation and isolation**. Vesicles were generated from human monocytes, mouse bone marrow-derived monocytes, or THP-1 cells or M1 macrophages or whole-blood cells (human/mouse) after 1 h of induction or from untreated control cells in $M_{induction}$ medium at 37 °C, 95% humidity, and 5% $CO_2$. Cells were incubated with opsonized *C. albicans* (MOI 1:1) to isolate infection-derived vesicles. For measuring CR3 dependency, cells were treated with 30 μM simvastatin (Sigma-Aldrich) prior to incubation with *C. albicans*. In absence of *C. albicans*, cells were induced with 50 μg/mL of iC3b (CompTech), or 10 μg/10$^6$ cells of *S. cerevisiae*-derived sβG (InVivoGen), or 10 μl/10$^6$ cells of *C. albicans*-derived sβG. In case of apoptotic body-mediated induction, cells were incubated with opsonized apoptotic bodies which were isolated from double the number (MOI 1:2) of induced cells. For measuring CR3 dependency, cells were treated with 30 μM simvastatin (Sigma-Aldrich) prior to incubation with apoptotic bodies. Cells and *C. albicans* were removed by centrifugation at 3000×*g* for 15 min at 4 °C from the supernatant and accumulated EVs collected isolated using ExoQuick-TC (System Biosciences) according to the manufacturer's protocol.

Size distribution of TGF-β1-transporting vesicles was performed by isolating these vesicles from the total vesicle population from whole-blood cells ($1 \times 10^9$) treated with *C. albicans* using M-pluriBead (pluriSelect) according to the manufacturer's protocol. TGF-β1-transporting vesicles were isolated by positive selection with mouse anti-TGF-β1 antibody (R&D Systems, cat no. MAB240). Vesicle detection and size distribution of CD9 and TGF-β1-transporting vesicles was performed by precipitation of TGF-β1-transporting vesicles. From this population, CD9-positive vesicles were isolated using CD9 M-pluriBead (pluriSelect, cat no. 19-00900-20). Therefore, the final vesicles obtained were both CD9 and TGF-β1-transporting vesicles. After removing cell debris, cells, and *C. albicans*, MEVs were in parallel also isolated by ultracentrifugation at 120,000×*g* for 1 h at 4 °C.

Vesicles were also isolated from 1 ml of human blood treated with *C. albicans* ($1 \times 10^9$) using size-exclusion chromatography with sepharose CL-2B (Sigma-Aldrich) according to Boing et al.[70]. In total, 14 different 1-ml fractions were collected from chromatography, and each subjected to vesicle detection. The Isolated EVs were stored at −20 °C for later use. For subsequent uses, EVs were slowly thawed on ice.

Recombinant TGF-β1 (R&D Systems) (2000 pg/ml) was dissolved in $M_{induction}$ medium. TGF-β1 precipitation was performed using the same ExoQuick-TC. Precipitate and supernatant were collected after ExoQuick-TC treatment. Similarly, precipitation experiment was also performed with recombinant latent TGF-β1 (R&D Systems).

**ELISA**. Cytokines were measured in EVs isolated from $5 \times 10^5$ human monocytes, $5 \times 10^5$ human M1 macrophages, $1 \times 10^6$ mouse monocytes, or $1 \times 10^7$ human/ mouse whole-blood cells. TGF-β1 was measured with Human TGF-β1 DuoSet ELISA (R&D Systems, cat no. DY240) and mouse TGF-β1 DuoSet ELISA (R&D Systems, cat no. DY1679) or Human/Mouse TGF-β1 ELISA Ready-SET-Go! Kit (eBioscience, cat no. 88-8350-88). IL-10, IL-6, and Il-1β were measured using the human IL-10 ELISA Ready-SET-Go! Kit (eBioscience, cat no. 88-7106-22), human IL-6 ELISA Ready-SET-Go! Kit (eBioscience, cat no. 88-7066-22), or human IL-1β ELISA Ready-SET-Go! Kit (eBioscience, cat no. 88-7261-86). In all cases, the ELISA was performed according to the manufacturer's protocol. CD9 and HSP90 were measured with ELISA on immobilized vesicle using anti-CD9 antibody (Novus Biologicals, cat no. NB500-327) (2 μg/mL) and anti-HSP90 antibody (Abcam, cat no ab79848) (2 μg/mL), respectively. Vesicles in different fractions of size-exclusion chromatography was immobilized with TGF-β1 capture antibody from Human TGF-β1 DuoSet ELISA, and CD9 or HSP90 was detected on vesicles by biotinylated CD9 and HSP90 antibody and streptavidin-conjugated HRP.

**Vesicle counting**. Isolated vesicles were counted using NS300 dynamic light-scattering microscope (Malvern) fitted with NanoSight NTA 3.2 software. Vesicles isolated from $5 \times 10^5$ monocytes or $5 \times 10^7$ whole-blood cells were dispersed in 1 mL of DPBS and injected though the microscope at 100 (AU) pump flow rate. Videos were capture at 24 fps, three times 60 s for each sample, and analyzed using NanoSight NTA 3.2. Fractions of 1 mL obtained from size-exclusion chromatography were similarly screened.

**Cryo-TEM**. EVs isolated from $4 \times 10^4$ infected monocytes (MOI 1:1) were collected in 4-μL suspension, and were applied onto copper EM-grids covered by a QUANTIFOIL Multi A holey carbon film (Quantifoil Micro Tools), and excess of liquid was blotted automatically between two strips of filter paper. Subsequently, the samples were rapidly plunge-frozen in liquid ethane (cooled by liquid nitrogen at about −180 °C) in a cryobox (Carl Zeiss). Excess of ethane was removed with a

piece of filter paper. The samples were transferred immediately using a Gatan 626 cryo-transfer holder (Gatan) into a pre-cooled cryo-electron microscope (Philips) operated at 120 kV and imaged under low dose conditions. The images were recorded with a 2k F216-CMOS-camera (CMOS-camera and acquisition software, EMMANU4 v 4.00.9.17, TVIPS). In order to minimize the noise, four images were recorded and averaged to one image.

**Freeze-fracture TEM.** Vesicles isolated from $5 \times 10^4$ infected monocytes (MOI 1:1) were collected in 5-µL suspension and enclosed between two 0.1-mm-thick copper profiles as used for the freeze-fracture sandwich double-replica technique. The sandwiches were physically fixed by rapid plunge-freezing in a liquid ethane/propane mixture, cooled by liquid nitrogen. Freeze-fracturing was performed at −150 °C in a BAF400T freeze-fracture unit (BAL-TEC) using a double-replica stage. The fractured samples were shadowed with 2 nm Pt/C (platinum/carbon) at an angle of 35 °C, followed by perpendicular evaporation of a 15–20-nm-thick carbon layer. The evaporation of Pt/C was controlled by a thin-layer quartz crystal monitor; the thickness of the carbon layer was controlled optically. The obtained freeze-fracture replica was transferred to a cleaning solution (commercial sodium hypochlorite, containing 12% active $Cl_2$) for 30 min at 45 °C. Then, the replica was washed four times in distilled water and transferred onto unfiled copper EM-grids for examination in a Zeiss EM902A electron microscope (Carl Zeiss) operated at 80 kV. Images were recorded with a 1 k (1024 × 1024) FastScan-CCD-camera (CCD-camera and acquisition software EMMANU4 v 4.00.9.17, TVIPS).

**Label-free LC–MS/MS.** EVs were isolated from $1 \times 10^7$ human monocytes——control (MEVs), *C. albicans*-infected (MEVs$_{Ca}$), *C. albicans*-derived sβG-induced (MEVs$_{Ca-sβG}$), and collected in a final volume of 100 µL of DPBS and stored frozen until use. Frozen EVs were thawed with 400 µL chilled MeOH. Forced precipitation and phase separation was performed with 100 µL chilled chloroform 300 µL pure ice-cold water, followed by chilling and centrifugation at 18,000×*g*. The aqueous layer was discarded, and 400 µL of MeOH was added followed by centrifugation at 18,000×*g*. After centrifugation, MeOH was removed, and samples were almost dried. Samples were resolubilized in 100 µL denaturation buffer (50 mM triethylammonium bicarbonate (TEAB) in 50/50 trifluoroethanol (TFE)/H2O (v/v)) by pipetting and ultrasonication. After solubilization, samples were incubated at 55 °C for 1 h with 2 µL of reduction buffer (500 mM TCEP (tris (2-carboxyethyl)phosphine) in 100 mM TEAB) followed by incubation at RT for 30 min with 2 µL alkylation buffer (625 mM CAA (2-chloroacetamide) in 100 mM TEAB). After alkylation, the precipitation, phase separation, and drying steps were repeated as mentioned before. Samples were then resolubilized in 100 µl 100 mM TEAB by sonication. C-terminal of lysine of sample proteins was cleaved with 2 µL of rLys C (1 µg/µL) (Promega) by incubating for 2 h at 37 °C. Samples are then subjected to trypsinization using 2 µl of Trypsin Gold protease (2 µg/µl) (Promega) by mixing and incubating for 16 h at 37 °C. the reaction was stopped with 10 µl of 10% HCOOH. Samples were dried and solubilized in 0.05% trifluoroacetic acid and 2% acetonitrile in water by pipetting, vortexing, and ultrasonication. Samples were filtered through 0.2-µm spin filter, transferred to HPLC vials, and stored at −80 °C until LC–MS/MS analysis.

LC–MS/MS analysis was performed on an Ultimate 3000 nano RSLC system connected to a Qexactive Plus mass spectrometer (both Thermo Fisher Scientific, Waltham, MA, USA). Peptide trapping for 5 min on an Acclaim Pep Map 100 column (2 cm × 75 µm, 3 µm) at 5 µl/min was followed by separation on an analytical Acclaim Pep Map RSLC nano column (50 cm × 75 µm, 2 µm). Mobile phase gradient elution of eluent A (0.1% (v/v) formic acid in water) mixed with eluent B (0.1% (v/v) formic acid in 90/10 acetonitrile/water) was performed as follows: 0–5 min at 4% B, 30 min at 7% B, 60 min at 10% B, 100 min at 15% B, 140 min at 25% B, 180 min at 45% B, 200 min at 65% B, 210–215 min at 96% B, 215.1–240 min at 4% B. Positively charged ions were generated at spray voltage of 2.2 kV using a stainless steel emitter attached to the Nanospray Flex Ion Source (Thermo Fisher Scientific). The quadrupole/orbitrap instrument was operated in Full MS/data-dependent MS2 (Top10) mode. Precursor ions were monitored at m/z 300–1500 at a resolution of 70,000 FWHM (full width at half maximum) using a maximum injection time (Itmax) of 120 ms and an AGC (automatic gain control) target of 1e6. Precursor ions with a charge state of $z = 2$–5 were filtered at an isolation width of m/z 1.6 amu for further HCD fragmentation at 30% normalized collision energy (NCE). MS2 ions were scanned at 17,500 FWHM (Itmax = 120 ms, AGC = 2e5) using a fixed first mass of m/z 120 amu. Dynamic exclusion of precursor ions was set to 30 s, and the underfill ratio was set to 1.0%. The LC–MS/MS instrument was controlled by Chromeleon 7.2, Qexactive HF Tune 2.8 and Xcalibur 4.0 software. Tandem mass spectra were searched against the UniProt databases of *Homo sapiens* (2018/02/02; http://www.uniprot.org/proteomes/UP000005640) and using Proteome Discoverer (PD) 2.2 (Thermo) and the algorithms of Sequest HT (version of PD2.2) and MS Amanda 2.0, and Mascot v2.4.1 (Matrix Science, UK). Two missed cleavages were allowed for the tryptic digestion. The precursor mass tolerance was set to 10 ppm, and the fragment mass tolerance was set to 0.02 Da. Modifications were defined as dynamic Met oxidation and protein N-term acetylation as well as static Cys carbamidomethylation. At least two peptides per protein and a strict false discovery rate (FDR) < 1% (peptide and protein level) were required for positive protein hits. The percolator node of PD2.2 and a reverse decoy database was used for *q*-value validation of spectral matches.

Only rank 1 proteins and peptides of the top scored proteins were counted. Label-free protein quantification was based on the Minora algorithm of PD2.2 using a signal-to-noise ratio >5. Only unique peptides were considered for quantification. Data normalization was deliberately not used in order to display differences on the protein level in the course of the host–pathogen interaction. *P*-values were calculated based on the 2 s *t* test using the abundance values of all biological replicates. *P*-values < 0.05 were considered significant.

**Immunofluorescence.** Monocytes were seeded on poly-L-Lysine-coated 12-mm glass cover slips and infected with opsonized *C. albicans* (MOI 1:1) for 30 min or 1 h in M$_{induction}$ medium at 37 °C, 95% humidity, and 5% $CO_2$. Control monocytes were prepared in a similar manner without infection. After co-incubation cells were fixed with 4% formaldehyde.

For detection of intracellular vesicle or TGF-β1, cells were permeabilized with 0.1% saponin, and blocked with blocking solution (10% FCS, 1% BSA, and 0.1% Tween 20 in DPBS). CD63 was stained with Alexa Fluor 647 anti-human CD63 antibody (BioLegend, cat no. 353015) (1:100), CD14 with Alexa Fluor 488 anti-human CD14 antibody (BioLegend, cat no. 367130) (1:100), and intracellular TGF-β1 with mouse anti-TGF-beta 1 antibody (R&D Systems, cat no. MAB240) (3 µg/mL) and Alexa Fluor 647 goat anti-mouse IgG (H + L) secondary antibody (Thermo Fisher, cat no. A-21235) (1:500). Nucleic acid was stained with sytox orange (Thermo Fischer) (5 µM). Coverslips were fixed on a glass slide, and images were capture with LSM 710 fitted with ZEN 2011 software. Intracellular TGF-β1 or CD63 was measured with ZEN 2011 by measuring the fluorescence intensity inside the cell.

For detection of TGF-β1 vesicles, cells were blocked with blocking solution, and stained with Brilliant Violet 421 anti-human LAP (TGF-β1) antibody (BioLegend, cat no. 349613) (1:100). Coverslips were fixed on a glass slide, and images were capture with LSM 710 fitted with ZEN 2011 software. TGF-β1 vesicles were counted from 212 µm$^2$ of a captured area by subjecting the images to following Image J (v1.52) algorithm:

run("8-bit"); setAutoThreshold("Yen dark"); //run("Threshold…"); setThreshold(60, 255); //setThreshold(60, 255); setOption("BlackBackground", false); run("Convert to Mask"); run("Watershed"); run("Analyze Particles…", " show=Outlines display clear summarize");

For detection of intracellular TGF-β1 in HUVEC cells, HUVEC cells were seeded onto 12-mm glass cover slip. Seeded cells were incubated with BCEV or BCEV$_{Ca}$ isolated from equal amount of blood cells after 6 h. After fixation and blocking the cells, intracellular TGF-β1 was stained with mouse anti-TGF-β1 antibody (R&D Systems, cat no. MAB240) (3 µg/mL) and Alexa Fluor 488 donkey anti-mouse IgG (H + L) secondary antibody (Thermo Fischer, cat no. A32766) (1:500). Nucleic acid was stained with sytox orange (Thermo Fischer) (5 µM). Intracellular TGF-β1 was measured with ZEN 2011 by measuring the fluorescence intensity inside the cell.

PLA assay was used to describe the interaction between soluble βG or iC3b and CD11b. Monocytes were seeded onto poly-L-Lysine-coated 12-mm glass cover slip and infected with opsonized *C. albicans* (MOI of 1:1) or 10 µl/10$^6$ cells of *C. albicans*-derived sβG or 50 µg/mL of iC3b for 15 min. Cells were fixed with 4% formaldehyde, permeabilized with 0.1% saponin, blocked with Duolink blocking solution (Sigma-Aldrich). Cells were treated with mouse anti-(1–3)-β-D-glucan antibody (Biosupplies, cat no. 400-2) (1:200) (for opsonized *C. albicans* and sβG) or mouse anti-iC3b antibody (Calbiochem; cat no. MABF982) (1:200) and rabbit anti-CD11b antibody (Abcam, cat no. 133357) (1:200). PLA assay was performed using the Duolink In Situ Red Starter Kit Mouse/Rabbit (Sigma-Aldrich, cat no. DUO92101) according to the manufacturer's protocol.

The PLA assay was also used to describe the interaction between TGF-β1 and TGFβRII. Monocytes or M1 macrophages or HUVECs were seeded onto 6.7-mm poly-L-Lysine-coated diagnostic slides. Monocytes and macrophages were treated with MEV or MEV$_{CA}$ —isolated from equal amounts of cells—for 15 min, whereas HUVECs were treated with BEV or BEV$_{CA}$—isolated from equal amounts of blood cells—for 30 min. Cells were fixed with 4% formaldehyde, permeabilized with 0.1% saponin and blocked with Duolink blocking solution (Sigma-Aldrich). Cells were treated with mouse anti-TGF-β1 (R&D Systems, cat no. MAB240) (3 µg/mL) and rabbit anti-TGFβRII antibodies (Sigma-Aldrich, cat no. AV44743) (1:200). PLA assay was performed with Duolink In Situ Red Starter Kit Mouse/Rabbit (Sigma-Aldrich, cat no. DUO92101) according to the manufacturer's protocol. PLA fluorescence signals were measured with ZEN 2011.

For the detection of ROS, human monocytes ($1 \times 10^5$) were seeded into 96-well black flat bottom plates (Greiner). Cells were also treated with 50 µg/mL of iC3b, 10 µl/10$^6$ cells of *C. albicans*-derived sβG. The reactions were also incubated with CellROX deep red reagent (Thermo Fischer) (5 µM). After 30 min, ROS was detected by measuring the fluorescence intensity of CellROX using a Safire plate reader (Tecan).

**Live cell imaging.** Monocytes were incubated with opsonized *C. albicans* (MOI 1:1) cells in a 30-mm culture dish in M$_{induction}$ medium, placed in the incubation chamber of LSM710 at 37 °C and 5% $CO_2$. Alexa Fluor 488 anti-human CD14 antibody (BioLegend, cat no. 367130) (1:200) and sytox blue (5 µM) were added to the M$_{induction}$ medium during co-incubation. To observe vesicle generation pictures were taken every 30 s over a time period of 2 h using ZEN 2011.

Monocytes with opsonized *C. albicans* cells (MOI 1:1) in M$_{induction}$ medium were also injected into the flow cell of NanoSight NS300. After monocytes attached to the flow cells, video of scattered light was recorded in real time for 10 min with NTA 3.2 and subsequently analyzed. Human blood was diluted with M$_{induction}$ medium and incubated with *C. albicans* (MOI 1:1) cells in a 30-mm culture dish placed into the incubation chamber of LSM710 at 37 °C and 5% $CO_2$. To observe vesicle formation from monocytes in whole blood, the reaction was treated with Alexa Fluor 488 anti-human CD14 antibody (BioLegend, cat no. 367130) (1:200), Alexa Fluor 647 anti-human CD63 antibody (BioLegend, cat no. 353015) (1:200), and sytox orange (5 μM). To observe TGF-β1 vesicle formation in monocytes, the reaction was treated with Alexa Fluor 488 anti-human CD14 antibody (BioLegend, cat no. 367130) (1:200), Brilliant Violet 421 anti-human LAP (TGF-β1) antibody (BioLegend, cat no. 349613) (1:200), and sytox orange (5 μM). To observe vesicle generation, pictures were taken every 5 min for 1 h using ZEN 2011. After 1 h, images of TGF-β1 vesicles were taken from different places both from control and infected blood. TGF-β1 vesicles were counted from an area of 212 μm$^2$ by subjecting the images to following Image J (v1.52) algorithm:
run("8-bit"); setAutoThreshold("Yen dark"); //run("Threshold…"); setThreshold(60, 255); //setThreshold(60, 255); setOption("BlackBackground", false); run("Convert to Mask"); run("Watershed"); run("Analyze Particles…", " show=Outlines display clear summarize"); close();

### Mouse model of disseminated candidiasis for immunohistochemistry.
Protocols were approved by the responsible Federal State authority and ethics committee (Thüringer Landesamt für Verbraucherschutz, permit number: 03-006/09). For mouse determination of vesicles in liver tissue, BALB/c mice (Charles River) were housed in ventilated cages with free access to water and food. Male and female mice with the age between 9 and 22 weeks were used for the experiments. For infection, *C. albicans* was grown for 12 h at 30 °C in YPD medium, washed three times in sterile PBS, and diluted to the desired concentrations. The mice were infected via the lateral tail vein with $2.5 \times 10^4$ *C. albicans* cfu/g body weight. Liver was collected 24 h post infection, fixed in 10% neutral buffered formalin (Histofix, Carl Roth, Karlsruhe, Germany), embedded in paraffin, and sectioned at 4-μm thickness. For staining the tissues, paraffin-embedded liver sections were deparaffinized by treatments with Roticlear (Carl Roth), 100% ethanol, 90% ethanol subsequently. Sectioned tissues were boiled in 10 mM Na citrate buffer (pH 6.5) for antigen retrieval, and permeabilized with 0.1% saponin (Sigma-Aldrich). Tissues were blocked with normal serum block (Biolegend) and mouse FcR Blocking Reagent (Miltenyi) diluted in antibody diluent (Carl Roth). Tissues were stained for either TGF-β1 with rabbit anti-mouse/human TGF-β1 antibody (Abcam, cat no. ab215715) (1:100) or RhoA with rabbit anti-mouse/human RhoA antibody (Abcam, cat no. ab187027) (1:100) diluted with antibody diluent. Goat anti-Rabbit IgG (H + L) Alexa Fluor 647 was used as secondary antibody (Thermo Fischer, cat no. A-21244) (1:500) diluted with antibody diluent. Sytox orange was used as nuclear staining. Images were capture with LSM 710 fitted with ZEN 2011 software. TGF-β1 vesicles were counted from 212 μm$^2$ of an captured area by subjecting the images to following Image J (v1.52) algorithm:
run("8-bit"); setAutoThreshold("Yen dark"); //run("Threshold…"); setThreshold(40, 255); //setThreshold(40, 255); setOption("BlackBackground", false); run("Convert to Mask"); run("Watershed"); run("Analyze Particles…", " show=Outlines display clear summarize"); close();
RhoA vesicles were counted from 212 μm$^2$ of a captured area by subjecting the images to following Image J (v1.52) algorithm:
run("8-bit"); setAutoThreshold("Yen dark"); //run("Threshold…"); //setThreshold(15, 255); setOption("BlackBackground", false); run("Convert to Mask"); run("Watershed"); run("Analyze Particles…", " show=Outlines display clear summarize"); close();

### Mouse model of disseminated candidiasis for whole-blood vesicle analysis.
All experiments were approved by the local Ethics Committee for Animal Care and Use (BMBWF-66.011/0102-V/3b/2018). C57BL/6 mice (Charles River) were housed in ventilated cages with free access to water and food. Male and female mice with the age between 9 and 22 weeks were used for the experiments. The mice were infected intravenously with $6 \times 10^2$ cfu/g body weight. Blood was collected from submandibular vein in EDTA capillary tubes from anesthetized mouse 24 h post infection. Plasma was isolated by centrifugation at 1377×*g* and stored before use at −80 °C. Plasma was centrifuged at 3000×*g* for 15 min at 4 °C to remove cell debris and diluted with M$_{induction}$ medium. EVs were then isolated using ExoQuick-TC (System Biosciences) according to the manufacturer's protocol. Isolated EVs were stored at −20 °C until use. TGF-β1 from EVs isolated from infected or control 10 μL of plasma was measured by ELISA (R&D Systems, cat no. DY1679).

### EV interaction with macrophage.
M1 macrophage generated from human monocytes were seeded in the M$_{induction}$ medium. Macrophages were treated with 15 ng/mL of LPS-B5 (Invivogen) for induction of inflammation. To asses EV activity, cells were treated with MEV or MEV$_{Ca}$ isolated from equal amounts of monocytes. In all, 5 μg/mL of human anti-TGF-β1-neutralizing antibody (Invivogen, cat no. maba-htgfb-3) was used to block TGF-β1 of MEV$_{Ca}$. In all

cases, the interaction was carried out for 4 h at 37 °C, 95% humidity, and 5% $CO_2$. Released IL-6 was measured in the supernatant using the human IL-6 ELISA Ready-SET-Go! Kit (eBioscience, cat no. 88-7066-22) according to the manufacturer's protocol.

### RNA-seq.
RNA from monocytes infected with opsonized *C. albicans* for 1 h and from untreated monocytes from four different donors was isolated using a total RNA purification kit (Norgen Biotek). Probes from different donors were pooled, and RNA was polyA sequenced using the Illumina method (by LC Sciences). The total RNA quantity and purity were analyzed by Bioanalyzer 2100 using RNA 6000 Nano LabChip Kit (Agilent) with RIN number >7.0. Approximately 10 μg of the total RNA was subjected to isolate Poly (A) mRNA with poly-T oligoattached magnetic beads (Invitrogen). Following purification, the poly(A)- or poly(A) + RNA fractions were fragmented into small pieces using divalent cations under elevated temperature. Then the cleaved RNA fragments were reverse-transcribed to create the final cDNA library in accordance with the protocol for the mRNA-Seq sample preparation kit (Illumina). The average insert size for the paired-end libraries was 300 bp (±50 bp). Paired-end sequencing was performed on an Illumina Hiseq 4000 (lc-bio) following the vendor's recommended protocol. First, Cutadapt and perl scripts in house were used to remove the reads that contained adaptor contamination, low quality bases, and undetermined bases. Then sequence quality was verified using FastQC (http://www.bioinformatics.babraham.ac.uk/projects/fastqc/). HISAT2 was used to map reads to the genome of *Homo sapiens*. The mapped reads of each sample were assembled using StringTie. Then, all transcriptomes from different samples were merged to reconstruct a comprehensive transcriptome using perl scripts and gffcompare. After the final transcriptome was generated, StringTie and Ballgown was used to estimate the expression levels of all transcripts. StringTie was used to calculate expression levels for mRNAs by calculating FPKM. The differentially expressed mRNAs were selected with log2 (fold change) >1 or log2 (fold change) <−1 and with statistical significance (*p*-value < 0.05) by R package Ballgown. DAVID 6.7 was used to analyze the significantly upregulated differentially expressed mRNAs from the samples.

### Ex vivo *C. albicans* infection.
Fresh blood was collected from healthy human volunteers in Na-heparin tubes (BD Biosciences) after informed consent. In total, 1 ml of blood was diluted with 4 ml M$_{induction}$ medium and infected with $1 \times 10^8$ *C. albicans* and incubated for 1 h (for western blot) or 4 h (for qPCR) at 37 °C, 95% humidity, and 5% $CO_2$. To block TGF-β1, blood was treated with 10 μg of human anti-TGF-β1-neutralizing antibody (Invivogen, cat no. maba-htgfb-3), or 300 mM SB 431542 (R&D Systems) before infecting with *C. albicans*.
For mouse experiments, mouse fresh blood was collected immediately after euthanized mouse by cardiac puncture in Na-heparin tubes (BD Biosciences). In total, 100 μl of blood was diluted with 400 μl M$_{induction}$ medium and infected with $1 \times 10^7$ *C. albicans* for 4 h at 37 °C, 95% humidity, and 5% $CO_2$.

### EV interaction with HUVEC cells.
HUVEC cells were treated with EVs isolated from equal numbers of infected (BCEV$_{Ca}$) and control blood cells (BCEV) in the M$_{induction}$ medium for 6 h at 37 °C, 95% humidity, and 5% $CO_2$. In absence of EVs, HUVECs were treated with 0.5 ng/mL or 0.8 ng/mL human recombinant rTGF-β1 (R&D Systems) or *C. albicans* (MOI 10:1). For detection of TGF-β1, HUVECs were treated with BCEV or BCEV$_{Ca}$ for 30 min, unbound vesicles were removed, and after 6 h cells were lysed, and TGF-β1 was measured with human TGF-beta 1 DuoSet ELISA.

### Western blot.
Cells or vesicles were lysed in lysis buffer (25 mM Tris HCl (pH 7.6) + 150 mM NaCl, 1% NP-40 1% sodium deoxycholate, 1% SDS) in presence of 1 mM PMSF (phenylmethanesulfonyl fluoride), and centrifuged at 14,000×*g* to separate membrane and intracellular proteins. Proteins from equal numbers of cells were resolved on 10% SDS-PAGE, and blotted on a nitrocellulose membrane. CD9 protein was detected using mouse anti-human CD9 antibody (Novus Biologicals, cat no. NB500-327) (1 μg/mL) and a goat anti-mouse immunoglobulin-conjugated HRP secondary antibody (Dako, cat no. P044701-2) (1:1000) in nonreducing conditions. Smad2/3 proteins were detected using goat anti-human/mouse Smad2/3 antibody (R&D Systems, cat no. AF3797) (1 μg/mL) and a rabbit anti goat immunoglobulins-conjugated HRP secondary antibody (Dako, cat no. P044901-2) (1:1000) in reducing conditions. Phosphorylated Smad2/3 proteins were detected using rabbit anti-human Phospho-Smad2/3 antibody (R&D Systems, MAB8935) (1 μg/mL) and a goat anti-rabbit immunoglobulin-conjugated HRP secondary antibody (Dako, cat no. P044801-2) (1:1000) in reducing conditions. The Smad7 protein was detected using mouse anti-human/mouse/rat smad 7 antibody (R&D Systems, MAB2029) (1 μg/mL) and a goat anti-mouse immunoglobulin-conjugated HRP secondary antibody (Dako, cat no. P044702-2) (1:1000) in reducing conditions (uncropped blot picture in Supplementary Fig. 7d). CD11b from THP-1 cells was detected using a mouse anti-human CD11b antibody (BioLegend, cat no. 393102) (1:200) and a goat anti-mouse immunoglobulin-conjugated HRP secondary antibody (Dako, cat no. P044701-2) (1:1000) (uncropped blot picture in Supplementary Fig. 7c). TGF-β1 in the vesicle membrane protein fraction was detected using a mouse anti-human/mouse TGF-β1antibody (R&D System, cat no. MAB240) (1 μg/mL) and a goat anti-mouse immunoglobulin-conjugated HRP secondary antibody (Dako, cat no. P044701-2)

(1:1000) in nonreducing conditions (uncropped blot picture in Supplementary Fig. 7e). Factor H in mouse serum was detected using a sheep anti-mouse Factor H Antibody (R&D Systems, cat no. AF4999) (1 μg/mL) and a rabbit anti sheep immunoglobulin-conjugated HRP secondary antibody (Dako, cat no. P0163) (1:1000) in reducing conditions.

**qPCR**. RNA from HUVEC cells or blood cells or monocytes were isolated with Universal RNA/miRNA Purification Kit (EURx) according to the manufacturer's protocol. cDNAs were generated with NG dART RT-PCR kit (EURx) or High-Capacity cDNA Reverse Transcription Kit (ThermoFischer) following the manufacturer's protocol. Comparative qPCR was performed in a StepOnePlus Real-Time PCR System (Applied Biosystems) with PerfeCTa SYBR Green FastMix, Low ROX (Quantabio). For comparative qPCR, 40 cycles were used with initial denaturation at 95 °C for 10 min, denaturation at 95 °C for 15 s, annealing, extension, and reading fluorescence at 60 °C for 1 min. Human *ACTB* was used as housekeeping gene for relative quantification in human blood cell or monocyte qPCR, *LGALS1* was used for HUVEC qPCR, and murine *ACTB* was used for mouse blood cell qPCR. Primers used for the study are listed in Supplementary Table 1. qPCR data were captured and analyzed using StepOne v2.3.

## Data availability

The mass spectrometry proteomics data have been deposited in the ProteomeXchange Consortium under accession code PXD015780 via the PRIDE partner repository with the dataset identifier (data presented in Fig. 2 f–j, Fig. 5 a–c and Supplementary Fig. 4a–d). Tandem mass spectra were searched against the UniProt databases of *Homo sapiens* (2018/02/02; http://www.uniprot.org/proteomes/UP000005640). RNA-seq data are deposited in Gene Expression Omnibus (GEO) with the primary accession code GSE138681 (data presented in Fig. 6 f, g).

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

## Acknowledgements

This work was supported by the 'Deutsche Forschungsgemeinschaft' TransRegio Funginet 124 projects C4 (C.S. and B.J.), C5 (I.D.J.), C6 (P.F.Z. and N.B.) and Z2 (O.K.) SK46/4-1 (CS) and ZI432/19-1. L.D.H. is member of the International Leibniz Research School (ILRS) in Jena, Germany.

## Author contributions

Conceptualization: C.S. and L.D.H.; methodology: L.D.H., C.S., E.A.H.J., M.W., and T.K.; analysis: L.D.H. and T.K.; investigation: L.D.H., E.A.H.J., M.Z.H., M.F.G., T.K., M.W., D.I.P., G.B., I.D.J., N.B., and C.Sp.; writing——original draft: C.S. and L.D.H.; review & editing: C.S., L.D.H., and P.F.Z.; visualization: C.S. and L.D.H.; funding acquisition: C.S.; resources: C.S., P.F.Z., B.J., O.K., I.D.J., M.W., C.Sp., and N.B.; project administration: C.S. and L.D.H.; supervision: C.S. and L.D.H.

## Competing interests

The authors declare no competing interests.
