## [Peer Review File · Nature Communications]

Reviewers' comments:

Reviewer #1 (Remarks to the Author):

The research in this manuscript shows that *C. albicans* infection of monocytes/macrophages from human and murine sources leads to formation of extracellular microvesicles that are notable in TGF- β 1 content. The TGF- β 1 is expressed on the surface of the vesicles and is capable of binding to available TGF- β 1 receptors in macrophages and vascular endothelial cells. Macrophages respond to these vesicles with anti-inflammatory properties and endothelial cells are shown to further produce TGF- β 1 in a feed-forward fashion. The proposed significance is an anti-inflammatory effect on the candida-infected host which is postulated to favor further growth and invasion of the microorganism. In general, the manuscript is well presented and the findings may provide new insight into the pathogenesis of candida. The strength of the work is in the molecular and cell biology that depicts the role of CR3 as a candida-binding receptor and the subsequent kinetics of vesicle formation as well as the thorough physical characterization and determination of the protein content of the vesicles. PLA is well-applied to lend support to several of the important receptor:ligand interactions under study including CR3 binding to both soluble glucan and candida as well as TGF- β 1-expressing vesicles to endothelial cells and macrophages. Some conceptual and technical gaps are listed as concerns for the authors to address but overall the demonstration of TGF- β 1-containing vesicle generation following CR3-dependent uptake of opsonized candida in vitro and in vivo is quite convincing. The overarching limit of the current study is whether these interesting vesicles are important in vivo as explained below.

Major Concerns:

1. A limit to the study is that it does not show the effects of TGF- β 1 MVca on the whole animal which is postulated to develop a state of immunosuppression based on the effect of the vesicles. It seems that the severity of a candida infection needed to cause generalized immune suppression by TGF- β 1-containing MV would have to be extensive in order to produce sufficient circulating MVs to be systemically bioactive. This is in contrast to the speculation that the early stage of candida infection gives rise to these vesicles which predispose the host to increasing candidiasis. This should be considered. The authors do show that vesicles are found in the locally infected liver but there is no evidence that the tissue is somehow immunosuppressed and/or that the Kupffer cells are altered in immune responsiveness. They also show vesicles in the serum of candida infected mice but this is not studied further.

If candida infection does indeed suppress host defense systemically as a result of circulating vesicles generated at the site of fungal infection, then the following should be demonstrable: 1. TGF- β 1 MV should be detectable in the serum of an infected animal and indeed this is shown in Figure 3J; 2. if the concentration of TGF- β 1 MV in the serum is sufficient to cause an immune effect then the serum should demonstrate suppression in an assay such as shown in Fig. 5h and/or Fig. 7h-k; and 3. suppression should be relieved by centrifugal depletion of MVs from the serum and by blockade with anti- TGF- β 1 antibodies. If this can be shown then the biological relevance of the work becomes heightened and should be described in the Abstract. Further proof of concept would be to show that the candida-infected animals are more susceptible to a secondary infectious challenge or manifest a diminished response to a PAMP.

2. The authors study the induction of vesicles by iC3b opsonized candida. The relevance of opsonization is not clear. Given that iC3b is later shown to be unable to induce vesicle formation on its own, it would be important for the authors to show, or describe, the importance of opsonization in this work since it is carried throughout the experiments. For example, does opsonization improve efficiency of phagocytosis and as such is it relevant to magnitude of TGF- β 1-containing MVs?

3. The quantitation of soluble TGF- β 1 (ie, non-vesicular) produced by macrophages following candida uptake is not shown. If it greatly exceeds that contained within vesicles on a per cell basis, would that overshadow the biological importance of the TGF- β 1 contained in the MVca on host

immune status?

4. Soluble vs. yeast beta-glucan. Presentation of soluble glucan to CR3 will not have the same effect on the macrophages as presentation of glucan within a phagocytosable particle such as zymosan, whole glucan particles or yeast (see Qi et al, Blood, 2011, doi: 10.1182/blood-2011-02-339812). Also, soluble glucan does not result in full CR3 activation (O'Brien, JBC, 2012). Moreover, just because extracted glucan added in vitro induces vesicles does not mean that glucan is elaborated in soluble from yeast at sufficient quantities to activate CR3. Therefore this cannot be taken as a mechanism of action without direct proof.

5. iC3b: Supplementary Figure 2 seems to be missing. The reason it is of interest is that it would be relevant to show a comparison of the KEGG enrichment analysis comparing MVca vs. MVca-sbg. This comparison would be more informative than comparing MV vs. MV-ca or ca-sbg. This is so because if CR3 stimulation by soluble glucan is relevant to the formation of MVs as induced by opsonized Candida, then one might suggest that the content of the vesicles is similar or overlapping with the content of MVca-induced vesicles.

Similarly, the authors should demonstrate whether soluble iC3b treatment of macrophages had an effect on CR3 activation (such as CBRM1/5 antibody staining) on human monocytes. If soluble iC3b has no demonstrable effect on the activation state of CR3 then it cannot be determined conclusively one way or the other whether iC3b ligation of CR3 was or was not capable of inducing TGF β -containing vesicles. A PLA would be beneficial.

The authors should show iC3b deposition on apoptotic bodies by anti-iC3b immunostaining. Also, it is news to this reviewer that apoptotic bodies can be complement-opsonized and phagocytosed as a result. Is there precedent in the literature for this? If so it should be cited. Moreover, the authors find a significant role of iC3b on the induction of MVabS by CR3 whereas iC3b is not thought to play a role in inducing MVca in a CR3-dependent mechanism. Can the authors please speculate on the differential stimulatory role of iC3b on opsonized candida vs. opsonized apoptotic bodies?

6. Human immune cells use CR3 as the dominant glucan receptor whereas mice use dectin especially for the phagocytic uptake of candida. Does dectin blockade or depletion from mouse macrophage affect TGF- β 1-MV production?

7. All figure legends should note the statistical methods used and the number of replicate experiments performed.

Minor:

1. Line 89: is describing findings in Fig.1A that vesicles are detected extracellularly. Extracellular vesicles in Fig. 1A are not indicated. Figure 1B might be the appropriate reference to this statement.

2. Line 105: text indicates 10 mins, figure indicates 15 mins.

3. What is the technical difference between immunostaining for CD14 in Figure 1a (cell surface distribution) and 1c (intracellular distribution)? I cannot figure out what that mass is that the arrow is indicating in Figure 1C.

4. Why are the cells in Fig. 1b stained with anti-CD14 noticeably smaller than cells in panel a?

5. The upregulation of complement components in vesicles is surprising. I thought complement was synthesized in the liver.

6. The information to be derived from Supplemental Figure 4a is not evident. The figure actually seems empty.

7. Line 43: current work only studies opsonized apoptotic bodies so this should be indicated as

such in the Abstract.

8. Experiments using Simvastin should be confirmed by antibody blockade.

9. The authors show the vascular endothelial cell to be a target of MV TGF- β 1. Do the authors have any suggestions as to how TGF- β 1 effects vasculature function in the host and whether this effect may be a significant part of the pathobiology of their system?

10. Is there published evidence that the switch from commensal to infection in humans is due to an immunosuppressive mechanism? If so it should be referenced.

Reviewer #2 (Remarks to the Author):

The authors describe a very thorough analysis, in vitro, ex vivo and in vivo, on the TGF β 1 containing extracellular vesicles secreted by monocytes after 1 hour of interaction with pre-opsonized *C. albicans*. Furthermore, they used a wide range of interesting techniques for the EVs analysis, such as dynamic light scattering microscopy and freeze fracture electron microscopy.

Nevertheless, there are some points I would like to be further elucidated.

- In Materials and Methods, you collected and isolated the EVs using ExoQuick-TC and then EVs were stored at -20C. Can you give more details on how do you prevent EVs burst after the freezing – unfreezing cycle.

- Regarding the proteomics analysis, it would be important if you give more details on the number of replicates you used and if you have these data in any repository like PRIDE (<https://www.ebi.ac.uk/pride/archive/>). These data should be publically available for the scientific community. Can you also further detail if you separated the *C. albicans* vesicles from the human EVs and if you found *C. albicans* proteins in the EV's?

- Have you check if there is the secretion of these anti-inflammatory EVs after monocytes interaction with non-opsonized *Candida*, once it seems that CR3 is essential for the release of TGF- β 1-transporting vesicles.

- The observation of a generally anti-inflammatory response was quite striking. Have you performed the monocytes interaction with intracellular facultative bacteria or with *C. glabrata*, which are pathogens that can live inside macrophages and decrease the inflammatory response in order to achieve that? Do you have any hypothesis for the secretion of these EV's after interaction with *C. albicans* and the pathway of immunosuppression you pointed in discussion? Have you tried longer time points to see if this behavior is maintained at longer incubation times?

Reviewer #3 (Remarks to the Author):

Dear Editor,

In this manuscript, Halder et al study the role of extracellular vesicles (EV) that derived from monocytes during the fungal (*Candida albicans*) exposure. The study looks into various aspect of fungal (derived component) exposure to monocytes including the complement receptor 3. Study claims to identify TGF β 1 on vesicles to be responsible for immune modulator in recipient cells. This is an interesting study and this will target the growing EV community. Similar line of thinking has

been proposed earlier (PMID: 27390779, 25287304).

The study present high-end techniques and mouse models, however, it fails with critical aspect i.e. EV preparation. In literature, it is well discussed that association of any protein to EV must be demonstrated in a very stringent manner, since, EV isolation methods can be crude and can be contaminated with free proteins. Most of the guideline can be found in the position paper form the International Society of Extracellular vesicles ISEV (PMID: 30637094).

The biggest concern with the study the method used for EV isolation. But this work can be considered as new-submission if the authors can demonstrate that all/most of the reported TGF β 1 is present on vesicles and address the following concerns.

Comments:

EV Isolation: This work uses Exo-Quick a protein precipitation reagent that pulldown vesicles as well as free proteins present in extracellular space.

EV Purification: No purification was used to separate the free protein from EV. Hence, the downstream functional analyses and conclusion may not remain the same.

Results:

- 1) In Fig 1 A. The incubation time for C.albican and monocyte is 1 hrs (line 83). However, the expression of CD63 was quite low in monocytes alone compared to monocyte with C albican. Can the author explain how the expression of CD63 has increased within 1hr?
- 2) In Fig 1 A. The Nucleic acid stain is from vesicles from monocytes or apoptotic bodies or opsonized C. albicans? Does opsonized of C. albicans induce damage to the membrane of C. albicans?
- 3) Vesicle generation was seen in 10 mins (line 105, Fig 1C). Why do the author think that the patch of CD63 stained structure is vesicles? This looks very different than "vesicles" seen Fig 1b 42 min. Can author clarify this and/or demonstrate that the secreted structures are vesicles?
- 4) Fig 1D shows SEM of the infected cell. It would be nice to see the uninfected monocyte cell. Since the opsonized C. albicans-derived Vesicles will interfere with monocyte cell-derived vesicles. C. albicans could potentially release vesicles (PMID: 25287304, 27390779).
- 5) The point mentioned in question 4 also demands the same thing with DLSM data (Fig 2B) were opsonized C. albicans-derived Vesicles/structure could potentially interfere with vesicles detection.
- 6) Figure 2d and 2e show cryo-image of single vesicles (size: ~250 nm and 450 nm), Image from the larger area must be included to see the heterogeneity (including small size) in different vesicles preparation as demonstrated in Fig 2C and in the literature (Fig 1F of the article (PMID: 26563734).
- 7) Indeed it is very interesting to see the abundance of activation of complement and coagulation cascade as published in macrophages, showing the release of tissue factor in microvesicles (PMID: 31076358). Also, C. albicans activating monocyte is well known in the literature. Since activated monocytes can release lots of soluble and EV associated factors. It is essential to validate the proteins that are really associated with vesicles or secreted freely. The method used in this paper does not help us distinguish vesicles-associated secreted factor Vs free-secreted factors. Authors either can perform density floatation of vesicles or size exclusion chromatography to demonstrate if the floated vesicles have TGF β 1 and IL-6 as well as some of the proteins described in Fig.2I.
- 8) In Fig 3D. The labelling of TGF β 1 is not clear if it is inside or outside? The localization of TGF β 1 +ve vesicles must be shown in Z-stack imaging of cells.
- 9) In general, the author has demonstrated the presence of TGF β 1 it was not clear what proportion of TGF β 1 is active and inactive. This study (<https://doi.org/10.1101/172213>) demonstrate the distribution of active- and inactive -TGF β 1 that can be used to determine the levels.
- 10) In supplementary fig 4c is missing the control with antibody-coated-beads samples to see if DLSM measurement is not picking up protein aggregates of antibodies falling from the beads. Or can perform immunoblotting of TGF- β 1 +ve vesicles were for EV-associated markers.
- 11) It is very interesting to see the signal of TGF β 1 in C. albicans treated cells. However, it would be of great value if we know if the release if TGF β 1 vesicles are dependent on just membrane budding or its endocytic release by MVB fusion to cells surface. TGF β 1 is present in the secreted

granules and release after activation of cells, hence this information becomes crucial. It is unclear if all the secretion happens in such manner or not? This can be demonstrated by the use of RAB27a and RAB27b knock out/down cells.

12) Line 356-357 and fig 4c claim "CD11b KO 356 THP-1 cells completely failed to release TGF- β 1-containing vesicles". CD11b KO alters the expression of TGFB1 in cells as shown fig 4d /4c. This means that the failure of TGFB1 could be due to the absence or reduction of TGF- β 1 in CD11b KO cells. Authors could highlight/provide evidence for the claim that supports the idea that "release TGF- β 1-containing vesicles in CD11b KO".

13) The author used CD11b in context monocytes marker but they are pan-myeloid marker hence specific claims can be avoided in general.

14) A lot of places the topological localization of TGFB1 is not explained and many times confusing if the TGFB1 is inside or on the surface of cells or at the surface of vesicles? This must be addressed.

15) In result section line 406: Maybe I have missed this but the experimental rationale of using apoptotic bodies are not very clear. Authors can clarify these.

16) TGFB1 signalling is quite rapid and activated SMAD proteins can be seen within 5-min of legend-receptor interactions. Author have looked at 4 hr of time point fig 6e. Does it serve any specific purposes? At 4 hrs vesicles, exposure cells could also release TGFB1 in an autocrine manner. Authors already showed the release of TGFB1 by cell within 1-hr of C. Albicans exposure. It would be really informative if the rapid signalling can see in early time point.

17) Line 596: Polymer precipitation can isolate free proteins and EVs together and can give results that cannot explain real EV based processed.

Line 600 use lysis and pelleting the membrane with high-speed centrifugation to demonstrate the association of TGFB1 with EVs. This could be due to co-precipitation. This must be demonstrated using EV material by re-floatation of membrane fraction.

18) In line 714-719: Author discusses that protein on vesicles from monocytes are consistent with some Cd11b and Cd14. Does this mean that vesicles are membrane budded structure of monocyte and not related endocytic origin since it more matches with cell surface markers than endosomal component?

19) Kindly discuss the results in immunes' cell in light of non-infectious condition. (<https://doi.org/10.1101/172213>).

Terminology:

It is confusing to use MVs for Monocyte vesicles. It is a misnomer for Microvesicles (MVs). Kindly use an appropriate term like EV (Extracellular Vesicles).

Technical:

1) The Oder of Figs in text and in the figures is different e.g Fig 6c comes after Fig 6a and 6b.

Kindly make them in order for an easy read. At the current state, it is hard to comprehend.

2) Kindly provide dot blot plot for the graphs for easy interpretation.

3) Which statistical test was used in Fig 7j? The error bars are too large to make this a highly significant (****) as depicted.

Reviewers' comments:

Reviewer #1 (Remarks to the Author):

The research in this manuscript shows that *C. albicans* infection of monocytes/macrophages from human and murine sources leads to formation of extracellular microvesicles that are notable in TGF- β 1 content. The TGF- β 1 is expressed on the surface of the vesicles and is capable of binding to available TGF- β 1 receptors in macrophages and vascular endothelial cells. Macrophages respond to these vesicles with anti-inflammatory properties and endothelial cells are shown to further produce TGF- β 1 in a feed-forward fashion. The proposed significance is an anti-inflammatory effect on the candida-infected host which is postulated to favor further growth and invasion of the microorganism. In general, the manuscript is well presented and the findings may provide new insight into the pathogenesis of candida. The strength of the work is in the molecular and cell biology that depicts the role of CR3 as a candida-binding receptor and the subsequent kinetics of vesicle formation as well as the thorough physical characterization and determination of the protein content of the vesicles. PLA is well-applied to lend support to several of the important receptor:ligand interactions under study including CR3 binding to both soluble glucan and candida as well as TGF- β 1-expressing vesicles to endothelial cells and macrophages. Some conceptual and technical gaps are listed as concerns for the authors to address but overall the demonstration of TGF- β 1-containing vesicle generation following CR3-dependent uptake of opsonized candida in vitro and in vivo is quite convincing. The overarching limit of the current study is whether these interesting vesicles are important in vivo as explained below.

We thank the reviewer for an in-depth evaluation of our work. We also appreciate both the acknowledgement of our work as a new insight into the pathogenesis of *C. albicans* infection as well as conceptual and technical suggestions raised by the reviewer to strengthen the story. We have addressed all the concerns in the revised manuscript.

Major Concerns:

1. A limit to the study is that it does not show the effects of TGF- β 1 MVca on the whole animal which is postulated to develop a state of immunosuppression based on the effect of the vesicles. It seems that the severity of a candida infection needed to cause generalized immune suppression by TGF- β 1-containing MV would have to be extensive in order to produce sufficient circulating MVs to be systemically bioactive. This is in contrast to the speculation that the early stage of candida infection gives rise to these vesicles which predispose the host to increasing candidiasis. This should be considered. The authors do show that vesicles are found in the locally infected liver but there is no evidence that the tissue is somehow immunosuppressed and/or that the Kupffer cells are altered in immune responsiveness. They also show vesicles in the serum of candida infected mice but this is not studied further.

If candida infection does indeed suppress host defense systemically as a result of circulating vesicles generated at the site of fungal infection, then the following should be demonstrable: 1. TGF- β 1 MV should be detectable in the serum of an infected animal and indeed this is shown in Figure 3J; 2. if the concentration of TGF- β 1 MV in the serum is sufficient to cause an immune effect then the serum should demonstrate suppression in an assay such as shown in Fig. 5h and/or Fig. 7h-k; and 3. suppression should be relieved by centrifugal depletion of MVs from the serum and by blockade with anti-TGF- β 1 antibodies. If this can be shown then the biological relevance of the work becomes heightened and should be described in the Abstract. Further proof of concept would be

to show that the candida-infected animals are more susceptible to a secondary infectious challenge or manifest a diminished response to a PAMP.

Response:

In our manuscript we identified *C. albicans* induced vesicles *in vitro* (Fig. 1 and Fig. 2 of revised manuscript), *ex vivo* and *in vivo* (Fig. 3 of revised manuscript). In this first identification of TGF- β 1 transporting vesicles generated in response to *C. albicans* we focused on the characterization of the vesicles, the mechanism of induction and their functional activity. Vesicle size and composition is demonstrated and that TGF- β 1 is transported on their outer surface. In addition, we analyzed in detail the key role of the CR3 receptor in vesicle generation, which was confirmed by blood cells and monocytes from CR3 ko mice that were not able to release these type of vesicles. TGF- β 1-transporting vesicles were identified also *in vivo* in mice upon infection with *C. albicans*. Vesicle transported TGF- β 1 is functionally active and binds to the TGF- β 1 receptor, thereby inducing transcription of TGF- β 1 in endothelial cells and inhibiting cytokine expression in monocytes.

The role of TGF- β 1 in immune suppression in mouse and man is well documented (DOI: 10.1016/j.it.2010.04.002). In our experiments, TGF- β 1 transporting vesicles detected in mouse blood upon infection increased by about 6 fold (Fig. 3j of revised manuscript) and thereby reached physiologically relevant concentrations as previously described (DOI: 10.18632/oncotarget.9416, DOI: 10.1371/journal.pone.0008476). The immune regulating role of transporting vesicles is confirmed in a further experiment where addition of anti-TGF β 1 antibodies to blood reduced suppression and induced a strong IL-1 β response when this blood was infected with *C. albicans* (Fig. 6i of revised manuscript). To provide more evidence for the functional role of generated TGF- β 1 transporting vesicles we performed new *ex vivo* experiments. Centrifugal depletion of EVs from blood was not possible without changing metabolism and activity of the blood cells. As this reviewer acknowledged the clear role of the CR3 receptor in TGF β 1 transporting vesicle generation we now infected whole blood from CR3 ko mice with *C. albicans* and determined the IL-1 β expression by qPCR. No TGF- β 1 transporting vesicles were released in CR3 deficient blood, in contrast to blood from control mice with CR3. Accordingly IL-1 β expression is not dampened by TGF- β 1 in CR3 ko blood cells (Fig. 1 below and Fig. 6j of revised manuscript) and significantly increased. The new results are now shown in Fig 6j in the revised manuscript and we changed the abstract accordingly.

Figure 1: CR3 dependent anti-inflammatory effect

No upregulation of IL-1 β was observed in whole blood of wild-type mice (C57BL/6) infected with *C. albicans*, but significant upregulation of IL-1 β was observed in infected whole blood from CR3-deficient (CD11b KO) mice (B6.129S4-Itgam^{tm1Myd}/J) (***) ($p < 0.001$, unpaired two tailed t-test, n=4 different donors). After 4 h whole blood infection, cells were lysed, and RNA was isolated and subjected to comparative qPCR.

2. The authors study the induction of vesicles by iC3b opsonized candida. The relevance of opsonization is not clear. Given that iC3b is later shown to be unable to induce vesicle formation on its own, it would be important for the authors to show, or describe, the importance of opsonization in this work since it is carried throughout the experiments. For example, does opsonization improve efficiency of phagocytosis and as such is it relevant to magnitude of TGF- β 1-containing MVs?

Previously we reported the importance of opsonization in phagocytosis of *C. albicans* (Fig. 2 below, DOI: [10.3389/fimmu.2016.00671](https://doi.org/10.3389/fimmu.2016.00671)). Opsonization of *C. albicans* leads to enhanced phagocytosis. As phagocytosis is an important part of the immune response we did not want to change this mechanism during host - microbe interaction; therefore opsonization was applied throughout the experiments. In addition Timar et al. (DOI: [10.1182/blood-2012-05-431114](https://doi.org/10.1182/blood-2012-05-431114)) previously reported that opsonization is a prerequisite for vesicle formation.

[REDACTED]

3. The quantitation of soluble TGF- β 1 (ie, non-vesicular) produced by macrophages following candida uptake is not shown. If it greatly exceeds that contained within vesicles on a per cell basis, would that overshadow the biological importance of the TGF- β 1 contained in the MVca on host immune status?

To answer this question we performed new experiments and isolated vesicles from human blood with and without infection with *C. albicans*. Then we determined the amount of TGF- β 1 in both fractions the precipitated vesicles and supernatants . The experiment revealed that TGF- β 1 associates with the vesicles released after infection. The supernatants after vesicle isolation from both uninfected and infected blood contained low amounts of TGF- β 1 (Fig. 3 below and Supplementary Fig. 6a of revised manuscript). The vesicles isolated from infected blood contained TGF- β 1. This result together with the dependency on CR3 for TGF- β 1-transporting vesicle release strongly supports the vesicle-dependent immunosuppression induced by *C. albicans*.

Figure 3: TGF-β1 is bound to vesicles during whole blood infection

A significant increase of TGF-β1-transporting vesicles were observed upon whole blood ex vivo infection with *C. albicans* (**p < 0.001, ****p < 0.0001, unpaired two tailed t-test, and one way ANOVA n=3 different donors). Soluble TGF-β1 was present in low amounts compared to TGF-β1-transported on vesicles.

4. Soluble vs. yeast beta-glucan. Presentation of soluble glucan to CR3 will not have the same effect on the macrophages as presentation of glucan within a phagocytosable particle such zymosan, whole glucan particles or yeast (see Qi et al, Blood, 2011, doi: 10.1182/blood-2011-02-339812). Also, soluble glucan does not result in full CR3 activation (O’Brien, JBC, 2012). Moreover, just because extracted glucan added in vitro induces vesicles does not mean that glucan is elaborated in soluble from yeast at sufficient quantities to activate CR3. Therefore this cannot be taken as a mechanism of action without direct proof.

We have isolated the soluble β-glucan rich fraction from *C. albicans* according to a previously published method (DOI: 10.1016/s0008-6215(99)00049-x, DOI: 10.1111/j.1348-0421.2003.tb03382.x). In addition we have also used commercially available soluble β-glucan from *S. cerevisiae* and both induced TGF-β1-transporting vesicle release. At the moment it is unclear whether soluble β-glucan is released in sufficient amounts from *C. albicans*. However, soluble β-glucan has been demonstrated to be exposed on the *C. albicans* surface (DOI: 10.1091/mbc.E15-06-0355). Indeed, microscopy confirmed that *C. albicans* surface exposed β-glucan interacts with the CR3 receptor (Fig. 4k in revised manuscript).

5. iC3b: Supplementary Figure 2 seems to be missing. The reason it is of interest is that it would be relevant to show a comparison of the KEGG enrichment analysis comparing MV_{ca} vs. MV_{ca-sbg}. This comparison would be more informative than comparing MV vs. MV-_{ca} or _{ca-sbg}. This is so because if CR3 stimulation by soluble glucan is relevant to the formation of MVs as induced by opsonized Candida, then one might suggest that the content of the vesicles is similar or overlapping with the content of MV_{ca}-induced vesicles.

We did not forget supplementary fig. 2 but had previously added all proteomics data in form of supplementary data 2 to the manuscript. Maybe the data were not available to the reviewer. In agreement with this reviewer we now also added the KEGG analysis for MV_{ca} vs. MV_{ca-sbg} to supplementary Fig. 5d of the revised manuscript. A very similar increase of 68 vesicle marker proteins (ExoCarta) in both MV_{ca} vs. MV_{ca-sbg} compared to MEVs is already demonstrated in Fig.5c in the revised manuscript. .

Similarly, the authors should demonstrate whether soluble iC3b treatment of macrophages had an effect on CR3 activation (such as CBRM1/5 antibody staining) on human monocytes. If soluble iC3b has no demonstrable effect on the activation state of CR3 then it cannot be determined conclusively one way or the other whether iC3b ligation of CR3 was or was not capable of inducing TGF β -containing vesicles. A PLA would be beneficial.

To demonstrate that iC3b alone binds to the CR3 receptor new experiments were performed. iC3b was incubated with monocytes and binding to the CR3 receptor was assayed by PLA (Fig. 4k of the revised manuscript). Functional interaction of iC3b with CR3 was investigated by formation of ROS as previously reported (DOI: 10.1016/j.redox.2017.09.017). Binding of iC3b to the CR3 receptor is clearly visible by PLA and signals are very similar to soluble β glucan binding to CR3 (Figure 4a below and Supplementary Fig. 4e of the revised manuscript). Both, iC3b and soluble β glucan activated the CR3 receptor as seen by ROS induction (Fig. 4b below and Supplementary Fig. 4f of the revised manuscript). However, only soluble β -glucan not iC3b induced TGF- β 1-transporting vesicle release

The authors should show iC3b deposition on apoptotic bodies by anti-iC3b immunostaining. Also, it is news to this reviewer that apoptotic bodies can be complement-opsonized and phagocytosed as a result. Is there precedent in the literature for this? If so it should be cited. Moreover, the authors find a significant role of iC3b on the induction of MVabS by CR3 whereas iC3b is not thought to play a role in inducing MVca in a CR3-dependent mechanism. Can the authors please speculate on the differential stimulatory role of iC3b on opsonized candida vs. opsonized apoptotic bodies?

As soon as *C. albicans* enters the blood stream or is in active human serum, the surface becomes opsonized by complement proteins. Subsequently phagocytes take up *C. albicans* much faster (see also Fig. 2). Therefore we mimicked these conditions when we isolated extracellular vesicles. Similarly, opsonization is an essential part of apoptotic cell clearance (DOI: 10.1038/cdd.2009.195) contributing to phagocytosis (DOI: 10.1016/s0014-5793(96)01197-0). In both cases apoptotic bodies and *C. albicans* opsonization (i.e. deposited iC3b) enhances uptake by the monocytes. However, our study does not investigate the role of opsonization on release of TGF- β 1-transporting vesicles, we rather show that iC3b alone binds to CR3 (Figure 4a below and Supplementary Fig. 4e of revised manuscript) stimulates ROS via binding (DOI: 10.1074/jbc.M111.298307) (Figure 4b below and Supplementary Fig. 4f of revised manuscript) and in contrast to soluble β -glucan does not induce TGF- β 1-transporting vesicle release. The difference is explained by different binding sites for iC3b and soluble β -glucan in CR3.

Figure 4: iC3b binds to CR3

(a) Soluble iC3b binds to CR3 on blood monocytes as determined by PLA assay. PLA staining: iC3b-CD11b complexes using CLSM. Scale bar: 10 μ m. (b) iC3b and s β G induce ROS formation in human monocytes within 30 min of co-incubation. ROS formation was followed by measuring fluorescence in presence of CellROX stain in a fluorescence plate reader (*p < 0.05, **p < 0.01, unpaired two tailed t-test, n=3 different donors).

6. Human immune cells use CR3 as the dominant glucan receptor whereas mice use dectin especially for the phagocytic uptake of candida. Does dectin blockade or depletion from mouse macrophage affect TGF- β 1-MV production?

As the involvement of the dectin receptor was not investigated in our current project we cannot really accurately comment on the involvement of the dectin receptor in the process.

7. All figure legends should note the statistical methods used and the number of replicate experiments performed.

Figures and figure legends are updated accordingly.

Minor:

1. Line 89: is describing findings in Fig.1A that vesicles are detected extracellularly. Extracellular vesicles in Fig. 1A are not indicated. Figure 1B might be the appropriate reference to this statement.

Extracellular vesicles attached to *C. albicans* in Figure 1a is now marked with an arrow.

2. Line 105: text indicates 10 mins, figure indicates 15 mins.

The image is representative of several such experiment, the fastest we could determine was 10 min. The figure legend is update to reflect this.

3. What is the technical difference between immunostaining for CD14 in Figure 1a (cell surface distribution) and 1c (intracellular distribution)? I cannot figure out what that mass is that the arrow is indicating in Figure 1C.

During live cell imaging we sometimes observed an accumulation of vesicles. In these cases it was very difficult to detect individual vesicles with an antibody based marker (anti-CD63). Because the vesicles are too small for the available resolution the arrow points to a vesicle accumulation. To show the vesicle accumulation in more detail a magnification of a second spot of accumulated vesicles is shown below (see below Fig. 5 and also Fig. 10)

Figure 5: MEV_{Ca} in whole blood infection

Release of MEV_{Ca} from the infected monocyte was visualized during live cell imaging using CLSM. Staining: CD63, CD14, and nucleic acids. Scale bar: 10 μ m. The monocyte released a high number of CD63 positive vesicles (red dots).

4. Why are the cells in Fig. 1b stained with anti-CD14 noticeably smaller than cells in panel a?

The size marker in Figure 1a is 5 μ M not 10 μ M. We apologize for the mistake and confusion. The figure legend is updated in the revised manuscript.

5. The upregulation of complement components in vesicles is surprising. I thought complement was synthesized in the liver.

Beside liver cells many other cells are known to produce different complement components (DOI: [10.1016/j.smim.2018.03.003](https://doi.org/10.1016/j.smim.2018.03.003), DOI: [10.1016/j.immuni.2013.10.018](https://doi.org/10.1016/j.immuni.2013.10.018)).

6. The information to be derived from Supplemental Figure 4a is not evident. The figure actually seems empty.

Supplemental Fig. 4a or Supplement Fig. 3a of the revised manuscript show vesicles identified with Image J software from CLSM images to quantify the amount of vesicles. The vesicles appear as tiny black dots which are now marked with arrows for better understanding. The control is expected to look empty,

7. Line 43: current work only studies opsonized apoptotic bodies so this should be indicated as such in the Abstract.

The abstract is updated.

8. Experiments using Simvastatin should be confirmed by antibody blockade.

As we could not be sure whether antibody blockade would also activate CR3, we decided to knock out CD11b in THP-1 cells and to use monocytes from a CR3 ko mouse to show CR3 dependency for TGF- β 1-transporting vesicle generation.

9. The authors show the vascular endothelial cell to be a target of MV TGF- β 1. Do the authors have any suggestions as to how TGF- β 1 effects vasculature function in the host and whether this effect may be a significant part of the pathobiology of their system?

We demonstrate that a high number of TGF- β 1 transporting vesicles attach to the endothelial cells (Fig 7a of the revised manuscript) in liver tissue and hypothesize that they induce apoptosis and blood vessel damage (DOI: 10.1158/1541-7786.MCR-11-0507). This process could be one key factor of *C. albicans* to explain entering and growing in liver tissue.

10. Is there published evidence that the switch from commensal to infection in humans is due to an immunosuppressive mechanism? If so it should be referenced.

Occurrence of *C. albicans* infection due to several immunosuppressive conditions is reported in the literature and mentioned in the introduction section. Line 54-58 of the revised manuscript

Reviewer #2 (Remarks to the Author):

The authors describe a very thorough analysis, in vitro, ex vivo and in vivo, on the TGF β 1 containing extracellular vesicles secreted by monocytes after 1 hour of interaction with pre-opsonized *C. albicans*. Furthermore, they used a wide range of interesting techniques for the EVs analysis, such as dynamic light scattering microscopy and freeze fracture electron microscopy.

We thank this reviewer for taking the time for a careful review of our manuscript and for the kind comments and approval of our study. We hope to have addressed each point adequately.

Nevertheless, there are some points I would like to be further elucidated.

- In Materials and Methods, you collected and isolated the EVs using ExoQuick-TC and then EVs were stored at -20C. Can you give more details on how do you prevent EVs burst after the freezing –unfreezing cycle.

We apologize for the confusion in our material and method section. We always thawed the vesicle on ice to avoid unwanted rupture. Also we never repeated freeze and thaw cycles. The method is updated to reflect this.

- Regarding the proteomics analysis, it would be important if you give more details on the number of replicates you used and if you have these data in any repository like PRIDE (<https://www.ebi.ac.uk/pride/archive/>). These data should be publically available for the scientific community. Can you also further detail if you separated the *C. albicans* vesicles from the human EVs and if you found *C. albicans* proteins in the EVs?

Proteomics data as well as RNA seq data are uploaded to the public servers.

For PRIDE reviewer account details:

Username: reviewer70283@ebi.ac.uk

Password: uPDhXQIy

- Have you check if there is the secretion of these anti-inflammatory EVs after monocytes interaction with non-opsonized *Candida*, once it seems that CR3 is essential for the release of TGF- β 1-transporting vesicles.

Opsonization of foreign (like *C. albicans*) or modified self (like apoptotic cells) is a fundamental process in blood to enhance clearance via phagocytosis. To keep natural conditions we performed all experiments with opsonization. In addition Timar et al. (DOI: [10.1182/blood-2012-05-431114](https://doi.org/10.1182/blood-2012-05-431114)) reported previously that opsonization is a prerequisite for vesicle formation. Therefore we performed all experiment with opsonization.

- The observation of a generally anti-inflammatory response was quite striking. Have you performed the monocytes interaction with intracellular facultative bacteria or with *C. glabrata*, which are pathogens that can live inside macrophages and decrease the inflammatory response in order to achieve that? Do you have any hypothesis for the secretion of these EVs after interaction with *C. albicans* and the pathway of immunosuppression you pointed in discussion? Have you tried longer time points to see if this behavior is maintained at longer incubation times?

Although it would be interesting to test vesicle formation in response to different organism we focused on *C. albicans* and investigated apoptotic cells in addition to find the natural inducer for this kind of TGF- β 1-transporting vesicle release. However, from our experiments we hypothesize that TGF- β 1-transporting vesicle are released also in response to other organisms expressing soluble β glucan. How these vesicles are generated and released is topic of current research in our laboratory.

Regarding the incubation time we observed the release of TGF- β 1-transporting vesicle also 4 h after of co-incubation and in infected mice after 1 day (Fig. 3j of revised manuscript) of infection.

Reviewer #3 (Remarks to the Author):

Dear Editor,

In this manuscript, Halder et al study the role of extracellular vesicles (EV) that derived from monocytes during the fungal (*Candida albicans*) exposure. The study looks into various aspect of fungal (derived component) exposure to monocytes including the complement receptor 3. Study claims to identify TGF β 1 on vesicles to be responsible for immune modulator in recipient cells. This is an interesting study and this will target the growing EV community. Similar line of thinking has been proposed earlier (PMID: 27390779, 25287304).

The study present high-end techniques and mouse models, however, it fails with critical aspect i.e. EV preparation. In literature, it is well discussed that association of any protein to EV must be demonstrated in a very stringent manner, since, EV isolation methods can be crude and can be contaminated with free proteins. Most of the guideline can be found in the position paper from the International Society of Extracellular vesicles ISEV (PMID: 30637094).

The biggest concern with the study the method used for EV isolation. But this work can be considered as new-submission if the authors can demonstrate that all/most of the reported TGF β 1 is present on vesicles and address the following concerns.

Comments:

EV Isolation: This work uses Exo-Quick a protein precipitation reagent that pulldown vesicles as well as free proteins present in extracellular space.

EV Purification: No purification was used to separate the free protein from EV. Hence, the downstream functional analyses and conclusion may not remain the same.

We thank the reviewer for his/her time for a comprehensive review. We also acknowledge the appreciation to use high end techniques in our study.

Precipitation of vesicles with ExoQuick is an accepted method for vesicle isolation and used in several publications in highly ranked international journals (DOI: [10.1038/s41551-016-0021](https://doi.org/10.1038/s41551-016-0021), DOI: [10.1038/s41556-019-0299-0](https://doi.org/10.1038/s41556-019-0299-0), DOI: [10.1038/s41556-018-0039-x](https://doi.org/10.1038/s41556-018-0039-x), DOI: [10.1038/s41586-018-0482-7](https://doi.org/10.1038/s41586-018-0482-7), DOI: [10.1038/nnano.2017.134](https://doi.org/10.1038/nnano.2017.134)).

However, to address the key concerns of this reviewer regarding vesicle isolation we have performed several new experiments. First, we tried to answer the question whether the precipitation method used (ExoQuick) precipitates soluble TGF- β 1, and dissolved recombinant TGF- β 1 in the medium used for maintaining the cells. Applying the same ExoQuick method no precipitation of TGF- β 1 was observed and all recombinant TGF- β 1 resided in the supernatant. This demonstrates that the ExoQuick mediated precipitation method cannot precipitate soluble TGF- β 1 (Fig. 6 below and Supplementary Fig . 2c of revised manuscript).

Figure 6: Precipitation of TGF-β1. ExoQuick-TC does not precipitate recombinant TGF-β1 from the solution and TGF-β1 (2 ng/ml) remained in the ExoQuick-TC soluble fraction. TGF-β1 was determined by sandwich ELISA. (**p < 0.001, unpaired two tailed t-test, n=3 different donors).

Secondly: We have also used the most widely used exosome isolation method: ultracentrifugation (PMID: 26044649) to isolate the vesicles released from *C. albicans* infected human monocytes. Similarly to the vesicle precipitation method we detected a substantial increase in TGFβ1-transporting vesicle release from infected monocytes (Fig. 7 below and Supplementary Fig . 2a and 2c of revised manuscript) with this isolation method.

Figure 7: EVs isolated by ultracentrifugation

EVs were isolated from 2×10^6 uninfected or opsonized *C. albicans*-infected monocytes (MEVs or MEV_{Ca}, respectively) by ultracentrifugation. (a) TGF-β1 but not (b) IL-6 was significantly increased in MEV_{Ca} compared to MEVs. Cytokines from MEVs and MEV_{Ca} (each isolated from blood monocytes (2×10^6)), were determined by sandwich ELISA (**p < 0.01, unpaired two tailed t-test, n=3 different donors).

Thirdly: As suggested by this reviewer, vesicles were also isolated using the size exclusion technique as described in PMID: 25279113. Whole blood was infected with *C. albicans* and vesicles were isolated by size exclusion chromatography using Sepharose CL-2B. 14 fractions were collected from the chromatography and tracked for the presence of EVs. Vesicle rich fraction 6 but not fraction 3 or 14 with a low number of vesicles was found to contain substantial amounts of TGF-β1 (Fig. 8 below and Supplementary Fig . 2d-f of revised manuscript).

Figure 8: EVs isolated by size exclusion chromatography

(a) EVs detected by DLSM in different fractions after size exclusion chromatography. EVs were isolated from 1×10^8 blood cells infected with 1×10^8 *C. albicans*. (b) EVs in Fraction 6 have sizes ranging from 20-200 nm. (c) Significant amounts of TGF-β1 were detected in fraction 6 (***) $p < 0.001$, unpaired two tailed t-test, $n=3$ different donors) compared to fraction 3.

Furthermore, immunogold labeling of TGF-β1 in SEM also clearly demonstrates the presence of TGF-β1 on the surface of the EVs (Fig. 9 below and Fig. 3c of revised manuscript).

Figure 9: Gold labeled TGF-b1 on an MEV detected by SEM

TGF-β1 (arrows) was identified on the membranes of MEV_{Ca} but not MEVs as seen by immunogold labeling of TGF-β1 and detection using SEM. Scale bar: 100 nm.

Results:

1) In Fig 1 A. The incubation time for *C.albican* and monocyte is 1 hrs (line 83). However, the expression of CD63 was quite low in monocytes alone compared to monocyte with *C albican*. Can the author explain how the expression of CD63 has increased within 1hr?

In our manuscript we report for the first time that TGF-β1 transporting vesicles are generated and released by monocytes in response to *C. albicans*. How these vesicles are generated and released is topic of current research in our laboratory. An early induction of vesicles in neutrophils was previously reported in response to an *S. aureus* infection (DOI: 10.1182/blood-2012-05-431114) .

2) In Fig 1 A. The Nucleic acid stain is from vesicles from monocytes or apoptotic bodies or opsonized *C. albicans*? Does opsonized of *C. albicans* induce damage to the membrane of *C. albicans*?

In this experiment 0.1% saponin was used as a permeabilization agent after cell fixation, which can easily permeabilize *C. albicans*. Therefore, nucleic acid staining is observed within *C. albicans*.

3) Vesicle generation was seen in 10 mins (line 105, Fig 1C). Why do the author think that the patch of CD63 stained structure is vesicles? This looks very different than "vesicles" seen Fig 1b 42 min. Can author clarify this and/or demonstrate that the secreted structures are vesicles?

We agree with the reviewer that this staining looks very different compared to the nucleic acid stained vesicle in Figure 1b of the revised manuscript. This is explained by the different staining methods used. Unlike nucleic acid staining which can easily diffuse into vesicles, antibody mediated staining during live cell imaging as used here requires more accumulation of the target elements to produce enough signals, as the resolution of confocal laser scanning microscopy is particularly low. Therefore only an accumulation of vesicles is visible by antibody staining like that directed to tetraspanin CD63 which is commonly found on EVs(DOI: [10.1189/jlb.0812391](https://doi.org/10.1189/jlb.0812391)). To show the vesicle accumulation in more detail a magnification of a second spot of accumulated vesicles is shown below (Fig. 9)

Figure 10: MEVs_{Ca} in whole blood infection

Release of MEVs_{Ca} from the infected monocyte was visualized with live cell imaging using CLSM. Staining: CD63, CD14, and nucleic acids. Scale bar: 10 μ m. The monocyte released a high number of CD63 positive vesicles (red dots).

4) Fig 1D shows SEM of the infected cell. It would be nice to see the uninfected monocyte cell. Since the opsonized *C. albicans*-derived Vesicles will interfere with monocyte cell-derived vesicles. *C. albicans* could potentially release vesicles (PMID: 25287304, 27390779).

That the released TGF- β 1 transporting vesicles from infected monocytes were of human origin is demonstrated in many ways such as proteomics, surface staining of a human receptor, etc.). As suggested by this reviewer we now added a new SEM picture showing an uninfected human monocyte which is releasing MEVs (Fig. 11 below and Fig. 1d of revised manuscript)

Figure 11: MEVs released by an uninfected blood monocyte

Scanning electron microscopy (SEM) confirms the presence of MEVs (arrow) released by a human blood monocyte (corner). Scale bar: 1 μ m.

5) The point mentioned in question 4 also demands the same thing with DLSSM data (Fig 2B) were opsonized *C. albicans*-derived Vesicles/structure could potentially interfere with vesicles detection.

To exclude any potential *C. albicans* derived vesicles we also used *C. albicans* derived carbohydrate—soluble β -glucan to induce vesicle release in monocytes. Soluble β -glucan induced similar numbers of TGF- β 1 transporting vesicles (10 fold increase) like *C. albicans*, thus demonstrating that the vesicles are released by monocytes (Fig. 4i and 4j of revised manuscript).

6) Figure 2d and 2e show cryo-image of single vesicles (size: \sim 250 nm and 450 nm), Image from the larger area must be included to see the heterogeneity (including small size) in different vesicles preparation as demonstrated in Fig 2C and in the literature (Fig 1F of the article (PMID: 26563734).

To demonstrate the heterogeneity of vesicles by EM, a new freeze-fracture electron microscopy (FFEM) image is added to Fig. 2e of the revised manuscript which is shown in Figure 12 below.

Figure 12: MEVs_{Ca} released by monocytes as shown by FFEM

Isolated MEVs_{Ca} have different sizes as observed using FFEM. Scale bar: 100 nm.

7) Indeed it is very interesting to see the abundance of activation of complement and coagulation cascade as published in macrophages, showing the release of tissue factor in microvesicles (PMID: 31076358). Also, *C. albicans* activating monocyte is well known in the literature. Since activated monocytes can release lots of soluble and EV associated factors. It is essential to validate the proteins that are really associated with vesicles or secreted freely. The method used in this paper does not help us distinguish vesicles-associated secreted factor Vs free-secreted factors. Authors either can perform density floatation of vesicles or size exclusion chromatography to demonstrate if the floated vesicles have TGF β 1 and IL-6 as well as some of the proteins described in Fig.2I.

As summarized above under comments and as shown in Fig. 8 (and Supplementary Fig . 2d-f of revised manuscript) we performed size exclusion chromatography of the isolated blood vesicles. We collected 14 fractions and demonstrate presence of TGF- β 1 in vesicle fraction 6.

8) In Fig 3D. The labelling of TGF β 1 is not clear if it is inside or outside? The localization of TGF β 1 +ve vesicles must be shown in Z-stack imaging of cells.

Fig. 3d in the manuscript shows images taken during live cell imaging. As monocytes are highly motile, Z stack during live cell imaging is technically not possible. However, according to our observations after 15 min of *C. albicans* infection vesicles were tracked inside the monocyte, and after 45 min, the vesicles were detected extracellularly. This is now marked with an arrow in Fig. 3d revised manuscript.

9) In general, the author has demonstrated the presence of TGF β 1 it was not clear what proportion of TGF β 1 is active and inactive. This study (<https://doi.org/10.1101/172213>) demonstrate the distribution of active- and inactive -TGF β 1 that can be used to determine the levels.

The reviewer is correct, existence of active and inactive TGF- β 1 is described (DOI: 10.1242/jcs.00229). Here we describe vesicles that transport TGF- β 1. This TGF- β 1 is functionally active as it binds to TGFRII or reduces inflammation in blood cells by Smad7 pathway (Fig. 6 e-h of revised manuscript) and induces TGF- β 1 expression in endothelial cells (Fig. 7 h-l of revised manuscript). Whether TGF- β 1 is first inactive on the vesicle and needs activation remains to be determined.

10) In supplementary fig 4c is missing the control with antibody-coated-beads samples to see if DLSM measurement is not picking up protein aggregates of antibodies falling from the beads. Or can perform immunoblotting of TGF- β 1 +ve vesicles were for EV-associated markers.

In the experiment shown in supplementary Fig. 3c of the revised manuscript antibodies were degraded with enzymes (Pluriselect), therefore we excluded to measure protein aggregates. However, to exclude any protein aggregates we performed a new experiment. *C. albicans* induced vesicles released from human blood cells were isolated and subsequently TGF- β 1-transporting vesicles were isolated from the pool of vesicles via anti- TGF- β 1 antibodies coated beads. The obtained vesicles were precipitated and concentrated. These vesicles were recaptured with CD9 coated beads and selected vesicles were measured by DLSM. Size distribution of these CD9 and TGF- β 1-transporting vesicles was 50 to 150 nm (Fig. 13 below and Supplementary Fig. 3 e-g of revised manuscript).

Figure 13: MEVs_{Ca} carry TGF-β1 and CD9

(a) CD9 & TGF-β1-transporting vesicles were tracked by DLSM and show a (b) size distribution predominantly below 100 nm (c) TGF-β1-transporting vesicles were validated for the presence of CD9 using CLSM. Scale bar: 10 μm. TGF-β1-transporting vesicles were precipitated and presence of vesicle marker CD9 confirmed by selection via anti-CD9 antibody coated beads and staining with anti CD9 .

11) It is very interesting to see the signal of TGFβ1 in *C. albicans* treated cells. However, it would be of great value if we know if the release of TGFβ1 vesicles are dependent on just membrane budding or its endocytic release by MVB fusion to cells surface. TGFβ1 is present in the secreted granules and release after activation of cells, hence this information becomes crucial. It is unclear if all the secretion happens in such manner or not? This can be demonstrated by the use of RAB27a and RAB27b knock out/down cells.

We agree that the pathway of TGF-β1 generation and especially how release of the vesicles is organized is of high interest and important. This is topic of the current research in our laboratory and out of the scope of this manuscript.

12) Line 356-357 and fig 4c claim CD11b KO THP-1 cells completely failed to release TGF-containing vesicles;. CD11b KO alters the expression of TGFβ1 in cells as shown fig 4d /4c. This means that the failure of TGFβ1 could be due to the absence or reduction of TGFβ1 in CD11b KO cells. Authors could highlight/provide evidence for the claim that supports the idea that release TGF-containing vesicles in CD11b KO

It is a very interesting question whether TGF-β1 expression determines TGF-β1 vesicle release. We performed new experiments and analyzed TGF-β1 gene expression and also followed the TGF-β1 protein in the cells after *C. albicans* infection. The results demonstrate that *TGF-β1* gene expression is not upregulated in the cell upon infection and that it is the present protein in the cell that is released on vesicles upon infection (Figure 14 and Supplementary Fig. 3h and 3i in the revised manuscript). Therefore, we conclude that the release of TGF-β1 on vesicles is induced by signaling pathways via the CR3 receptor and not by an increase of TGF-β1 concentration.

Figure 14:

(a) Intracellular TGF-β1 staining shows the release of intracellular TGF-β1 from human monocytes in form of vesicles (dots in lower panel) upon *C. albicans* infection. Staining: TGF-β1 and nucleic acids. Scale bar: 10 μm. (b) Intracellular TGF-β1 was measured by ZEN 2011 (**** $p < 0.0001$, unpaired t-test, $n=11$ individual cell from 3 different donors). (c) TGF-β1 transcription is not upregulated but reduced after *C. albicans* infection as shown by comparative qPCR (** $p < 0.001$, unpaired t-test, $n=3$ different donors, non infected blue, infected red).

13) The author used CD11b in context monocytes marker but they are pan-myeloid marker hence specific claims can be avoided in general.

We acknowledged our mistake and changed the term accordingly.

14) A lot of places the topological localization of TGFb1 is not explained and many times confusing if the TGFb1 is inside or on the surface of cells or at the surface of vesicles? This must be addressed.

TGF-β1 is localized on the surface of the vesicle (Fig. 9 and Supplementary Fig. 3c of revised manuscript). This is now explained in the text and marked in the revised manuscript.

15) In result section line 406: Maybe I have missed this but the experimental rationale of using apoptotic bodies are not very clear. Authors can clarify these.

To determine whether *C. albicans* exploits a physiological regulatory mechanism to dampen the immune response to the fungus, we aimed to identify a physiological condition where TGF-β1-transporting vesicles are released by monocytes in the absence of any infection. Apoptosis is a well-organized

process of dead cell clearance without the induction of inflammation. Therefore apoptotic bodies were used.

16) TGFb1 signalling is quite rapid and activated SMAD proteins can be seen within 5-min of legend-receptor interactions. Author have looked at 4 hr of time point fig 6e. Does it serve any specific purposes? At 4 hrs vesicles, exposure cells could also release TGFb1 in an autocrine manner. Authors already showed the release of TGFb1 by cell within 1-hr of C. Albicans exposure. It would be really informative if the rapid signalling can see in early time point.

The proteins collected for western blot analysis in Fig. 6e of the revised manuscript were detected after 1h and not 4h of infection. We apologize as this detail was not mentioned clearly in the method section. The method section is now changed accordingly in the revised manuscript.

17) Line 596: Polymer precipitation can isolate free proteins and EVs together and can give results that cannot explain real EV based processed.

Line 600 use lysis and pelleting the membrane with high-speed centrifugation to demonstrate the association of TGFb1 with EVs. This could be due to co-precipitation. This must be demonstrated using EV material by re-floatation of membrane fraction.

Please see the response to comments pages 11 and 12 above where we show by size exclusion chromatography that TGF- β 1 is associated with the vesicle and that ExoQuick does not precipitate soluble TGF-b1.

18) In line 714-719: Author discusses that protein on vesicles from monocytes are consistent with some Cd11b and Cd14. Does this mean that vesicles are membrane budded structure of monocyte and not related endocytic origin since it more matches with cell surface markers than endosomal component?

Whether TGF- β 1-transporting veiscles are released by membrane budding or by multivesicular bodies is unclear at the moment and needs further investigations.

19) Kindly discuss the results in immunex2019; cell in light of non-infectious condition. (<https://doi.org/10.1101/172213>).

The suggested article is now discussed in the manuscript.

Terminology:

It is confusing to use MVs for Monocyte vesicles. It is a misnomer for Microvesicles (MVs). Kindly use an appropriate term like EV (Extracellular Vesicles).

The reviewer is correct that it is confusing to use the abbreviation for monocytic vesicles. Therefore we decided to use MEV for monocytic extracellular vesicles. The term is now introduced into the revised manuscript.

Technical:

1) The Order of Figs in text and in the figures is different e.g Fig 6c comes after Fig 6a and 6b. Kindly make them in order for an easy read. At the current state, it is hard to comprehend.

As suggested by this reviewer the order of the Figures in the text is changed accordingly in the revised manuscript.

2) Kindly provide dot blot plot for the graphs for easy interpretation.

All graphs are updated in the revised manuscript.

3) Which statistical test was used in Fig 7j? The error bars are too large to make this a highly significant (****) as depicted.

In the referred experiment a high number of individual cells were used, as now depicted in the new graph. Therefore, the statistical significance is high.

Reviewers' comments:

Reviewer #1 (Remarks to the Author):

My compliments to the authors as they have done a thorough job addressing my concerns. My only suggestion is, in the abstract, on line 44, add the word complement to read "human complement-opsonized apoptotic bodies". I will mesh well with CR3 as the operative receptor and clarify that opsonization does not refer to any other opsonins.

Reviewer #2 (Remarks to the Author):

The authors have carefully revised all the points indicated and answered them satisfactorily. The manuscript has improved considerably. I think this work is very interesting and an excellent contribution to the study of the early immune response against *Candida albicans*.

Reviewer #3 (Remarks to the Author):

Reviewer #3 (Remarks to the Author):

Dear Editor,

In this manuscript, Halder et al study the role of extracellular vesicles (EV) that derived from monocytes

during the fungal (*Candida albicans*) exposure. The study looks into various aspect of fungal (derived

component) exposure to monocytes including the complement receptor 3. Study claims to identify TGFb1 on vesicles to be responsible for immune modulator in recipient cells. This is an interesting study

and this will target the growing EV community. Similar line of thinking has been proposed earlier (PMID:

27390779, 25287304).

The study present high-end techniques and mouse models, however, it fails with critical aspect i.e. EV

preparation. In literature, it is well discussed that association of any protein to EV must be demonstrated in a very stringent manner, since, EV isolation methods can be crude and can be contaminated with free proteins. Most of the guideline can be found in the position paper form the International Society of Extracellular vesicles ISEV (PMID: 30637094).

The biggest concern with the study the method used for EV isolation. But this work can be considered as

new-submission if the authors can demonstrate that all/most of the reported TGFb1 is present on vesicles and address the following concerns.

Comments:

EV Isolation: This work uses Exo-Quick a protein precipitation reagent that pulldown vesicles as well as

free proteins present in extracellular space.

EV Purification: No purification was used to separate the free protein from EV. Hence, the downstream

functional analyses and conclusion may not remain the same.

We thank the reviewer for his/her time for a comprehensive review. We also acknowledge the Appreciation to use high end techniques in our study.

Precipitation of vesicles with ExoQuick is an accepted method for vesicle isolation and used in several publications in highly ranked international journals (DOI: 10.1038/s41551-016-0021, DOI: 10.1038/s41556-019-0299-0, DOI: 10.1038/s41556-018-0039-x, DOI: 10.1038/s41586-018-

0482-7, DOI: 10.1038/nnano.2017.134).

Reviewer C1:

Research in EV-field and methods are evolving as we speak. Precipitation of vesicles (with ExoQuick) is crude way of enriching vesicles and does not guarantee the purity of EVs. Moreover, publications in "highly ranked international journal" does not guarantee that ExoQuick is state-of-the-art/correct method for the questions that attempted to address in this manuscript. Nature of current paper talks about the presence of secreted TGF β -1 on/in EVs, hence, the purity of EV holds critically important.

However, to address the key concerns of this reviewer regarding vesicle isolation we have performed

several new experiments. First, we tried to answer the question whether the precipitation method used

(ExoQuick) precipitates soluble TGF- β 1, and dissolved recombinant TGF- β 1 in the medium used for maintaining the cells. Applying the same ExoQuick method no precipitation of TGF- β 1 was observed and

all recombinant TGF- β 1 resided in the supernatant. This demonstrates that the ExoQuick mediated precipitation method cannot precipitate soluble TGF- β 1 (Fig. 6 below and Supplementary Fig . 2c of

revised manuscript).

Reviewer C2:

To circumvent the problem authors have used of 2ng in the recombinant TGF- β 1 (active form) and tested if precipitation or not with ExoQuick. However, EV harbours very low (1-2%) of such active form and remaining 98% remain latent/inactive form (LAP-TGF β 1) (PMID: 31595182; 21098712; 28053020). Recombinant TGF- β 1 used here represent active fraction and hence this is not the appropriate control to be test and demonstrate the ability of Exo-Quick to precipitate latent/inactive-TGF β 1 (which is a major component of EV). Consider appropriate controls to increase the quality of claims made to the manuscript. Experimental design use in following papers by Webber et.al or Shelke et al (PMID: 21098712; 31595182).

Secondly: We have also used the most widely used exosome isolation method: ultracentrifugation (PMID: 26044649) to isolate the vesicles released from *C. albicans* infected human monocytes.

Similarly to the vesicle precipitation method we detected a substantial increase in TGF β 1-transporting vesicle release from infected monocytes (Fig. 7 below and Supplementary Fig. 2a and 2c of revised manuscript) with this isolation method.

Reviewer C3:

Again, ultracentrifugation, a widely used method is only for EV isolation and does not guarantee the purity of vesicles. Coupling of the UC-isolated EV either with density gradient or SEC will substantiate the claim of the paper.

Thirdly: As suggested by this reviewer, vesicles were also isolated using the size exclusion technique as

described in PMID: 25279113. Whole blood was infected with *C. albicans* and vesicles were isolated by

size exclusion chromatography using Sepharose CL-2B. 14 fractions were collected from the chromatography and tracked for the presence of EVs. Vesicle rich fraction 6 but not fraction 3 or 14 with

a low number of vesicles was found to contain substantial amounts of TGF- β 1 (Fig. 8 below and Supplementary Fig. 2d-f of revised manuscript).

Reviewer C4:

I appreciate the effort of authors to choose this method used in this article (PMID: 25279113). This would have certainly clarify the claim. However, the results in (fig 8) does not match with ref (PMID: 25279113). Based on the size of EVs, they should be eluted in fraction 9-12, instead, these result show majority of EVs were seen in 5-7. This could also be due to the way experiments were performed. However, authors should have provided analysis of each of faction for EV enriched

proteins (CD63/81/9) as well as ELISA for TGF β 1. Also DLSM measure particles and does not guarantee vesicles as depicted in Y-axis of Fig 8a. Hence, more information is required to this data to make any certain claim that EVs are really associated TGF β 1.

Furthermore, immunogold labelling of TGF- β 1 in SEM also clearly demonstrates the presence of TGF- β 1 on the surface of the EVs (Fig. 9 below and Fig. 3c of revised manuscript).

Reviewer C5:

I appreciate authors that they were able to demonstrate presence of TGF β 1 on EV with SEM since it's hard to make it work. However it would be nice to have larger field of view to see the frequency of such events. Also the size of EV in SEM is around 300-400 nm this make me wonder if they are derived from outward budding of cell membrane?

Additionally addressing the above 4 comments would help substantiate the claim and bring up the quality of data to Nat Communication standards.

Results:

1) In Fig 1 A. The incubation time for *C.albican* and monocyte is 1 hrs (line 83). However, the expression of CD63 was quite low in monocytes alone compared to monocyte with *C albican*. Can the author explain how the expression of CD63 has increased within 1hr?

In our manuscript we report for the first time that TGF- β 1 transporting vesicles are generated and released by monocytes in response to *C. albicans*. How these vesicles are generated and released is

topic of current research in our laboratory. An early induction of vesicles in neutrophils was previously

reported in response to an *S. aureus* infection (DOI: 10.1182/blood-2012-05-431114) .

Reviewer C6:

This is critical issue that need to address the baseline problem. Since the image is a snapshot of two cells. Authors can perform Flow cytometry analysis of cells with *C. albicans* to clarify this issue of aberrantly rapid and high expression of CD63. How it looks in bulk of cells? Also it seems a intracellular expression of CD63. Mechanism is not expected but appropriate quantification in cell population of intracellular and surface expression of CD63 is important. Other immune cells (mast cells and eosinophils) increase expression of surface CD63 during their activation hence clarification is must to support the presented co-localization data.

2) In Fig 1 A. The Nucleic acid stain is from vesicles from monocytes or apoptotic bodies or opsonized *C. albicans*? Does opsonized of *C. albicans* induce damage to the membrane of *C. albicans*?

In this experiment 0.1% saponin was used as a permeabilization agent after cell fixation, which can

easily permeabilize *C. albicans*. Therefore, nucleic acid staining is observed within *C. albicans*.

Reviewer C7:

Thanks for the clarification.

3) Vesicle generation was seen in 10 mins (line 105, Fig 1C). Why do the author think that the patch

of CD63 stained structure is vesicles? This looks very different than "vesicles" seen Fig 1b 42 min. Can the author clarify this and/or demonstrate that the secreted structures are vesicles?

We agree with the reviewer that this staining looks very different compared to the nucleic acid stained

vesicle in Figure 1b of the revised manuscript. This is explained by the different staining methods used.

Unlike nucleic acid staining which can easily diffuse into vesicles, antibody-mediated staining during live-cell imaging as used here requires more accumulation of the target elements to produce enough signals,

as the resolution of confocal laser scanning microscopy is particularly low. Therefore only an

accumulation of vesicles is visible by antibody staining like that directed to Tetraspanin CD63 which is commonly found on EVs (DOI: 10.1189/jlb.0812391). To show the vesicle accumulation in more detail a magnification of a second spot of accumulated vesicles is shown below (Fig. 9).

Reviewer C8:

Lot of claims were based on image analysis. As discussed the level of CD63 now gets very essential as Fig 1A shows that intercellular-CD63 expression is higher and seems it is expressed with one side of a cell showing polarity in EV-release. Inclusion of western blot (CD63/CD14) isolated EVs at different time point (as used in imaging) would be nice supplement to be included along with imaging data to help substantiate the claim.

4) Fig 1D shows SEM of the infected cell. It would be nice to see the uninfected monocyte cell. Since the opsonized *C. albicans*-derived Vesicles will interfere with monocyte cell-derived vesicles.

C. albicans could potentially release vesicles (PMID: 25287304, 27390779).

That the released TGF- β 1 transporting vesicles from infected monocytes were of human origin is demonstrated in many ways such as proteomics, surface staining of a human receptor, etc.). As suggested by this reviewer we now added a new SEM picture showing an uninfected human monocyte

which is releasing MEVs (Fig. 11 below and Fig. 1d of revised manuscript)

Reviewer C9:

Sure, point well taken.

5) The point mentioned in question 4 also demands the same thing with DLSS data (Fig 2B) were opsonized *C. albicans*-derived Vesicles/structure could potentially interfere with vesicles detection. To exclude any potential *C. albicans* derived vesicles we also used *C. albicans* derived carbohydrate—

soluble β -glucan to induce vesicle release in monocytes. Soluble β -glucan induced similar numbers of

TGF- β 1 transporting vesicles (10 fold increase) like *C. albicans*, thus demonstrating that the vesicles are

released by monocytes (Fig. 4i and 4j of revised manuscript).

Reviewer C10:

Point well explained.

6) Figure 2d and 2e show cryo-image of single vesicles (size: \sim 250 nm and 450 nm), Image from the

the larger area must be included to see the heterogeneity (including small size) in different vesicles

preparation as demonstrated in Fig 2C and in the literature (Fig 1F of the article (PMID: 26563734)).

To demonstrate the heterogeneity of vesicles by EM, new freeze-fracture electron microscopy (FFEM)

image is added to Fig. 2e of the revised manuscript which is shown in Figure 12 below.

Reviewer C11:

Point well done.

7) Indeed it is very interesting to see the abundance of activation of complement and coagulation cascade as published in macrophages, showing the release of tissue factor in microvesicles (PMID: 31076358). Also, *C. albicans* activating monocyte is well known in the literature. Since activated monocytes can release lots of soluble and EV associated factors. It is essential to validate the proteins

that are really associated with vesicles or secreted freely. The method used in this paper does not help

us distinguish vesicles-associated secreted factor Vs free-secreted factors. Authors either can

perform

density floatation of vesicles or size exclusion chromatography to demonstrate if the floated vesicles

have TGF β 1 and IL-6 as well as some of the proteins described in Fig.2I.

As summarized above under comments and as shown in Fig. 8 (and Supplementary Fig . 2d-f of revised

manuscript) we performed size exclusion chromatography of the isolated blood vesicles. We collected

14 fractions and demonstrate the presence of TGF- β 1 in vesicle fraction 6.

Reviewer C12:

Kindly refer to "Reviewer C4" of this document and provide data as requested.

8) In Fig 3D. The labelling of TGF β 1 is not clear if it is inside or outside? The localization of TGF β 1 +ve vesicles must be shown in Z-stack imaging of cells.

Fig. 3d in the manuscript shows images taken during live-cell imaging. As monocytes are highly motile, Z

stack during live-cell imaging is technically not possible. However, according to our observations after

15 min of *C. albicans* infection vesicles were tracked inside the monocyte, and after 45 min, the vesicles were detected extracellularly. This is now marked with an arrow in Fig. 3d revised manuscript.

Reviewer C13:

Point well understood

9) In general, the author has demonstrated the presence of TGF β 1 it was not clear what proportion

of TGF β 1 is active and inactive. This study (<https://doi.org/10.1101/172213>) demonstrate the distribution of active- and inactive -TGF β 1 that can be used to determine the levels.

The reviewer is correct, existence of active and inactive TGF- β 1 is described (DOI: 10.1242/jcs.00229).

Here we describe vesicles that transport TGF- β 1. This TGF- β 1 is functionally active as it binds to TGFRII

or reduces inflammation in blood cells by Smad7 pathway (Fig. 6 e-h of a revised manuscript) and induces

TGF- β 1 expression in endothelial cells (Fig. 7 h-l of revised manuscript). Whether TGF- β 1 is first inactive

on the vesicle and needs activation remains to be determined.

Reviewer C14:

Since this a major claim in the manuscript. Kindly refer to Shelke et al (PMID: 31595182) to determine this ratio.

10) In supplementary fig 4c is missing the control with antibody-coated-beads samples to see if DLSM measurement is not picking up protein aggregates of antibodies falling from the beads. Or can

perform immunoblotting of TGF- β 1 +ve vesicle were for EV-associated markers.

In the experiment shown in supplementary Fig. 3c of the revised manuscript antibodies were degraded

with enzymes (Pluriselect), therefore we excluded to measure protein aggregates. However, to exclude

any protein aggregates we performed a new experiment. *C. albicans* induced vesicles released from

human blood cells were isolated and subsequently TGF- β 1-transporting vesicles were isolated from the

pool of vesicles via anti- TGF- β 1 antibodies coated beads. The obtained vesicles were precipitated and

concentrated. These vesicles were recaptured with CD9 coated beads and selected vesicles were measured by DLSM. Size distribution of these CD9 and TGF- β 1-transporting vesicles was 50 to 150 nm

(Fig. 13 below and Supplementary Fig. 3 e-g of revised manuscript).

Reviewer C15:

Point well explained however this queries would be addressed if include western blotting data DLSM data may not explain all requested question. Additionally, I may have missed why change from CD63 to CD9 in this experiment.

11) It is very interesting to see the signal of TGFb1 in C. albicans treated cells. However, it would be of great value if we know if the release of TGFb1 vesicles are dependent on just membrane budding or its endocytic release by MVB fusion to cells surface. TGFb1 is present in the secreted granules and release after activation of cells, hence this information becomes crucial. It is unclear if all the secretion happens in such manner or not? This can be demonstrated by the use of RAB27a and RAB27b knock out/down cells.

We agree that the pathway of TGF- β 1 generation and especially how the release of the vesicles is organized is of high interest and importance. This is topic of the current research in our laboratory and out of the scope of this manuscript.

Reviewer C16:

I will leave this to the editor to decide. In my opinion basic clarification of TGF b1 is more in Multi-Vesicular bodies derived Exosomes or cell surface budding of Microvesicles. This would very well fall in the scope of Nat Communication.

12) Line 356-357 and fig 4c claim CD11b KO THP-1 cells completely failed to release TGF-containing vesicles;. CD11b KO alters the expression of TGFB1 in cells as shown fig 4d /4c. This means that the failure of TGFB1 could be due to the absence or reduction of TGFin CD11b KO cells. Authors could highlight/provide evidence for the claim that supports the idea that release TGF-containing vesicles in CD11b KO

It is a very interesting question of whether TGF- β 1 expression determines TGF- β 1 vesicle release. We

performed new experiments and analyzed TGF- β 1 gene expression and also followed the TGF- β 1 protein in the cells after C. albicans infection. The results demonstrate that TGF- β 1 gene expression is not upregulated in the cell upon infection and that it is the present protein in the cell that is released on

vesicles upon infection (Figure 14 and Supplementary Fig. 3h and 3i in the revised manuscript). Therefore, we conclude that the release of TGF-b1 on vesicles is induced by signaling pathways via the CR3 receptor and not by an increase of TGF-b1 concentration.

Reviewer C17:

Point well address. This point will further be substantiate if the author would address the point raised in "Reviewer C16".

13) The author used CD11b in context monocytes marker but they are pan-myeloid marker hence specific claims can be avoided in general.

We acknowledged our mistake and changed the term accordingly.

Reviewer:

Point well addressed.

14) A lot of places the topological localization of TGFb1 is not explained and many times confusing if the TGFb1 is inside or on the surface of cells or at the surface of vesicles? This must be addressed. TGF- β 1 is localized on the surface of the vesicle (Fig. 9 and Supplementary Fig . 3c of revised manuscript). This is now explained in the text and marked in the revised manuscript.

Reviewer C18:

Point well addressed.

15) In result section line 406: Maybe I have missed this but the experimental rationale of using apoptotic bodies are not very clear. Authors can clarify these.

To determine whether *C. albicans* exploits a physiological regulatory mechanism to dampen the immune response to the fungus, we aimed to identify a physiological condition where TGF- β 1-transporting vesicles are released by monocytes in the absence of any infection. Apoptosis is a well-organized process of dead cell clearance without the induction of inflammation. Therefore apoptotic bodies were used.

Reviewer C19:

Point well clarified

16) TGFb1 signalling is quite rapid and activated SMAD proteins can be seen within 5-min of legend receptor interactions. Author have looked at 4 hr of time point fig 6e. Does it serve any specific

purposes? At 4 hrs vesicles, exposure cells could also release TGFb1 in an autocrine manner.

Authors

already showed the release of TGFb1 by cell within 1-hr of *C. Albicans* exposure. It would be really informative if the rapid signalling can see in early time point.

The proteins collected for western blot analysis in Fig. 6e of the revised manuscript were detected after

1h and not 4h of infection. We apologize as this detail was not mentioned clearly in the method section.

The method section is now changed accordingly in the revised manuscript.

Reviewer C20:

Point well clarified. However 1 hrs is still long time to release TGFb1 from cells with *C. Albicans*.

Authors can provide western blot for any of the SMAD activation in 5, 15, 30, 60 mins with *C.*

Albicans exposure.

17) Line 596: Polymer precipitation can isolate free proteins and EVs together and can give results that cannot explain real EV based processed.

Line 600 use lysis and pelleting the membrane with high-speed centrifugation to demonstrate the association of TGFb1 with EVs. This could be due to co-precipitation. This must be demonstrated using

EV material by re-floatation of membrane fraction.

Please see the response to comments pages 11 and 12 above where we show by size exclusion chromatography that TGF- β 1 is associated with the vesicle and that ExoQuick does not precipitate soluble TGF-b1.

Reviewer C21:

This need readdress as discussed in "Reviewer C1-C4" of this document.

18) In line 714-719: Author discusses that protein on vesicles from monocytes is consistent with some Cd11b and Cd14. Does this mean that vesicles are membrane budded structure of monocyte and

not related endocytic origin since it more matches with cell surface markers than endosomal

component?

Whether TGF- β 1-transporting vesicles are released by membrane budding or by multi-vesicular bodies

is unclear at the moment and needs further investigations.

Reviewer C22:

This could be easily done using differential centrifugation of conditioned medium supernatant to isolate MV-enriched and exosomes enriched fraction to at least provide speculative evidence in which we can see if MVs or Exo has a higher level of TGF β 1.

19) Kindly discuss the results in immunes' cell in light of non-infectious condition.

(<https://doi.org/10.1101/172213>).

The suggested article is now discussed in the manuscript.

Reviewer C23:

Point well address.

Terminology:

It is confusing to use MVs for Monocyte vesicles. It is a misnomer for Microvesicles (MVs). Kindly use an

appropriate term like EV (Extracellular Vesicles).

The reviewer is correct that it is confusing to use the abbreviation for monocytic vesicles.

Therefore we

decided to use MEV for monocytic extracellular vesicles. The term is now introduced into the revised

manuscript.

Reviewer C24:

Point well address.

Technical:

1) The Oder of Figs in text and in the figures is different e.g Fig 6c comes after Fig 6a and 6b.

Kindly

make them in order for an easy read. At the current state, it is hard to comprehend.

As suggested by this reviewer the order of the Figures in the text is changed accordingly in the revised

manuscript.

Reviewer C25:

Point well address.

2) Kindly provide dot blot plot for the graphs for easy interpretation.

All graphs are updated in the revised manuscript.

Reviewer C26:

Point well address.

3) Which statistical test was used in Fig 7j? The error bars are too large to make this a highly significant (****) as depicted.

In the referred experiment a high number of individual cells were used, as now depicted in the new

graph. Therefore, the statistical significance is high.

Reviewer C27:

Point well address.

Reviewer C28:

4) Kindly use "Particles" instead of "vesicles" in manuscript both in figs and text wherever the data use DLSM.

Reviewers' comments:

Reviewer #1 (Remarks to the Author):

My compliments to the authors as they have done a thorough job addressing my concerns. My only suggestion is, in the abstract, on line 44, add the word complement to read "human complement-opsonized apoptotic bodies". I will mesh well with CR3 as the operative receptor and clarify that opsonization does not refer to any other opsonins.

Reviewer #2 (Remarks to the Author):

The authors have carefully revised all the points indicated and answered them satisfactorily. The manuscript has improved considerably. I think this work is very interesting and an excellent contribution to the study of the early immune response against *Candida albicans*.

Reviewer #3 (Remarks to the Author):

Reviewer #3 (Remarks to the Author):

Reviewer C1:

Research in EV-field and methods are evolving as we speak. Precipitation of vesicles (with ExoQuick) is crude way of enriching vesicles and does not guarantee the purity of EVs. Moreover, publications in highly ranked international journal; does not guarantee that ExoQuick is state-of-the-art/correct method for the questions that attempted to address in this manuscript. Nature of current paper talks about the presence of secreted TGF β -1 on/in EVs, hence, the purity of EV holds critically important.

We agree that the purity of extracellular vesicles is very important. Therefore we performed a number of different experiments and demonstrated by ultracentrifugation, by electron microscopy, by laser scanning microscopy, by particle counter, by size exclusion chromatography, immune absorption, co-staining for TGF- β 1 with CD9 and membrane fraction isolation of vesicles combined with western blotting that TGF- β 1 is located on extracellular vesicles from human blood monocytes. These experiments all revealed the presence of TGF- β 1 on vesicles. In addition TGF- β 1 is very recently -while our manuscript is under review and as reviewer 3 cited several times- described to be located on exosomes (DOI: 10.1080/20013078.2019.1650458) which independently confirmed our data.

Reviewer C2:

To circumvent the problem authors have used of 2ng in the recombinant TGF-beta1 (active form) and tested if precipitation or not with ExoQuick. However, EV harbours very low (1-2%) of such active form and remaining 98% remain latent/inactive form (LAP-TGF β 1) (PMID: 31595182; 21098712; 28053020). Recombinant TGF-beta1 used here represent active fraction and hence this is not the appropriate control to be test and demonstrate the ability of Exo-Quick to precipitate latent/inactive-TGF β 1 (which is a major component of EV). Consider appropriate controls to increase the quality of claims made to the manuscript. Experimental design use in following papers by Webber et.al or Shelke et al (PMID: 21098712; 31595182).

In the revised version of the manuscript we demonstrated that active soluble TGF- β 1 does not precipitate using Exoquick. As suggested by Reviewer 3 we now also precipitated the recombinant

inactive form of TGF- β 1. Similar to the active form of TGF- β 1 ExoQuick did not precipitate the inactive form of TGF- β 1 (LAP- TGF- β 1) (Figure 1 and Supplementary figure 2f in the revised manuscript). The results confirm the previous data that TGF- β 1 (if active or inactive) is associated with the vesicle.

Figure 1: Precipitation of TGF- β 1

Recombinant latent TGF- β 1 in solution did not precipitate using ExoQuick-TC and stayed in the solution after precipitation (** $p < 0.001$, unpaired two tailed t-test, $n=3$ different experiments).

Reviewer C4:

I appreciate the effort of authors to choose this method used in this article (PMID: 25279113). This would have certainly clarify the claim. However, the results in (fig 8) does not match with ref (PMID: 25279113). Based on the size of EVs, they should be eluted in fraction 9-12, instead, these result show majority of EVs were seen in 5-7. This could also be due to the way experiments were performed. However, authors should have provided analysis of each of fraction for EV enriched proteins (CD63/81/9) as well as ELISA for TGF β 1. Also DLSM measure particles and does not guarantee vesicles as depicted in Y-axis of Fig 8a. Hence, more information is required to this data to make any certain claim that EVs are really associated TGF β 1.

The differences between the methods used by us and that described in DOI: 10.3402/jev.v3.23430 are based on individual protocols, experiments and the selection of different samples.

To demonstrate that vesicles are found in TGF- β 1 containing fractions after size exclusion chromatography, we performed a new experiment. Vesicles were isolated from *C. albicans* infected human blood, separated by size exclusion chromatography, and vesicles identified by using two vesicle markers (Figure 2 and Supplementary figure 2h-m in the revised manuscript). Starting the experiment with a higher amount of vesicles as previously, TGF- β 1 was now detected in fractions 5-14 but not in fractions 1-4. TGF- β 1 positive particles in fractions 1-14 were then caught by anti TGF- β 1 antibodies immobilized on an Elisa plate. Subsequently presence of vesicles was evaluated with anti-CD9 and anti HSP90. Sandwich Elisa revealed presence of both CD9 and HSP90 in fractions 7-14, confirming that TGF- β 1 was on vesicles in fractions 7-14. The vesicle size ranged from 70-130 nm. The experiment thus confirmed that TGF- β 1 found in whole blood after *C. albicans* infection is predominantly attached to vesicles.

Figure 2: EVs isolated by size exclusion chromatography

Particle amount (a) and size (b) in different fractions after size exclusion chromatography determined by DLSM. EVs were isolated from 1 ml of human blood infected with 1×10^8 *C. albicans* cells. (c) Identification of TGF-β1 in fractions after size exclusion chromatography using sandwich ELISA. CD9 (d) and HSP90 (e) were determined on TGF-β1 captured particles in different fractions obtained from size exclusion chromatography. CD9 and HSP90 proteins were detected on anti TGF-β1 captured particles by sandwich ELISA. (f) Fractions 7 to 14 contain TGF-β1-transporting vesicles ranging from 70-130 nm as identified by CD9 and HSP90.

Reviewer C5:

I appreciate authors that they were able to demonstrate presence of TGFβ1 on EV with SEM since it's hard to make it work. However it would be nice to have larger field of view to see the frequency of such events. Also the size of EV in SEM is around 300-400 nm this make me wonder if they are derived from outward budding of cell membrane?

Additionally addressing the above 4 comments would help substantiate the claim and bring up the quality of data to Nat Communication standards.

A smaller (~125 nm) TGF-β1-transporting vesicle is now demonstrated which is covered with immunogold labeled TGF-β1 (Figure 3a and Figure 3c in the revised manuscript). Also a larger field of view is presented in Figure 3b.

Figure 3: Gold labeled TGF-β1 on a MEV detected by SEM

TGF-β1 (arrow) is observed on the surface of MEV_{sCa} by immunogold labeling of TGF-β1 in SEM. Scale bars: 200 nm.

Reviewer C6:

This is critical issue that need to address the baseline problem. Since the image is a snapshot of two cells. Authors can perform Flow cytometry analysis of cells with *C. albicans* to clarify this issue of aberrantly rapid and high expression of CD63. How it looks in bulk of cells? Also it seems a intracellular expression of CD63. Mechanism is not expected but appropriate quantification in cell population of intracellular and surface expression of CD63 is important. Other immune cells (mast cells and eosinophils) increase expression of surface CD63 during their activation hence clarification is must to support the presented co-localization data.

In Figure 1a of the manuscript we showed enhanced CD63 expression in monocytes after infection with *C. albicans*. For quantification we determined the CD63 fluorescence in several monocytes (n=30). Enhanced intracellular CD63 expression is confirmed as presented in Figure 4 and Supplementary Figure 2a in the revised manuscript.

Figure 4: Intracellular CD63 in monocytes

C. albicans-infected monocytes increase vesicle generation as detected by the intracellular vesicle marker protein—CD63 in confocal laser scanning microscopy (CLSM). Stained intracellular CD63 was measured by ZEN 2011 (**** $p < 0.0001$, unpaired two tailed t-test, $n=30$ individual cells from 3 different donors).

Reviewer C8:

Lot of claims were based on image analysis. As discussed the level of CD63 now gets very essential as fig 1A shows that intercellular-CD63 expression is higher and seems it is expressed with one side of a cells showing polarity in EV-release. Inclusion of western blot (CD63/CD14) isolated EVs at different time point (as used in imaging) would be nice supplement to be included along with imaging data to help substantiate the claim.

Figure 1A in the manuscript demonstrated intracellular vesicles generation in response to *C. albicans* by CD63 staining of the cells. This result is now confirmed in experiments presented in Figure 4 and Supplementary Figure 2a in the revised manuscript (see above). In addition, the intracellular expression of the vesicle marker—CD9 increased in *C. albicans* infected blood cells as detected by western blot Figure 5 and Supplementary Figure 2b in the revised manuscript. Isolated vesicles were also characterized by proteomics study which revealed 68 vesicle marker proteins Figure 5c in the manuscript. Furthermore, isolated vesicle fractions and captured TGF- β 1 transporting vesicles investigated by size exclusion chromatography showed CD9 and HSP90 vesicle markers (Figure 2 and Supplementary figure 2h-m in the revised manuscript).

Figure 5: Intracellular CD9 increase in whole blood cells

intracellular CD9 protein increased in whole blood cells compared to untreated cells after *C. albicans*-infection (1h). Cells were lysed and intracellular proteins screened for CD9 by Western blot.

Reviewer C12:

Kindly refer to Reviewer C4 of this document and provide data as requested.
Please see answer to C4

Reviewer C14:

Since this a major claim in the manuscript. Kindly refer to Shelke et al (PMID: 31595182) to determine this ratio.

As shown in Figure 6 and Supplementary Figure 2g in the revised manuscript we determined the active/inactive forms of TGF- β 1 on the vesicles. We found predominantly inactive TGF- β 1 on the vesicles. This is in agreement with previous reports (DOI: 10.3109/08977194.2011.595714) and with

the intracellular activation process of TGF- β 1 recently described in DOI: 10.1080/20013078.2019.1650458.

Figure 6

Latent TGF- β 1 was detected from the vesicles in the blood of *C. albicans* infected mice (24 h).

Reviewer C15:

Point well explained however this queries would be addressed if include western blotting data DLSM data may not explain all requested question. Additionally, I may have missed why change from CD63 to CD9 in this experiment.

Although western blotting is a very sensitive method the fractions used in DLSM after 2 rounds of vesicle purification are too low concentrated and would not be sufficient to perform western blotting in parallel. The change of antibodies is based experimentally due to the availability of antibodies for purification.

Reviewer C16:

I will leave this to the editor to decide. In my opinion basic clarification of TGF β 1 is more in Multi-Vesicular bodies derived Exosomes or cell surface budding of Microvesicles. This would very well fall in the scope of Nat Communication.

Please see answer to C22

Reviewer C17:

Point well address. This point will further be substantiate if the author would address the point raised in “Reviewer C16”.

Please see answer to point C22

Reviewer C20:

Point well clarified. However 1 hrs is still long time to release TGF β 1 from cells with *C. Albicans*. Authors can provide western blot for any of the SMAD activation in 5, 15, 30, 60 mins with *C. Albicans* exposure.

The reviewer is correct that 1 h is enough time to release TGF- β 1 transporting vesicles from monocytes. As a proof of principle we followed SMAD activation after 1 h. A substantial upregulation of SMAD is detected in macrophages upon interaction with TGF- β 1 transporting vesicles. Shelke et al. (Journal of extracellular vesicles 2019, vol 8, 1650458) observed minor effects at 30 min and enhanced SMAD2 phosphorylation after 60 min. Therefore we selected 60 min, as we expected lower effects on phosphorylation at earlier time points.

Reviewer C21:

This need readdress as discussed in “Reviewer C1-C4” of this document.

Please read answers to C1 to C4

Reviewer C22:

This could be easily done using differential centrifugation of conditioned medium supernatant to isolate MV-enriched and exosomes enriched fraction to at least provide speculative evidence in which we can see if MVs or Exo has a higher level of TGF β 1.

In supplementary Figure 3e-g of the manuscript vesicles selected via anti CD9 coated beads revealed a particle fraction of predominantly 90 nm. In addition, the major population of TGF- β 1, CD9 and HSP90 vesicles in size exclusion chromatography are 70-130 nm in size (Figure 2 and Supplementary figure 2h-m in the revised manuscript). This size would reflect a population of vesicles called exosomes. In addition, recently Shelke et al. (DOI: 10.1080/20013078.2019.1650458) demonstrated that TGF- β 1 is transported on exosomes. Thus we speculate that TGF- β 1 transporting extracellular vesicles released by monocytes in response to *C. albicans* are exosomes.

Reviewer C28:

4) Kindly use 'Particles'; instead of 'vesicles'; in manuscript both in figs and text

This is now changed in the text.

REVIEWERS' COMMENTS:

Reviewer #3 (Remarks to the Author):

Reviewer:

I congratulate authors who have now provided additional evidence to most of the requested queries. Kindly, make sure the text in the manuscript has good flow to it as the revision of the manuscript could be distorted. Discuss the overall relevance of the MS in light of method and possible pitfalls of EV precipitation. This case it was clear that latent TGF β co-stained with EV but this could not be true with other methods.

Few minor points can be discussed in the MS:

C2

Satisfied with explanation. However, this aspect of active Vs inactive TGF β 1 or growth factor on the surface of EVs general can be included in the discussion.

C4

I thanks authors for this experiments. Even experiments supports particles number and TGF β 1 distribution but the CD9 and HSP looked higher in later fraction. Please include possible explanation / clarification in results or discussion. It would be more wiser if authors could have used sandwiches ELISA to capture EV with any of the membrane proteins and probed for TGF β 1.

REVIEWERS' COMMENTS:

Reviewer #3 (Remarks to the Author):

Reviewer:

I congratulate authors who have now provided additional evidence to most of the requested queries. Kindly, make sure the text in the manuscript has good flow to it as the revision of the manuscript could be distorted. Discuss the overall relevance of the MS in light of method and possible pitfalls of EV precipitation. This case it was clear that latent TGFb co-stained with EV but this could not be true with other methods.

Response: Thank you for the advice of this reviewer. We modified the manuscript to make sure that it has a good flow. All changes in the manuscript are marked in yellow.

Few minor points can be discussed in the MS:

C2

Satisfied with explanation. However, this aspect of active Vs inactive TGFb1 or growth factor on the surface of EVs general can be included in the discussion.

Response: We now included a sentence in the discussion to make this point clearer

C4

I thanks authors for this experiments. Even experiments supports particles number and TGFb1 distribution but the CD9 and HSP looked higher in later fraction. Please include possible explanation / clarification in results or discussion. It would be more wiser if authors could have used sandwiches ELISA to capture EV with any of the membrane proteins and probed for TGFb1.

Response: A possible explanation is now included in the result section.